# Unraveling the neurophysiological correlates of phase-specific enhancement of motor memory consolidation via slow-wave closed-loop targeted memory reactivation

Judith Nicolas [1,2,3], Bradley R. King [4], David Lévesque[5], Latifa Lazzouni[6], Gaëlle Leroux [2], David Wang[7], Nir Grossman[8], Stephan P. Swinnen[1,3], Julien Doyon [6], Julie Carrier [5,9] & Geneviève Albouy [1,3,4] ✉

Memory consolidation can be enhanced during sleep using targeted memory reactivation (TMR) and closed-loop (CL) acoustic stimulation on the up-phase of slow oscillations (SOs). Here, we test whether applying TMR at specific phases of the SOs (up vs. down vs. no reactivation) can influence the behavioral and neural correlates of motor memory consolidation in healthy young adults. Results show that up- (as compared to down-) state cueing results in greater performance improvement. Sleep electrophysiological data indicate that up- (as compared to down-) stimulated SOs exhibits higher amplitude and greater peak-nested sigma power. Task-related functional magnetic resonance images reveal that up-state cueing strengthens activity in - and segregation of - striato-motor and hippocampal networks; and that these modulations are related to the beneficial effect of TMR on sleep features and performance. Overall, these findings highlight the potential of CL-TMR to induce phase-specific modulations of motor performance, sleep oscillations and brain responses during motor memory consolidation.

Memory consolidation is the process occurring offline, between testing sessions, by which labile memory traces become more robust[1]. Seminal rodent work suggests that consolidation relies on the strengthening of mnemonic representations by the spontaneous reoccurrence - during post-learning offline periods - of hippocampal firing patterns associated with the initial encoding[2–4]. This reactivation process has been particularly studied during post-learning sleep and there is consistent evidence that Non-Rapid Eye Movement (NREM)

sleep oscillations such as slow oscillations (SO - high amplitude oscillations in the 0.5–2 Hz frequency band) and spindles (short bursts of oscillatory activity in the 12–16 Hz sigma band) orchestrate the spontaneous occurrence of these hippocampal replays[1,5]. Spontaneous reactivations of task-related brain patterns have since been observed during post-learning sleep in humans after both declarative and motor learning (see[6] for a review). Building upon these observations, the field has recently seen a surge of research examining whether experimental

[1]Department of Movement Sciences, Movement Control and Neuroplasticity Research Group, KU Leuven, Leuven, Belgium. [2]Université Claude Bernard Lyon 1, CNRS, INSERM, Centre de Recherche en Neurosciences de Lyon CRNL U1028 UMR5292, 69500 Bron, France. [3]LBI—KU Leuven Brain Institute, KU Leuven, Leuven, Belgium. [4]Department of Health and Kinesiology, College of Health, University of Utah, Salt Lake City, UT, USA. [5]Center for Advanced Research in Sleep Medicine, Montreal, QC, Canada. [6]McConnell Brain Imaging Centre, Department of Neurology and Neurosurgery, Montreal Neurological Institute, McGill University, Montreal, QC, Canada. [7]Elemind Technologies Inc Massachusetts Institute of Technology, Cambridge, MA, USA. [8]Faculty of Medicine, Department of Brain Sciences, Imperial College London, London, UK. [9]Department of Psychology, Université de Montréal, Montreal, QC, Canada. ✉e-mail: Genevieve.albouy@kuleuven.be

interventions can induce such reactivations in the human brain and eventually enhance the memory consolidation process[7–9].

An experimental intervention that has shown promise to enhance memory consolidation is Targeted Memory Reactivation (TMR)[9]. TMR is a non-invasive procedure which consists of replaying, offline, sensory stimuli that were previously associated to the memory items during initial encoding[9,10]. Auditory TMR applied during post-learning NREM sleep has been consistently shown to boost both declarative and motor memory consolidation in healthy young adults e.g.,[11–15] and this process is thought to be mediated by the modulation of SO[14] and spindle[12,16] characteristics as well as their coupling[14]. Inspired by studies showing that auditory clicks delivered in a closed-loop (CL) fashion at the up-state of the SO can optimize declarative memory consolidation e.g.,[17], recent studies have applied TMR at the SO up-phase (as compared to the down-phase) in an attempt to further optimize consolidation. Such CL-TMR interventions have been shown to increase SO and sigma band power following cues presented at the up- as compared to the down-phase of the SOs[18,19] or as compared to a control night without stimulation[20]. These studies show an overall memory advantage following up-state TMR[18–20], albeit performance does not

always differ from all other stimulation conditions (e.g., from down-[18] or no-stimulation[19]). Altogether, studies causally linking memory consolidation to the specific SO phase during which memories are experimentally reactivated are sparse in the declarative memory domain and are non-existent in the motor memory domain. Additionally, the effect of slow-oscillation CL-TMR on the neurophysiological processes underlying memory reactivation and memory retention are poorly understood in both memory domains.

In this pre-registered study (https://osf.io/dpu6z), we used functional magnetic resonance imaging (fMRI) during task practice and electro-encephalography (EEG) during post-learning sleep to address these knowledge gaps and provide a comprehensive characterization of the neurophysiological processes supporting the effect of slow-oscillation CL-TMR on motor memory consolidation. Briefly, in a within-subject design, 31 young healthy participants (15 self-declared females) learned 3 different motor sequences that were each associated to one specific sound during learning while their brain activity was recorded with fMRI (Pre-night, Fig. 1a). During the subsequent post-learning night of sleep that was monitored with EEG (Night, Fig. 1a), SOs were detected in real-time during NREM sleep and

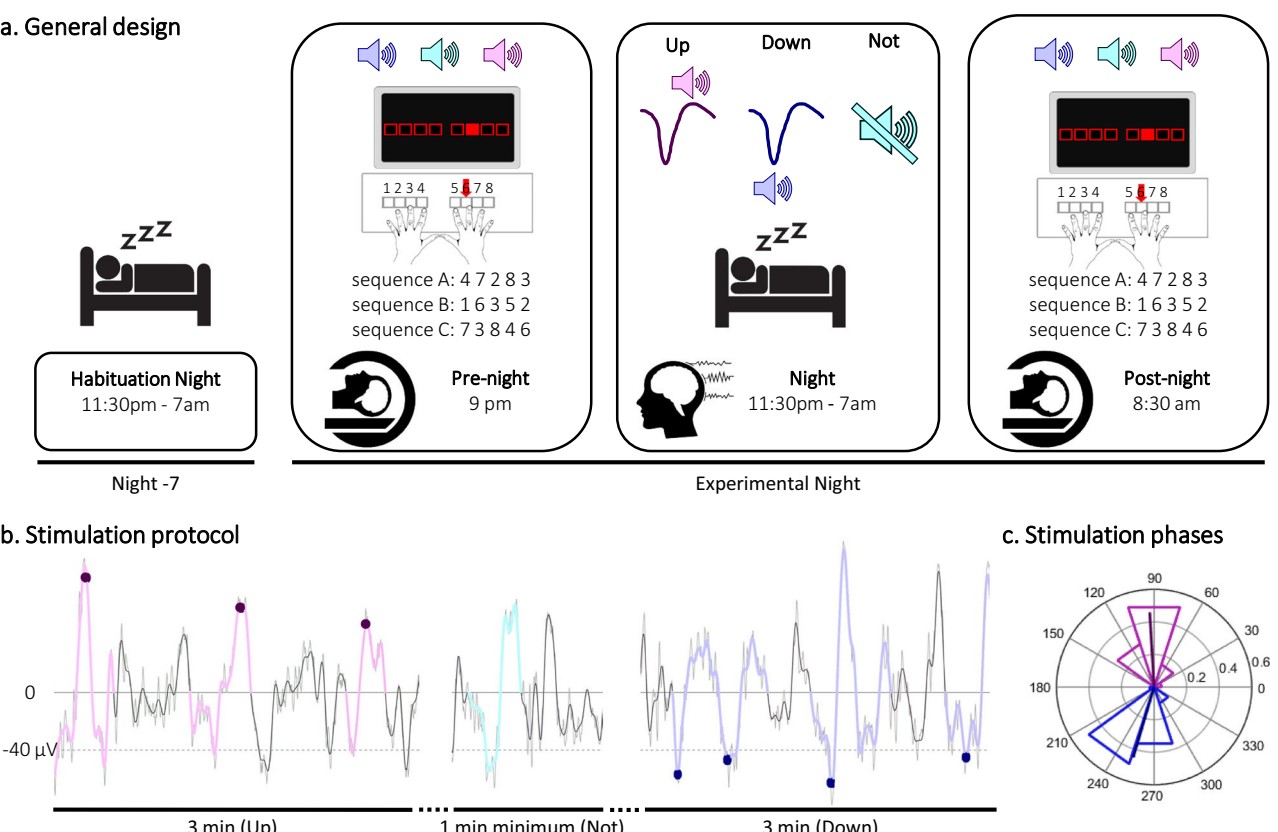

**Fig. 1 | Experimental protocol. a** General design. Following a habituation night that was completed approximately one week prior to the experiment, 31 participants (within-subject design) underwent a pre-night motor task session in the scanner, a full night of sleep in the sleep lab monitored with polysomnography during which slow-oscillation (SO) closed-loop targeted memory reactivation (CL-TMR) was applied, and a post-night retest session in the scanner. During the motor task (pre-night and post-night sessions), three movement sequences were performed (sequences A, B and C whereby 1 and 8 correspond to the right and left little fingers, respectively) and were cued by three different 100-ms auditory tones. Two of these sounds (memory cues) were replayed during post-learning sleep at specific phases of the SO (up vs. down, see panel b for details) while the third sound was a control condition, which was not replayed during the night. Panel images adapted from[14]. **b** Stimulation protocol. Sleep was recorded with EEG and the online SO detection algorithm (see "Methods") was launched whenever the participant was in

NREM sleep. The sounds associated to the up (or down)-reactivated sequence were then played on the peak (or trough) of the SOs within up (or down)-stimulation intervals (magenta and blue dots, respectively) that alternated with rest (no stimulation) periods. The colored SOs in each panel represent the SOs detected offline and used to compute, a posteriori, the accuracy of online detection and to detect SOs during rest intervals for further analyses (see "Methods"). The accuracy of the online SO detection was 82.2% [95CI: 79.6–84.8] across conditions and the number of true SOs stimulated did not differ between stimulation conditions (t = 1.65, df = 29, p = 0.11, Cohen's d = 0.3, see methods and Table S5 for further information). **c** Stimulation phases (n = 31). Density plot of phases in degrees (and mean direction and resultant vector length in bold) of the phase at which up (magenta) and down (blue) auditory cues were sent. The mean phase of up-stimulation was 92.62° [95CI: 92.10–93.13] and of down-stimulation was 253.48° [95CI: 253.11–253.87].

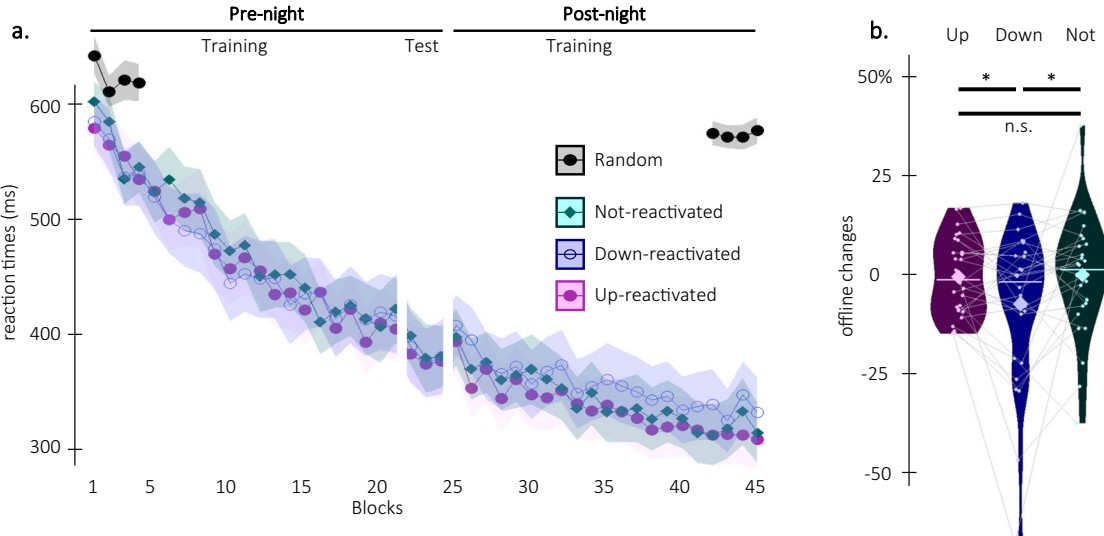

**Fig. 2 | Behavioral results (*n* = 28). a** Performance speed. Grand average across participants of median reaction time in ms for each practice block during the pre- and post-night sessions (±standard error in shaded regions) for the up-reactivated (magenta circles), the down-reactivated (blue empty circles), and the not-reactivated (green diamonds) sequences and for the random serial reaction time task performed at the start and end of the experiment (black overlay, see methods and supplements for corresponding results). A short break is introduced between the training and test runs during the pre-night session in order to minimize the confounding effect of fatigue on end-of-training performance[64] (see methods). The three motor sequences were learned to a similar extent during the pre-night session (see Supplementary results; note that performance improvement is reflected by a decrease in response times). **b** Offline changes in performance speed (% change between the average of the three blocks of pre-night test and the first three blocks of post-night training) averaged across participants for the up-reactivated (magenta), the down-reactivated (blue) and the not-reactivated (green) sequences.

Results of the rmANOVA show a main effect of Condition (F(2,54) = 3.88, *p* = 0.027 (0.034 sphericity corrected), η² = 0.13) whereby offline changes in performance speed were greater for the up- and not-reactivated as compared to the down-reactivated sequence (*: *p* < 0.05; improvement in performance from pre- to post-night sessions is reflected by a positive change). Note that the main effect of condition is marginally significant when excluding the participant showing the extreme datapoint in the down condition (F(2,52) = 2.73, *p* = 0.074, η² = 0.13). This data point was not excluded from the primary analysis as it did not meet any criterion for outlier exclusion as defined in our pre-registration. Violin plots: median (horizontal bar), mean (diamond), the shape of the violin plots depicts the kernel density estimate of the data. Colored points represent individual data, jittered in arbitrary distances on the x-axis within the respective violin plot to increase perceptibility. For each individual, performance on the different conditions is connected with a line between violin plots. n.s non-significant.

auditory cues that were associated to the motor sequence learning task (i.e., memory cues) were delivered to specific phases of the SO reflecting either high or low brain excitability. Specifically, one sound (e.g., memory 1 cue) was played at the peak of the SO (up-reactivated condition), another sound (memory 2 cue) was played at the trough of the SO (down-reactivated condition), while the last sound (memory 3 cue) was not replayed (not-reactivated, control, condition). To assess consolidation, motor task performance was retested on the three different conditions in the fMRI scanner the next morning (Post-night, Fig. 1a). Our main results showed that consolidation[18], SO amplitude[17,19,21], sigma band power[17,19] and task-related brain responses in hippocampo- and striato-cortical networks[12,22] were boosted by SO-up-phase TMR as compared to SO-down-phase TMR.

## Results

### The effect of TMR on motor performance depends on the phase of the stimulated SO

We tested whether the stimulation conditions (up-, down- and not-reactivated) influenced the behavioral index of motor memory consolidation, i.e., the offline changes in performance speed observed between the pre-night test session and the beginning of the post-night training session (see Fig. 2a; note that in panel a, performance improvement is reflected by a decrease in response time -RT). Results show that offline changes in performance speed differed depending on the phase of the stimulated SO (Condition effect (F(2,54) = 3.88, *p* = 0.027 (0.034 sphericity corrected)), η² = 0.13; Fig. 2b, *n* = 28; note that in panel b, a positive offline change in performance reflects overnight performance improvement). Specifically, offline changes in performance speed were greater for both the up- and not-reactivated

sequences as compared to the down-reactivated sequence (up vs. down: t = 2.32, df = 27, *p* = 0.014 (0.035 FDR-corrected), Cohen's d = 0.44; down vs. not: t = −2.09, df = 27, *p* = 0.023 (0.035 FDR-corrected), Cohen's d = 0.39). In contrast, performance speed did not significantly differ between the up- and not-reactivated sequences (up vs. not: t = −0.24, df = 27, *p* = 0.59 (0.59 FDR-corrected), Cohen's d = 0.045). Altogether, these behavioral results indicate that TMR differently altered the fate of the motor memory traces depending on the phase (up vs. down) of the SO during which reactivation was applied.

### SO-up-phase TMR enhances both SO amplitude and sigma oscillations

EEG data collected during the TMR episode were analyzed to test whether (the phase of the) stimulation modulated the characteristics of electrophysiological markers critically involved in motor memory consolidation[23] and reactivation[14] during sleep, i.e., SOs, spindle events, as well as sigma oscillations, taken as a proxy of spindle activity.

To examine the effect of stimulation on SO characteristics, we computed - for each of the 6 EEG channels (Fz, Cz, Pz, Oz, C3, C4) - event-related potentials locked to the trough (and see methods, Supplementary Information Supplementary section 1 and Fig. S1 for auditory cue-locked analyses) of the (1) up-stimulated SOs (detected online on Fpz and validated offline during up-stimulation intervals), (2) down-stimulated SOs (detected online on Fpz and validated offline during down-stimulation intervals) and (3) not-stimulated SOs (detected offline on Fpz during epochs free of stimulation, see Fig. 1b for a depiction of stimulation epochs). In this analysis, cluster-based permutations identified clusters on the basis of temporal and spatial (channel) adjacency (see "Methods"). Results are presented in Fig. 3

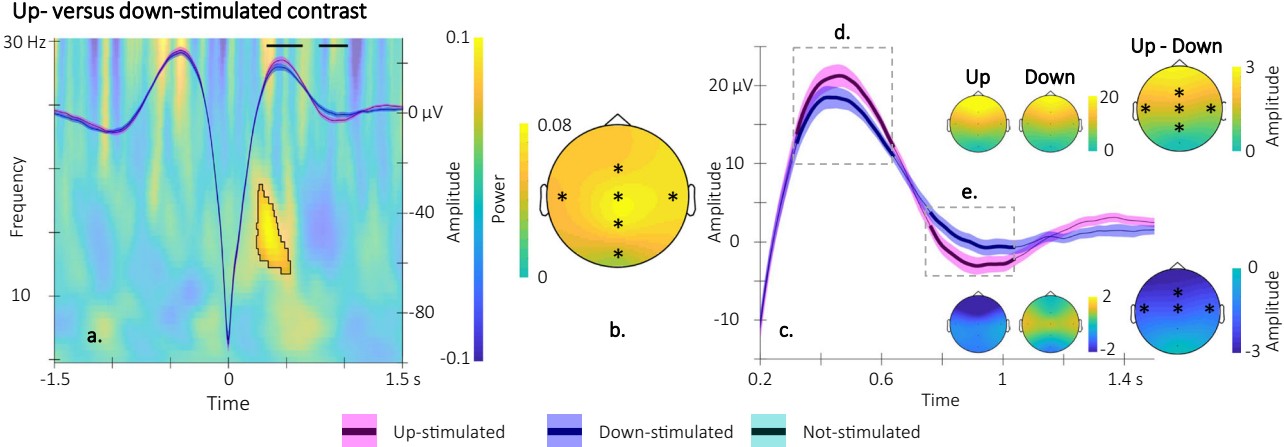

**Fig. 3 | Electrophysiological results of the up- vs. down-stimulated contrast (*n* = 30).** Participants' sleep was recorded using a 6-channel EEG montage during the night following learning. Statistical comparisons were performed using non-parametric cluster-based permutations. **a** Time-frequency representation (TFR) of the difference in power modulation illustrated at Cz around the trough of the up- and down-stimulated slow oscillations (SOs) on which the grand-average of the SOs illustrated on Fz is super-imposed (magenta(blue): up(down)-stimulated SO). Black lines represent the adjacent time points of the significant spatio-temporal clusters showing a difference in SO amplitude between the two trough-locked potentials (see **d**, **e** for the spatial dimension of the clusters and Fig. S2b in Supplementary information for channel level cluster depiction). Results show that the up-stimulated SO presented greater amplitude at their peak (from 0.33 to 0.64 s post-trough, cluster *p* = 0.0040; Cohen's d = 0.67) followed by a deeper deflection (from 0.78 to 1.03 s post-trough, cluster *p* = 0.0040; Cohen's d = −0.60). Further, the area highlighted in the TFR represents the adjacent time-frequency points of the significant spatio-temporal-frequency cluster showing a difference in power between conditions. Sigma power nested in the ascending phase of the up-stimulated SOs was greater than for the down-stimulated SOs (cluster *p* = 0.0080; Cohen's d = 0.66 from 0.25 to 0.4 s post-trough and from 12 to 17 Hz; and see (**b**) for the spatial dimension of the cluster as well as Fig. S2b in Supplementary information for channel level cluster depiction). **b** Topography of the significant sigma power modulation. **c** Zoom on the trough-locked SO peak and deflection at Fz (same color code as **a**) showing the significant differences in amplitude between up and down conditions (see text). **d**, **e** Topography of the significant differences in amplitude at the trough-locked SO peak and deflection. SO shaded areas represent ±standard error; (*) represents the electrodes included in the significant clusters.

(up vs. down comparison) and Fig. 4 (stimulated vs. not-stimulated comparisons) on a channel included in all the significant spatio-temporal clusters observed across contrasts (Fz) and results on all other channels are reported in Fig. S2a of the Supplementary information. Results of the up vs. down comparison (Fig. 3) indicated two significant spatio-temporal clusters in which the phase of the stimulation specifically influenced SO amplitude. Specifically, up-stimulated - as compared to down-stimulated - SOs showed (1) greater amplitude around the peak of the SO in a spatial cluster including all electrodes except Oz (positive cluster *p* = 0.0040; Cohen's d = 0.67) and (2) deeper deflection post-peak in a spatial cluster including frontal and central electrodes (negative cluster *p* = 0.0040; Cohen's d = −0.60). Figure 3a depicts the grand average of the SOs (super-imposed on a time frequency representation of the difference in power modulation, see below) for up and down conditions in which the horizontal black lines represent the significant temporal cluster. Figure 3c presents a zoomed inset of the temporal clusters showing a significant modulation of amplitude, the topography of these effects as well as the spatial dimension of the clusters (Fig. 3d, e).

Similar results - albeit on larger time windows - were observed when comparing both up- and down-stimulated versus not-stimulated SOs (up vs. not, Fig. 4a-1 and 4a-3 for zoomed inset; Fig. 4a-4: positive cluster *p* = 0.0020, Cohen's d = 1.12; Fig. 4a-5: negative cluster *p* = 0.0060, Cohen's d = −0.70; down vs. not, Fig. 4b-1 and 3c-3 for zoomed inset; Fig. 4b-4: positive cluster *p* = 0.0020, Cohen's d = 1.20; Fig. 4b-5: negative cluster *p* = 0.0040, Cohen's d = −0.57) but these effects were more pronounced during up- as compared to down-stimulation as shown in Fig. 3.

Note that analogous results were also observed using SO density metrics extracted from the stimulated and not-stimulated intervals (see Fig. S3a in Supplementary information showing greater density during up-stimulated intervals as compared to down-stimulated and not-stimulated intervals).

To investigate the effect of stimulation on oscillatory brain activity (and sigma oscillations in particular), we performed time-

frequency analyses locked to the trough of the stimulated and not-stimulated SOs on each EEG channel. Here, cluster-based permutation analyses identified clusters on the basis of temporal, frequency and spatial adjacency (see "Methods"). Results are presented in Figs. 3, 4 on a channel included in all the significant spatio-temporal-frequency clusters observed across contrasts (Cz) and results on all other channels are reported in Fig. S2a of the Supplementary information. Results of the up vs. down analyses (Fig. 3) indicated one significant spatio-temporal-frequency cluster in which sigma power was greater in the ascending phase of the up-stimulated SOs as compared to the down-stimulated SOs on all electrodes (cluster *p* = 0.0080; Cohen's d = 0.66). Figure 3a depicts a time-frequency representation of the significant cluster and Fig. 3b shows the topography of this difference. Results of the stimulated vs. not-stimulated analyses (Fig. 4) showed that oscillatory activity in the 5–18 Hz frequency range was lower in the descending phase of both up- and down-stimulated – as compared to not-stimulated – SOs on all electrodes (up vs. not, Fig. 4a, cluster *p* = 0.002, Cohen's d = −1.30; down vs. not, Fig. 4b, cluster *p* = 0.002, Cohen's d = −0.94). Power in lower frequencies (5–10 Hz) was greater for the down-, compared to the not-, stimulated conditions from 0.8 to 1.5 s post SO trough in a cluster including all electrodes (down vs. not, Fig. 4b, cluster *p* = 0.0020, Cohen's d = 0.90).

Analyses based on sleep spindle events detected from the sti-mulated and not-stimulated intervals show that spindle frequency and amplitude were unaffected by the stimulation while spindle density was lower during both up- and down-stimulated as compared to not-stimulated intervals, irrespective of the stimulation condition (see Fig. S3b–d in Supplementary information).

Altogether, these results indicate that up-phase, as compared to down-phase, CL-TMR resulted in enhanced SO density and amplitude as well as a stronger sigma power during the ascending phase of the SO. In contrast, oscillatory activity including the sigma band was decreased during the descending phase of the up- and down-stimu-lated, as compared to the not-stimulated. SOs and overall spindle density was lower under stimulation, irrespective of its phase.

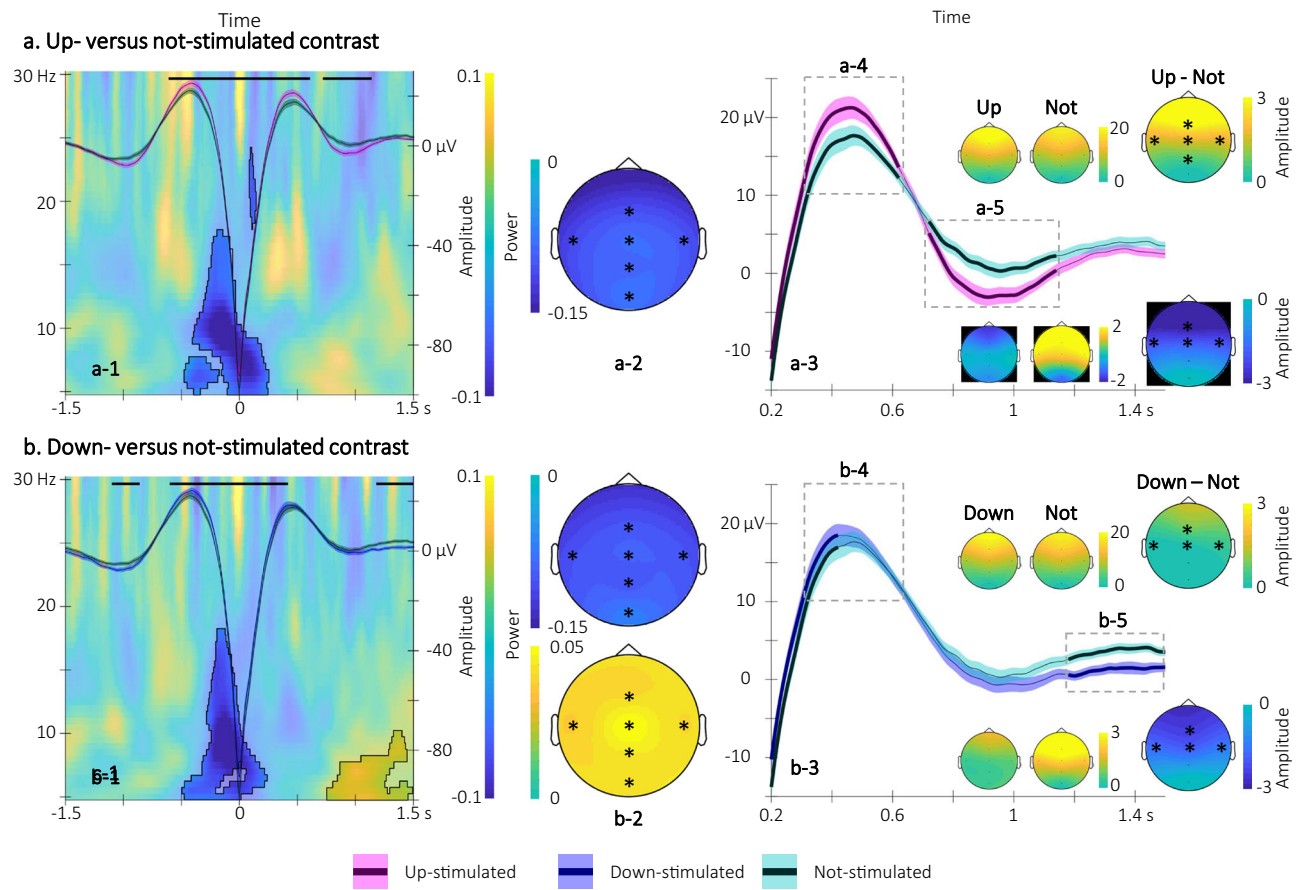

**Fig. 4 | Electrophysiological results of stimulated vs. not-simulated contrasts (n = 30).** Statistical comparisons were performed using non-parametric cluster-based permutations. **a** Up- vs. not-stimulated contrast. a-1. Time-frequency representation (TFR) of the difference in power modulation illustrated at Cz around the trough of the up- and not-stimulated slow oscillations (SOs) on which the grand-average of the SOs illustrated on Fz is super-imposed (magenta(green): up(not)-stimulated SO). Black lines represent the adjacent time points of the significant spatio-temporal clusters showing a difference in SO amplitude between the two trough-locked potentials (see Fig. S2b in Supplementary information for channel level analysis). The amplitude of the up-stimulated SOs was greater than the not-stimulated SOs from −0.61 to 0.61 s (cluster p = 0.0020, Cohen's d = 1.12) while it reversed from 0.72 to 1.14 s relative to the trough (cluster p = 0.0060, Cohen's d = −0.70). Power in the 5–17.5 Hz frequency range was lower in the up- as compared to the not-stimulated condition in the descending phase of the SOs (cluster p = 0.002, Cohen's d = −1.30, −0.45 to 0.24 s relative to SO trough). a-2. Topography of the significant cluster (between 7–12 Hz and −0.15–0 s time-frequency range).

a-3. Zoom on the differences in SW amplitude. a-4 and a-5. Topography of the differences in SO amplitude in the peak and deflection time-windows highlighted in the up- vs. down-stimulated contrast. **b** Down- vs. not-stimulated contrast. b-1. Same as a-1 for the down- and the not-stimulated trough-locked potentials (blue and green, respectively) and power modulation. The amplitude of the down-stimulated SOs was greater than the not-stimulated SOs from −0.60 to 0.42 s (cluster p = 0.0020, Cohen's d = 1.20) while it reversed from 1.18 to 1.50 s relative to the trough (cluster p = 0.0040, Cohen's d = −0.57). Power was lower in the 5–18 Hz frequency range in the down- as compared to the not-stimulated condition during the descending phase of SOs (cluster p = 0.002, Cohen's d = −0.94, −0.49 to 0.24 s relative to the SO trough) and greater in the 5–10 Hz from 0.76 to 1.50 s (cluster p = 0.0020, Cohen's d = 0.90). b-2. Topography of the negative (upper panel, 7–12 Hz and −0.15–0.08 s time-frequency range) and positive (lower panel, 5–8 Hz and 0.8–1.25 s time-frequency range) clusters. b-3-5. Same as a-3-5 for the down- vs. not-stimulated contrast. SO shaded areas represent ± standard error; (*) represents the electrodes included in the significant clusters.

## Phase-specific modulations of task-related hippocampal and striatal activity are related to the effect of TMR on motor performance

Brain imaging data were acquired during task practice before and after the night of stimulation (see Table S1 and Fig. S4 for brain activity elicited by task practice during initial learning). We first examined whether task-related brain activity increased from the pre-night to the post-night practice sessions within each condition. As expected, the results showed, for all conditions, a strong overnight increase in task-related brain activity in a set of striato-cortical regions including the putamen and the primary motor cortex (Fig. 5a and see Table S2-1 for a complete list of activations). Interestingly, the overnight increase in striatal activity was greater for the up-reactivated sequence as compared to the down-reactivated sequence, which in turn was greater than for the not-reactivated sequence (Fig. 5b; see Table S2-2 of the Supplementary information for details).

Importantly, the between-session increase in striato-motor activity reported above was correlated with the TMR index (i.e., the difference in offline changes in performance speed between the reactivated vs. the not-reactivated sequences) for both the up- and down-reactivated sequences (Fig. 6a; see Table S2-3). We also performed exploratory analyses to probe the link between the sleep EEG features showing the phase-specific modulation described above (i.e., SO amplitude and sigma power at the peak of the SO) and the between-session changes in brain activity. These analyses did not reveal any correlation between brain activity and sigma power but they showed that the overnight increase in activity in the basal-ganglia and the motor cortex was related to greater SO peak amplitude in the up and down conditions (Fig. 6b; see Table S2-4 of the Supplementary information). Altogether, the regression analyses indicate that greater overnight increase in striato-motor activity is related to both greater SO amplitude during the post-learning night and greater overnight

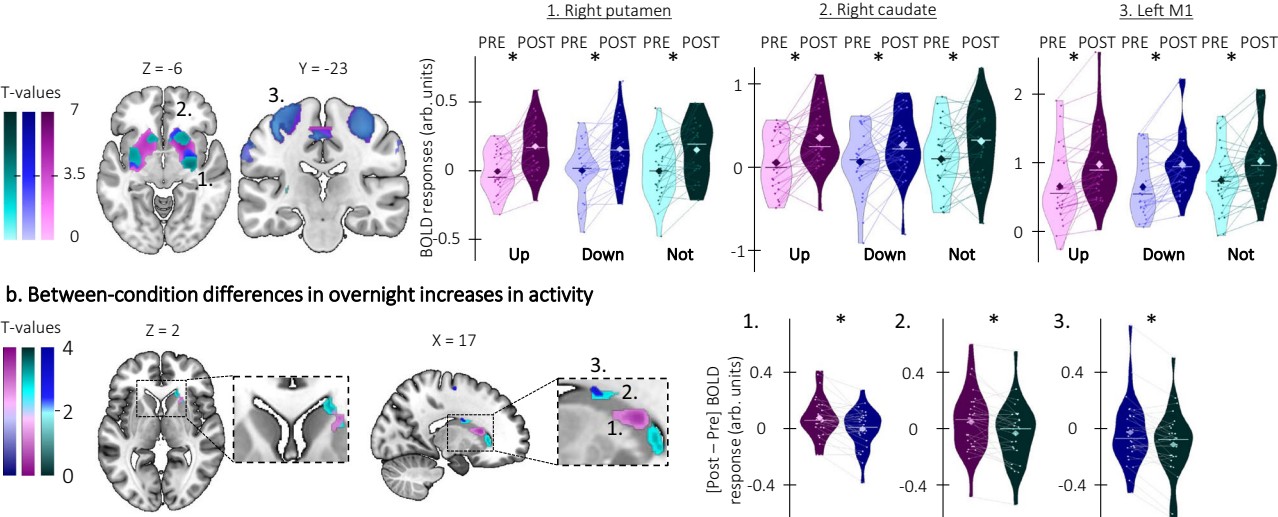

Fig. 5 | **Phase-specific modulation of task-related cortico-striatal activity** (**n** = 28). Statistical parametric maps were generated using a general linear model in SPM12 (see "Methods"). **a** Overnight increase in activity within condition. Brain activity increased overnight in a set of cortico-striatal regions for up- (right putamen: x = 24, y = 18, z = −10, Z = 5.33, $p_{SVC} < 0.001$; right caudate: x = 10, y = 4, z = 12, Z = 4.11, $p_{SVC} < 0.001$; left M1: x = −38, y = −20, z = 52, Z = 4.35, $p_{SVC} < 0.001$), down- (right putamen: x = 26, y = −8, z = −4, Z = 4.03, $p_{SVC} = 0.001$; right caudate: x = 10, y = 4, z = 16, Z = 3.29, $p_{SVC} = 0.004$; left M1: x = −38, y = −24, z = 54, Z = 4.52, $p_{SVC} < 0.001$), and not-reactivated sequences (right putamen: x = 28, y = −12, z = −6, Z = 3.88, $p_{SVC} = 0.002$; right caudate: x = 10, y = 6, z = 12, Z = 3.35, $p_{SVC} < 0.001$; left M1: x = −38, y = −26, z = 56, Z = 3.71, $p_{SVC} = 0.004$). Violin plots represent BOLD responses extracted from clusters overlapping between conditions (1. Right putamen: x = 26, y = −8, z = −4; 2. Right caudate: x = 10, y = 0, z = 14; 3. Left M1: x = −38, y = −20, z = 52). **b** Overnight increase in activity between conditions. The overnight increase in striatal (right caudate) activity was greater for the up- as compared to the down-reactivated sequence, which in turn was greater than for the not-reactivated sequence. Violin plots represent the difference in BOLD responses extracted from the activation peaks in the post- versus pre-night sessions (1. up vs. down: x = 20, y = 18, z = 12, Z = 3.41, $p_{SVC} = 0.009$; 2. up vs. not: x = 18, y = 28, z = 4, Z = 3.12, $p_{SVC} = 0.02$; 3. down vs. not: x = 16, y = −2, z = 26, Z = 3.22, $p_{SVC} = 0.016$). *: significant after small volume correction (SVC). Activation maps are displayed on a T1-weighted template image as part of the MRIcroGL software (https://www.mccauslandcenter.sc.edu/mricrogl) with an uncorrected threshold of $p < 0.005$. arb.: arbitrary. Violin plots: median (horizontal bar), mean (diamond), the shape of the violin plots depicts the kernel density estimate of the data. M1: primary motor cortex.

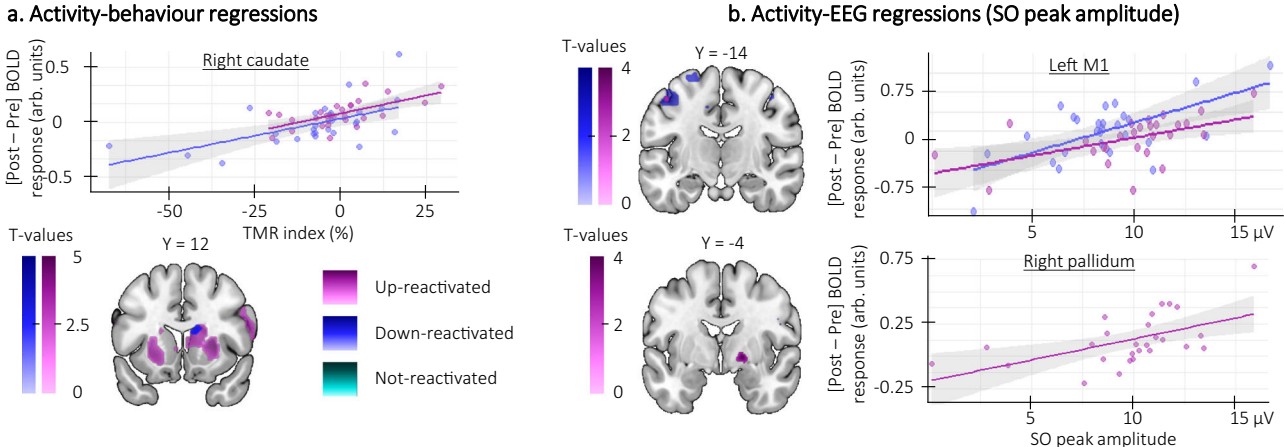

Fig. 6 | **Regression analyses (n = 28 unless specified otherwise).** Statistical parametric maps were generated using a general linear model in SPM12 (see "Methods"). **a** Brain activity-behavior regressions. The overnight increase in striato-motor activity was positively related to the Targeted Memory Reactivation (TMR) index for both the up- (right caudate: x = 20, y = 18, z = 12, Z = 3.67, $p_{SVC} = 0.005$) and down-reactivated sequences (right caudate: x = 16, y = −2, z = 26, Z = 3.07, $p_{SVC} = 0.024$) such that the greater the increase in brain activity (i.e., the more positive value on the y-axis), the greater the performance improvement on the reactivated as compared to the not-reactivated sequence (i.e., the more positive TMR index on the x-axis). Note that similar results were observed in the regression analyses using the contrasts testing for overnight changes in activity between conditions (i.e., reactivated (both up and down) vs. not-reactivated, see Supplementary Fig. S5). **b** Brain activity-EEG regressions ($n = 27$). The overnight increase in activity in the motor cortex (top panel) and the basal ganglia (bottom panel) was related to the slow oscillation (SO) peak amplitude in the up- (left M1: x = −46, y = −16, z = 48, Z = 2.72, $p_{SVC} = 0.055$; right pallidum: x = 20, y = −2, z = −6, Z = 3.49, $p_{SVC} = 0.045$) and down-stimulated (left M1: x = −44, y = −16, z = 52, Z = 3.33, $p_{SVC} = 0.012$) conditions such that the greater the SO peak amplitude during the night (x-axis), the greater the overnight increase in activity in these regions (y-axis). Activation maps are displayed on a T1-weighted template image as part of the MRIcroGL software (https://www.mccauslandcenter.sc.edu/mricrogl) with a threshold of $p < 0.005$ uncorrected. arb.: arbitrary. M1: primary motor cortex. Shaded area around the linear regression lines represents the 95% confidence interval.

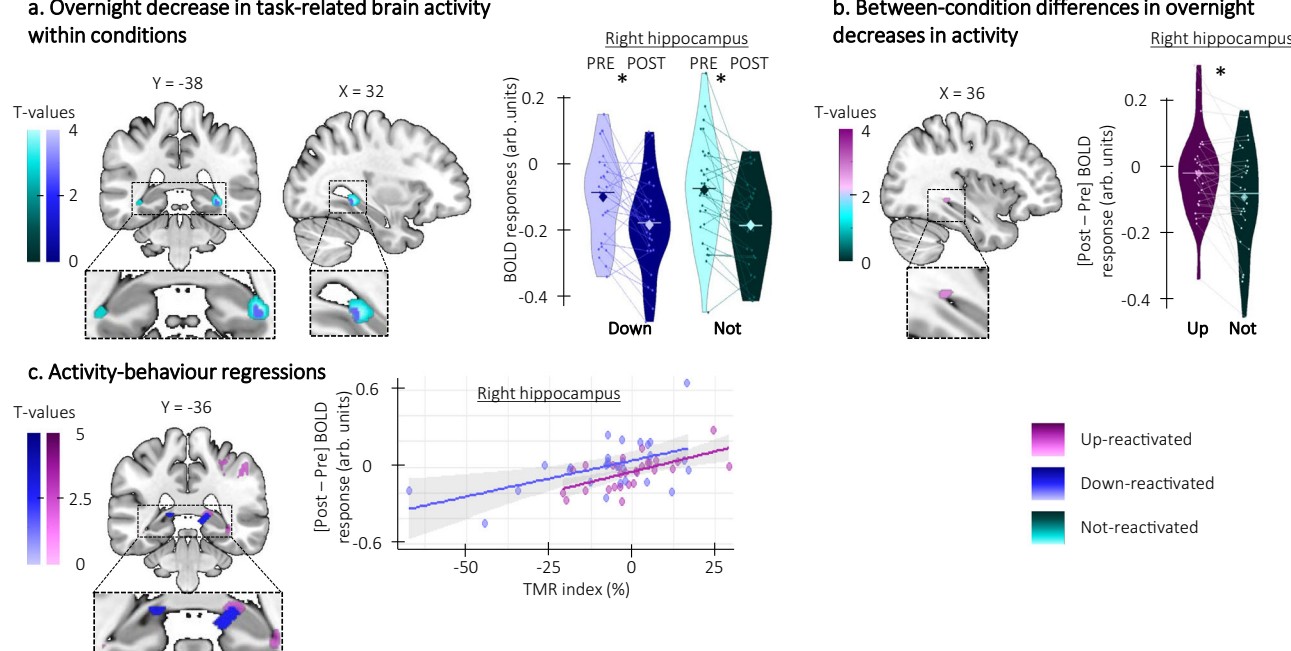

**Fig. 7 | Phase-specific modulation of task-related hippocampal activity ($n$ = 28).**
Statistical parametric maps were generated using a general linear model in SPM12 (see "Methods"). **a** Overnight decrease in activity within condition. Brain activity decreased overnight in the hippocampus for both down- (right hippocampus: x = 32, y = −38, z = −6, Z = 2.80, $p_{SVC}$ = 0.044) and not-stimulated sequences (right hippocampus: x = 34, y = −38, z = −6, Z = 3.67, $p_{SVC}$ = 0.005). Violin plots represent BOLD responses extracted from clusters overlapping between conditions (x = 32, y = −38, z = −6). **b** Overnight decrease in activity between conditions. The overnight decrease in hippocampal activity was greater for the not-reactivated sequence as compared to the up-reactivated sequence (right hippocampus: x = 36, y = −36, z = −4, Z = 2.98, $p_{SVC}$ = 0.028). **c** Brain activity-behavior regressions. The overnight increase in hippocampal activity was positively related to the Targeted Memory Reactivation (TMR) index for both up- (right hippocampus: x = 36, y = −38, z = −8,

Z = 3.64, $p_{SVC}$ = 0.005) and down-reactivated sequences (right hippocampus: x = 20, y = −34, z = 4, Z = 2.84, $p_{SVC}$ = 0.041) such that the lower the increase in brain activity (y-axis), the lower the performance improvement on the reactivated as compared to the not-reactivated sequence (x-axis). Note that similar results were observed in the regression analyses using the contrasts testing for overnight changes in activity between conditions (i.e., reactivated (both up and down) vs. not-reactivated, see Supplementary Fig. S5). *: significant after small volume correction (SVC). Activation maps are displayed on a T1-weighted template image as part of the MRIcroGL software (https://www.mccauslandcenter.sc.edu/mricrogl) with a threshold of $p < 0.005$ uncorrected. arb.: arbitrary. Violin plots: median (horizontal bar), mean (diamond), the shape of the violin plots depicts the kernel density estimate of the data. Shaded area around the linear regression lines represents the 95% confidence interval.

gains in motor performance in both up and down conditions. Interestingly, despite condition differences in overnight changes in brain activity (Fig. 5b), SO amplitude (Fig. 3) and motor performance speed (Fig. 2), the phase of the stimulation did not alter the relationship between these differences.

Next, we examined whether task-related brain activity decreased between the pre- and post-night practice sessions. Results showed that hippocampal activity decreased overnight for both the down- and the not-reactivated sequences while no significant changes were observed in the up condition (Fig. 7a; see Table S2-1 of the Supplementary information). The decrease in hippocampal activity observed for the not-reactivated sequence was greater than for the up-reactivated sequence (Fig. 7b; see Table S2-2 of the Supplementary information). Importantly, the between-session changes in hippocampal activity reported above were correlated with the TMR index for both the up- and down-reactivated sequences such that greater overnight decrease in activity was related to poorer performance speed (Fig. 7c; see Table S2-3 of the Supplementary information). Finally, we did not observe any relationships between EEG features and changes in hippocampal activity. Overall, these results suggest that up-stimulation prevented the overnight decrease in hippocampal activity observed in the other conditions, the amplitude of which is related to poorer performance speed.

Altogether, these results show that the amplitude of the changes in brain activity occurring in striato-hippocampo-motor areas as a result of the consolidation process were modulated by the phase of the

SO during which TMR was applied. Importantly, the magnitude of these changes was related to SO characteristics and changes in motor performance, both metrics that also showed a phase-specific modulation of amplitude.

## Phase-specific modulations of connectivity in striato-hippocampo-motor networks are related to the effect of TMR on motor performance

We examined whether stimulation modulated task-related connectivity patterns in the brain regions showing phase-specific modulation of activity described above (i.e., the hippocampus and the striatum, see methods and Table S3).

We observed an overnight decrease in hippocampo-motor connectivity for the up-reactivated sequence which was greater than for the not-reactivated sequence (Fig. 8a, b, see Tables S3-1.1.3 and -2.2.3 of the Supplementary information). Moreover, an overnight decrease in striato-cortical (Fig. S6a, right panel and Table S3-3.1.2 of the Supplementary information) and striato-hippocampal connectivity was related to a greater TMR index for the up-reactivated sequence (Fig. 8c, see Table S3-3.1.3 of the Supplementary information). This suggests that the beneficial effect of up-stimulation on performance speed was related to more segregation of task-relevant brain regions within their functional network, i.e., by a decrease in connectivity between these brain areas (which was also paralleled by an overall increase in activity within these brain regions, see above).

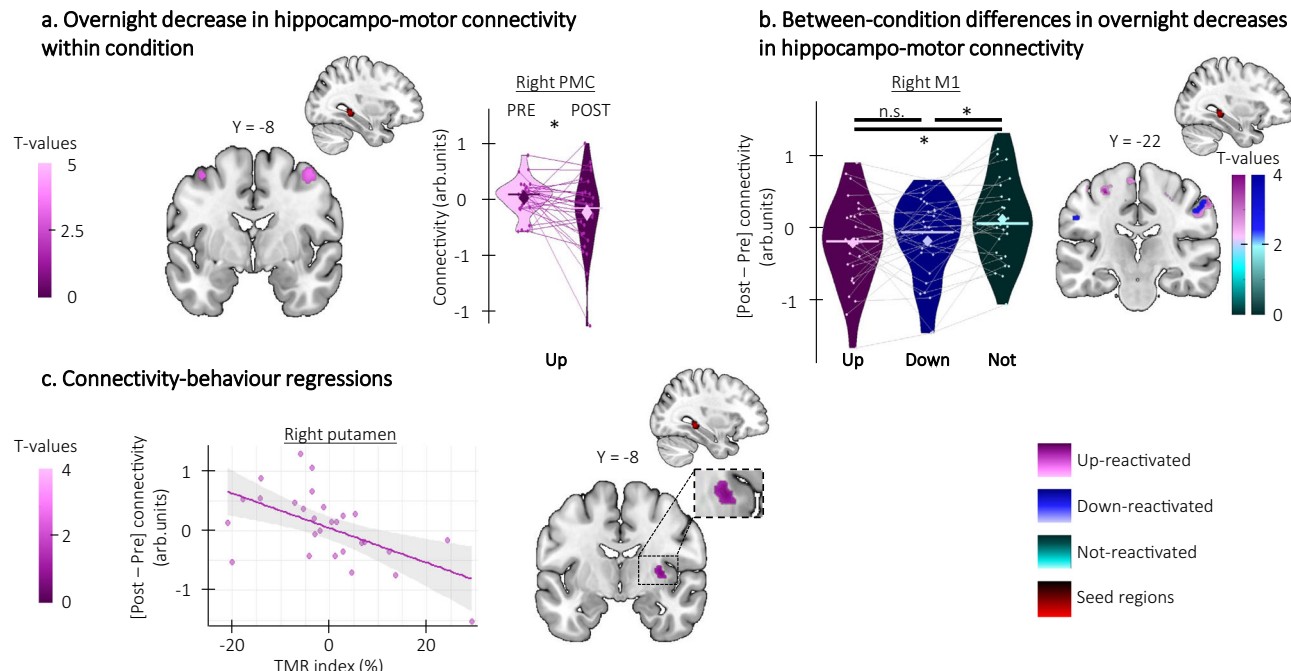

**Fig. 8 | Phase-specific modulation of task-related striato-hippocampo-cortical connectivity ($n = 28$).** Statistical parametric maps were generated using a general linear model in SPM12 (see "Methods"). **a** Overnight decrease in hippocampo-motor connectivity within condition. Hippocampo-motor connectivity decreased from pre- to post-night sessions for the up-reactivated condition (hippocampus-right PMC: x = 46, y = −8, z = 54, Z = 3.08, $p_{SVC}$ = 0.035). **b** Overnight decrease in hippocampo-motor connectivity between conditions. The overnight decrease in hippocampo-motor connectivity was greater in the up- (hippocampus-right M1: x = 58, y = −22, z = 46, Z = 3.19, $p_{SVC}$ = 0.033) and the down-reactivated sequences (hippocampus-right M1: x = 54, y = −22, z = 46, Z = 2.85, $p_{SVC}$ = 0.046) as compared to the not-reactivated sequence. Violin plots represent BOLD responses around a common significant voxel (x = 52, y = −24, z = 40). **c** Brain connectivity-behavior regressions. The overnight decrease in striato-hippocampal connectivity (hippo-campus-right putamen: x = 32, y = −8, z = 4, Z = 3.10, $p_{SVC}$ = 0.027) was negatively

correlated with the Targeted Memory Reactivation (TMR) index such that the greater the decrease in connectivity (y-axis), the greater the performance improvement on the up-reactivated as compared to the not-reactivated sequence (x-axis). Note that similar results were observed in the regression analyses using the contrasts testing for overnight changes in activity between conditions (i.e., reactivated (both up and down) vs. not-reactivated, see Supplementary Fig. S5). M1: primary motor cortex, PMC: premotor cortex; *: significant after small volume correction (SVC). Activation maps are displayed on a T1-weighted template image as part of the MRIcroGL software (https://www.mccauslandcenter.sc.edu/mricrogl) with a threshold of $p < 0.005$ uncorrected. arb.: arbitrary. Violin plots: median (horizontal bar), mean (diamond), the shape of the violin plots depicts the kernel density estimate of the data. Shaded area around the linear regression lines represents the 95% confidence interval.

In the down-reactivated condition, there was an overnight increase in striato-motor connectivity (Fig. 9a; see Table S3-1.2.2 of the Supplementary information) that was greater than for the up-reactivated sequence (Fig. 9b; see Table S3-2.1.2 of the Supplementary information). Interestingly, we observed an overall negative relationship between overnight increases in connectivity in hippocampo-striato-motor networks and sleep features such that lower SO amplitude and sigma power were related to greater overnight increases in connectivity (striato-hippocampal connectivity-sigma power: see Fig. 9c, left panel and Tables S3-4.2.1 and S3-4.2.2; striato-motor connectivity-SO amplitude: see Fig. 9c, right panel; and Tables S3-4.4.1 and 4.4.2; hippo-motor connectivity-SO amplitude: see Fig. S6b and Table S3-4.4.3; and striato-hippocampal connectivity-SO amplitude: see Fig. S6c; and Tables S3-4.4.1 and 4.4.2). These results suggest that the reduced amplitude of sleep features observed under down- (as compared to up-) stimulation was presumably related to compensatory overnight increases in connectivity in hippocampo-striato-motor networks. Importantly, these overnight increases in connectivity were differently related to behavior depending on the networks examined. Specifically, the overnight increase in striato-motor connectivity was related to poorer TMR index (Fig. 9d, left panel; see Table S3-3.2.2 of the Supplementary information) while the increase in striato-hippocampal connectivity was related to greater TMR index (Fig. 9d, right panel, see Table S3-3.2.1 of the Supplementary information). These findings suggest that increases in connectivity in striato-motor networks were ineffective to compensate for the negative effect of

down-stimulation on performance speed while increases in connectivity between the striatum and the hippocampus were related to greater performance improvement.

In sum, the connectivity results indicate an overall decrease in hippocampal and striatal connectivity after up-stimulation that was related to better performance. In contrast, down-stimulation resulted in an overall increase in connectivity in hippocampo-striato-motor networks that was related to the lower amplitude of sleep features during down-stimulation. Interestingly, these overnight increases in connectivity were differently related to performance improvement suggesting that different networks may play distinct roles to compensate for the reduced plasticity induced by down-stimulation during sleep.

## Discussion

The goal of this pre-registered study was to combine Targeted Memory Reactivation (TMR) and closed-loop (CL) stimulation approaches to (1) test whether reactivating motor memories at the up- (as compared to the down-) phase of slow oscillations (SOs) during post-learning sleep enhances motor memory consolidation; and to (2) provide a comprehensive characterization of the underlying neuro-physiological processes using sleep EEG and task-related fMRI. The results corresponding to the different modalities (performance, EEG and MRI) are summarized in Fig. 10. As hypothesized, overnight changes in performance speed were greater for motor sequences reactivated at the up-, as compared to the down-phase of the SO.

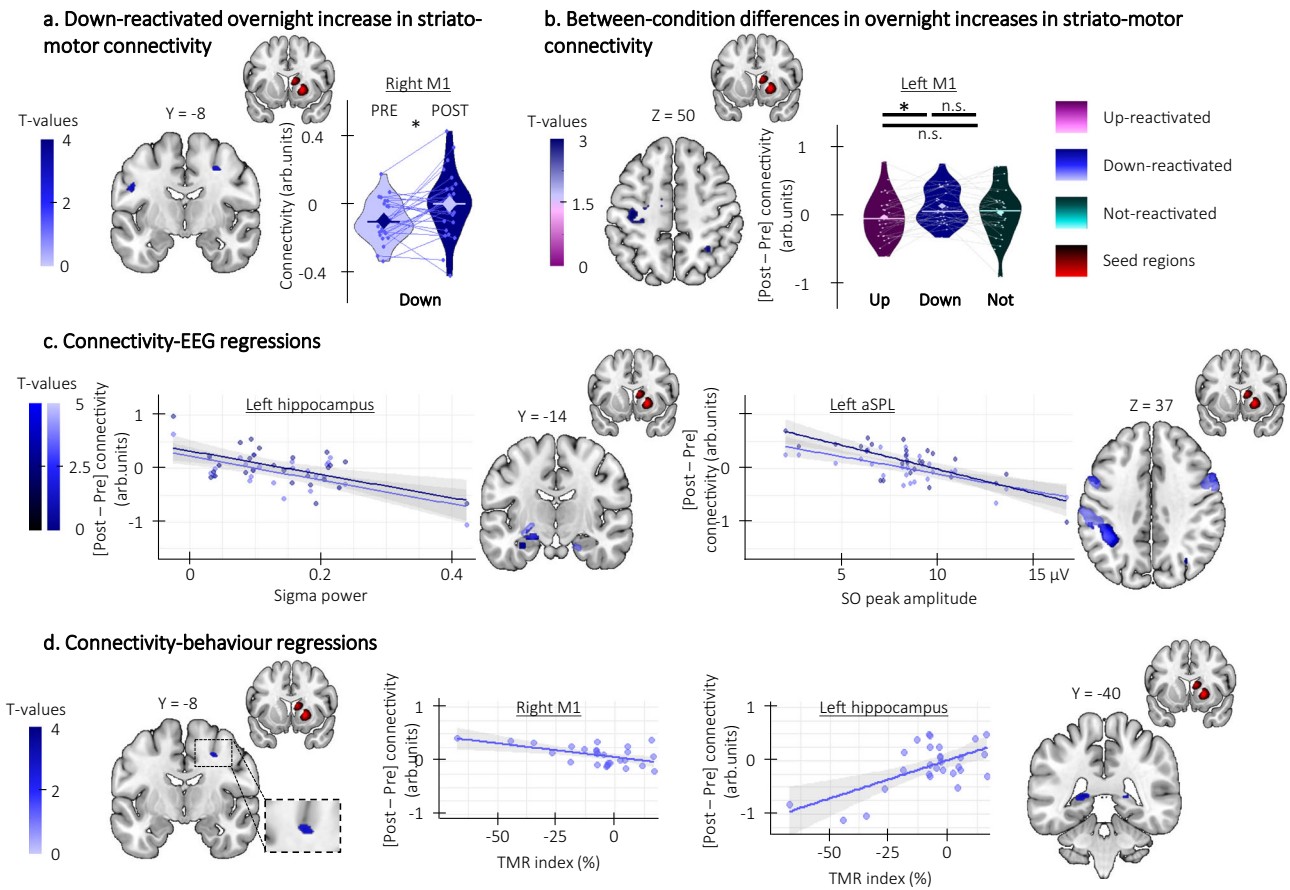

**Fig. 9 | Down-stimulation modulation of striato-hippocampo-cortical connectivity (n = 28 unless specified otherwise).** Statistical parametric maps were generated using a general linear model in SPM12 (see "Methods"). **a** Overnight increase in striato-motor connectivity. Striato-motor connectivity increased from pre- to post-night session for the down-reactivated condition (Putamen-right M1: x = 28, y = −8, z = 44, Z = 2.92, $p_{SVC}$ = 0.038). **b** Overnight increase in striato-motor connectivity between conditions. The overnight increase in striato-motor connectivity was greater in the down- as compared to the up-reactivated sequence (Putamen-left M1: x = −32, y = −20, z = 50, Z = 3.15, $p_{SVC}$ = 0.02). **c** Brain connectivity-EEG regressions (n = 27). The overnight decrease in striato-hippocampal (left panel, Caudate-left hippocampus (pale blue): x = −24, y = −14, z = −8, Z = 3.68, $p_{SVC}$ = 0.005; Putamen-left hippocampus (dark blue): x = −18, y = −10, z = −8, Z = 3.25, $p_{SVC}$ = 0.017) and striato-motor (right panel, Caudate-left aSPL (pale blue): x = −44, y = −44, z = 38, Z = 3.84, $p_{SVC}$ = 0.003; Putamen-left aSPL (dark blue): x = −38, y = −42, z = 36, Z = 4.56, $p_{SVC}$ < 0.001) connectivity were correlated with the sigma power and slow oscillation (SO) peak amplitude respectively such that the lower the SO peak amplitude and sigma power during the night (y-axis), the greater the overnight increase in connectivity (x-axis). **d** Brain connectivity-behavior regressions. The overnight increase in striato-motor connectivity was related to the

Targeted Memory Reactivation (TMR) index (left panel, Putamen-right M1: x = 26, y = −8, z = 44, Z = 3.09, $p_{SVC}$ = 0.025) such that the greater the increase in brain connectivity (y-axis), the lower the performance improvement on the down-reactivated as compared to the not-reactivated sequence (x-axis). In contrast, the overnight increase in striato-hippocampal connectivity was positively related to the TMR index (right panel, caudate-left hippocampus: x = −16, y = −40, z = 6, Z = 3.30, $p_{SVC}$ = 0.014) such that the greater the increase in brain connectivity (y-axis), the greater the performance improvement on the down-reactivated as compared to the not-reactivated sequence (x-axis). Note that similar results were observed in the regression analyses using the contrasts testing for overnight changes in activity between conditions (i.e., reactivated (both up and down) vs. not-reactivated, see Supplementary Fig. S5). M1: primary motor cortex, aSPL: anterior superior parietal lobule; *: significant after small volume correction (SVC). Activation maps are displayed on a T1-weighted template image as part of the MRIcroGL software (https://www.mccauslandcenter.sc.edu/mricrogl) with a threshold of p < 0.005 uncorrected. arb.: arbitrary. Violin plots: median (horizontal bar), mean (diamond), the shape of the violin plots depicts the kernel density estimate of the data. Shaded area around the linear regression lines represents the 95% confidence interval.

Unexpectedly though, only performance speed on the down-reactivated sequence differ from the not-reactivated one. Electrophysiological data showed that up-stimulated SOs were of higher amplitude and presented greater peak-nested sigma power (spindle frequency band) than down-stimulated SOs. Brain imaging data collected during task practice also indicated that the practice of up-, as compared to both down- and not-reactivated sequences, resulted in greater activity in striato-motor areas, greater maintenance in hippocampal activity, and decreased connectivity in these networks. Importantly, these modulations in brain responses were related to the up-TMR-induced increase in SO amplitude and improvement in performance. In contrast, down-stimulation resulted in a lower increase in striato-motor activity that was paralleled by significant increases in connectivity in striato-hippocampo-motor networks, and both were

related to the lower amplitude of sleep features during down-stimulation. Interestingly, the overnight increases in connectivity observed after down-stimulation were related to better (striato-hippocampus) or worse (striato-motor) performance speed on the down-reactivated sequence.

Our behavioral results indicate that TMR applied at the up-phase of the SO resulted in greater gains in motor performance than when administered at the down-phase of the SO. These phase-specific effects are in line with previous studies showing that acoustic stimulations delivered in a closed-loop fashion at the up-phase (or during the down-to-up transition) of the SO result in greater declarative memory consolidation[17,20,21,24–26] (but see refs. 25,27 for null effects). We are only aware of one study using closed-loop acoustic stimulation in the motor memory domain and results did not reveal any beneficial effect of SO

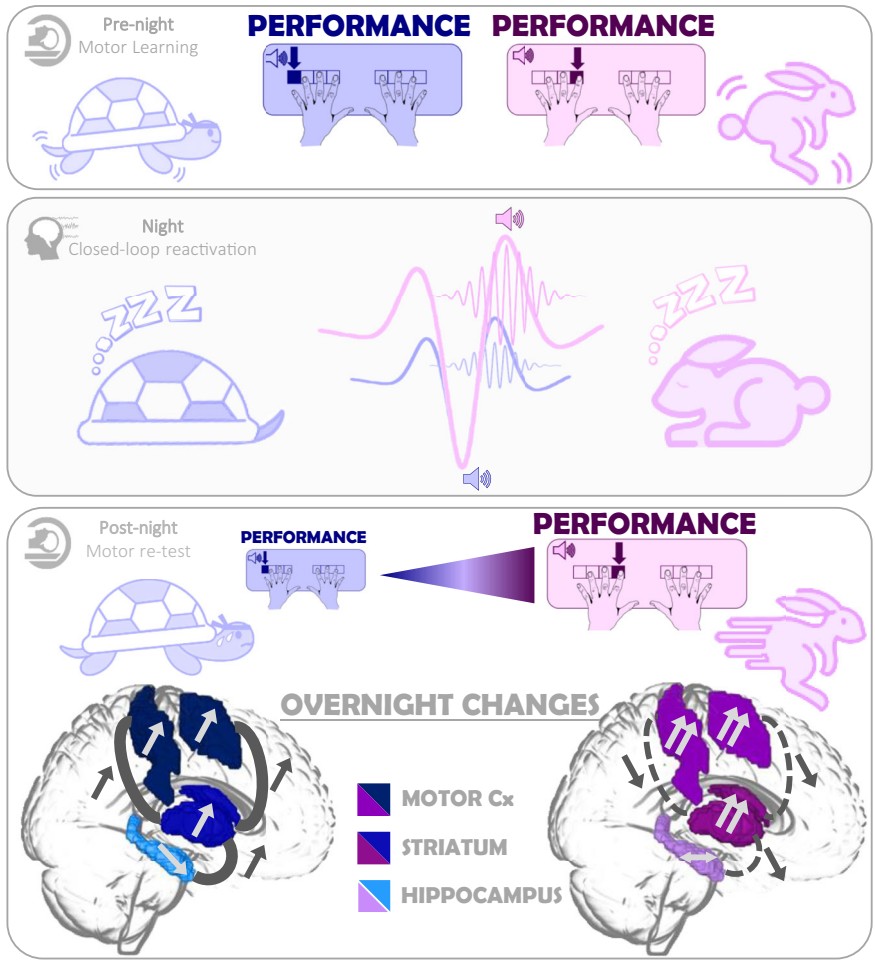

**Fig. 10 | Summary of the main findings.** Slow-oscillation (SO) closed-loop targeted memory reactivation (CL-TMR) was applied during post-learning sleep so that two sounds corresponding to two different memory traces initially encoded during the pre-night training session (i.e., the rabbit motor sequence and the turtle motor sequence in the Pre-night panel) were replayed at the up- (magenta shades) or the down- (blue shade) phases of the SO. During post-learning sleep, CL-TMR at the SO-up-phase – as compared to the SO-down-phase – enhanced SO amplitude and resulted in a greater burst of sigma oscillations at the peak of the SO (Night panel). Motor performance collected during the post-night retest (Post-night panel) indicated that up-state cueing resulted in greater performance improvement than down-state cueing (i.e., the rabbit motor sequence was executed faster than the turtle motor sequence). At the brain level, up stimulation – as compared to down stimulation – resulted in a greater overnight increase in activity in striato-motor regions and preserved hippocampal activity (see arrows within the brain regions of interest) as well as an overall decrease in hippocampo-striato-motor connectivity (represented by the dashed lines between brain regions). These up-state cueing strengthening in activity in – and segregation of – striato-motor and hippocampal networks was related to the beneficial effect of TMR on sleep features and performance. Down-stimulation (as compared to up-stimulation) resulted in reduced overnight changes in brain activity that were accompanied by increased connectivity in hippocampo-striato-motor networks. These overnight increases in connectivity were differently related to performance improvement suggesting that different networks may play distinct roles to compensate for the reduced plasticity induced by down-stimulation during sleep (see "Results" for details). Brain images are template images as part of the MRIcroGL software (https://www.mccauslandcenter.sc.edu/mricrogl). Turtle and rabbit adapted from La Fontaine. Cx: Cortex.

up-stimulation on motor performance[28]. The discrepancy between this recent research and our findings is unclear, but it is possible that methodological differences between studies, such as time afforded in NREM sleep (nap vs. night paradigm) or stimulation phase (380 ms post-trough vs. peak), might have contributed to these inconsistencies. Another notable difference is that the sounds used in the current research were memory cues while the earlier work cited above used acoustic stimuli that were not linked to any learned material (i.e., "clicks"). In line with the current findings, the few studies using SO-closed-loop stimulation with memory cues (i.e., CL-TMR) have shown an up-phase stimulation benefit on declarative memory consolidation as compared to no-reactivation[18] or down-stimulation[19]. Overall, our results thus concord with this earlier research and suggest that reactivating motor memories at the up-, as compared to the down-phase, of slow oscillations during post-learning sleep benefits motor memory consolidation. Interestingly, our results also indicate that down-

reactivated sequences presented significantly worse performance as compared to both up- and not-reactivated sequences. Together with our results showing no significant difference in performance between the up- and not-reactivated sequences, one could therefore argue that a more plausible interpretation of the data is that down-stimulation actively disrupted the motor memory consolidation process. While possible, this interpretation contradicts earlier studies showing no specific effect of down-, as compared to no-stimulation, on memory consolidation[17–19]. Based on these earlier studies and evidence that down-phases of SOs are silent phases of neuronal inactivity[29], we speculate that our results reflect the nature of our paradigm rather than an active disruptive effect of down-stimulation on memory consolidation. Specifically, it is possible that, due to our within-subject design, overall acoustic stimulation during post-learning sleep might also have boosted performance on the control (not-reactivated) condition. This could have accentuated the difference in performance

between down- and not-reactivated conditions and could also explain the lack of difference between up- and not-reactivated conditions in the current study. While this is hypothetical, it is in line with previous work using similar within-subject design showing no difference in performance between the up- and not-reactivated conditions[19].

Our electrophysiological data show phase-specific modulations of sleep oscillations such that up-stimulated SOs exhibited higher amplitude and presented greater sigma power nested at their peak as compared to down-stimulated SOs. In contrast, the different stimulation conditions did not modulate spindle event characteristics. Slow oscillations, sleep spindles (events and sigma oscillations taken as a proxy of spindle activity) and sharp wave ripples are part of the three cardinal NREM sleep oscillations playing a critical role in memory consolidation during sleep[1,6,30]. SO up-states have received particular attention as they are known to host heightened excitability[29]. It is thought that prolonging SO up-phase with stimulation increases the probability of neuronal ensembles to fire together and strengthen the memory traces encoded in these networks[31]. Accordingly, prior research has used experimental interventions to target plasticity processes during this window and, in turn, influence memory consolidation during sleep. In line with our findings, these earlier studies have collectively shown that stimulating SO at the up-phase of the slow-oscillation enhanced the amplitude of ongoing SOs[17-21,24-26,28,32], sigma oscillation power during the ascending phase of the SO[18-20] and memory consolidation[17-21,25,26]. It is worth noting though that we did not observe up-stimulation-induced modulations of spindle event characteristics. However, these null results do not rule out the involvement of spindle activity in TMR-related motor memory consolidation processes. Recent evidence has brought forward the idea that spindle event detection in general is less sensitive than the study of the sigma rhythm as a whole[14,33]. In line with these observations, our results show that sigma oscillation properties – as opposed to spindle events – were modulated by the phase of the stimulation. Interestingly, our results also indicate that these up-stimulation-induced modulations in sigma power emerged prior to auditory cue onset (i.e., prior to SO up-state). This result might be related to our repetitive stimulation paradigm which potentially allowed for the stimulation of consecutive SOs. Based on earlier work showing oscillatory entrainment by rhythmic acoustic stimulation[34,35], we suggest that our stimulation design particularly favored the entrainment of ongoing sleep oscillations resulting, with repetition, in power modulation prior to cue onset. Specifically, we propose that the phase-specific modulation of induced responses observed prior to cue onset reflects the effect of CL-TMR on ongoing top-down[36-38] (here cortico-hippocampal[39]) processes. Further, we propose that the reinforcement of ongoing SO-sigma coupling favored spindle-mediated top-down mechanisms supporting memory reactivation and in turn enhanced memory performance[39]. Altogether, our findings generally concord with earlier experimental work and with models suggesting that the synchronous neural firing orchestrated by the SOs leads to a higher probability of downstream synchronous neural firing favoring the occurrence of higher frequency oscillation bursts such as sigma oscillations that are beneficial for memory consolidation[40].

Our brain imaging data indicate that up-stimulation resulted in an overnight increase in task-related striato-motor activity and a maintenance of hippocampal activity as compared to the down- and no-stimulation conditions. The involvement of striato- and hippocampo-cortical networks in motor sequence learning and memory consolidation is well documented. Specifically, task-related striato-motor activity generally increases with consolidation and during later stages of learning e.g.,[22,41]. Hippocampal activity during both learning and delayed retests has also been associated to successful (sleep-related) motor memory consolidation[23,42-45]. The present data therefore suggest that up-stimulation further strengthened the modulation of brain activity that is usually observed as a result of the spontaneous motor memory consolidation process. Interestingly, our results also show that the modulations of hippocampal and striato-motor activity reported above were related to both the overnight performance improvement and the enhancement of sleep features related to up-stimulation. These findings generally concord with a series of earlier correlational studies. First, they are in line with an acoustic stimulation study showing that up-phase stimulation-induced increases in SO amplitude were correlated with greater hippocampal activity during subsequent (post-sleep) declarative learning[32]. They also concur with earlier studies in the motor memory domain showing a positive relationship between spindle-locked striatal and hippocampal reactivations during post-learning sleep and offline gains in motor performance[46]. They are also consistent with the findings of a previous TMR study showing that the increased striatal and hippocampal activity observed for reactivated (as compared to not-reactivated) motor sequences was related to the time spent in NREM sleep (during which TMR was applied)[12]. Based on the evidence reviewed above and the current data, we suggest that the up-stimulation-induced increase in SO amplitude and sigma power during post-learning sleep facilitated hippocampal and striatal reactivations which in turn resulted in greater activity in these networks during retest and better motor performance. It is worth noting that similar modulations of brain activity – albeit of lower amplitude – were observed for down-reactivated sequences. Interestingly, the relationship between the modulation in brain responses in the down condition and both performance and sleep features was similar as for the up condition but scaled down in terms of amplitude (i.e., lower changes in brain activity, lower performance gains and lower amplitude of sleep features). These results suggest that plasticity processes induced by down-stimulation during sleep were presumably reduced as compared to up-stimulation and therefore less favorable for reactivation processes to take place during post-learning sleep. This is in line with previous research suggesting that the efficacy of TMR follows a (down-to-up) gradient rather than an "all-or-none" principle[19]. Here, we speculate that such reduced plasticity resulted in lower activity in hippocampal and striatal areas during retest and poorer motor performance. Altogether, these brain imaging results further corroborate the hypothesis that down-stimulation did not actively disrupt motor memory consolidation, but rather failed to potentiate ongoing consolidation processes.

Our brain connectivity analyses revealed an overnight decrease in hippocampo-striato-cortical connectivity in the up-condition which was related to better performance. These observations stand in contrast with earlier reports of increased connectivity within striato-motor networks[47] and between the striatum and the hippocampus[23,48] as a result of spontaneous motor memory consolidation. They are also inconsistent with earlier TMR studies showing greater connectivity in these networks during the practice of reactivated as compared to not-reactivated motor sequences[12,49]. We speculate that the decrease in connectivity observed after up-stimulation in the current study reflects an advanced stage of consolidation in which brain regions are more segregated due to a decreased need for long-range integration as a result of consolidation[1,31,50,51]. Conversely, in the down condition, we observed an overnight increase in connectivity within these networks which was negatively correlated to sleep features such that the lower the SO amplitude and sigma oscillation power, the greater the connectivity increase. Together with the brain activation results showing lower modulation of brain activity after down-stimulation, it is tempting to speculate that the large increases in connectivity observed after down stimulation reflect compensatory processes. Importantly, the increases in connectivity observed for the down-reactivated sequences were differently related to changes in performance depending on the network involved. Specifically, the overnight increase in striato-hippocampal connectivity was associated to better performance while increases in striato-motor connectivity were related to worse performance. We speculate that connectivity within these

different networks reflects different processes. On the one hand, connectivity between the striatum and the hippocampus during initial motor learning has been proposed to reflect the integration of different task components (motor vs. spatial) that is necessary to optimize motor behavior[22,23]. On the other hand, increases in striato-motor connectivity occur at later stages of learning for the automatization of the motoric component of the task[41,52]. We argue that the sub-optimal plasticity process under down-stimulation slowed down consolidation such that greater interaction between the striatal and hippocampal systems - usually observed early during initial learning - was beneficial for performance at retest while greater connectivity in non-optimally consolidated motor networks failed to optimize behavior. This remains, however, speculative.

In summary, this study shows a phase-specific beneficial effect of slow-oscillation closed-loop TMR on consolidation such that motor memories reactivated during the up-phase of slow oscillations exhibited greater performance speed compared to memories reactivated at the down-phase. These beneficial consolidation effects were related to phase-specific modulations of activity and connectivity in task-relevant networks including the striatum, the hippocampus and the motor cortex which were also associated with phase-specific alterations of the characteristics of sleep EEG features involved in plasticity processes. Altogether, this study not only highlights the promise of CL-TMR to induce phase-specific modulations of motor performance but also sheds light on the complex interplay between sleep oscillations, task-related brain activity and connectivity patterns during motor memory consolidation.

## Methods

This study was pre-registered in the Open Science Framework (https://osf.io/). Our pre-registration document outlined our hypotheses and intended analysis plan as well as the statistical models used to test our a priori hypotheses (available at https://osf.io/dpu6z). Whenever an analysis presented in the current paper was not pre-registered, it is referred to as exploratory. Additionally, to increase transparency, any deviation from the pre-registration is marked in the methods section with a (#) symbol and listed in Table S4 of the Supplementary information together with a justification for the change.

### Participants

Young healthy volunteers (mean age: 23.7 yo ranging from 18 to 30) were recruited to participate in the proposed research project and received a monetary compensation for their time and effort. Self-reported sex was considered in the present study by (1) ensuring an equal sex ratio within the participants' sample (see below) and (2) by adapting the SO detection parameters according to the self-reported sex (see EEG data acquisition and closed-loop TMR section). Every participant gave written informed consent before participating in this research protocol, which was approved by the Medical Ethical Committee Research UZ / KU Leuven (B322201525025) and was conducted according to the declaration of Helsinki (2013). Inclusion criteria were: (1) no previous extensive training of dexterous finger movements via playing a musical instrument or as a professional typist, (2) free of medical, neurological, psychological, or psychiatric conditions, including depression[53] and anxiety[54], (3) no indications of self-reported abnormal sleep[55], (4) free of psychoactive and sleep-influencing medications, (5) eligible for MR measurements, and (6) right-handed[56]. None of the participants were shift-workers or did a trans-meridian trip in the month preceding the study.

We performed a power analysis based on our previous study investigating auditory TMR in an open-loop paradigm[14]. This power analysis was performed with the G*Power software[57]. The partial η² was calculated based on our behavioral main effect showing a significant effect of condition (reactivated vs. control) on offline changes in performance speed and was transformed to an effect size f (η² = 0.15;

f = 0.42). The correlation coefficient calculated between the offline changes in performance speed in the reactivated and the control conditions was 0.66, but due to the different nature of the design (e.g., 3 sequences instead of 2, 5-element sequences instead of 8), we set the average correlation coefficient between repeated measures at r = 0.5 for a more conservative power calculation. Finally, the sphericity correction was set to 0.5 since the primary factors of interest in our design had 3 levels (up-reactivated, down-reactivated and not-reactivated). The primary contrast of interest is a reactivation condition main effect on offline changes in performance speed tested with a one-way rmA-NOVA. The required sample size for a 95% power is 27 at an alpha error probability of 0.05. In total, 34 (17 self-declared females) participants completed the study. Two participants were excluded for experimental error (i.e., one participant because they were erroneously enrolled despite being left-handed and the other one because of technical errors with the scanner) and one for excessive movement in the scanner (see MRI section below). The remaining 31 participants were included in closed-loop stimulation analyses (i.e., SO detection accuracy and stimulation phase, N = 31), but only 27 participants presented a complete dataset (behavior, sleep EEG and MRI data). In three participants, only the sleep EEG data was analyzed as behavioral and MRI data of the post-night session was corrupted due to experimental error (sleep EEG analyses, N = 30). For one participant, only MRI and behavioral data were analyzed due to an EEG recording default (Behavioral and MRI analyses, N = 28). Also note that for five participants, the pre-night Psychomotor Vigilance Task data was overwritten due to experimental error whereas for one participant both the pre- and post-night data were overwritten. Participant's characteristics are reported in Table S5 in the Supplementary information.

### General design

In this full within-subject design presented in Fig. 1, participants were first invited, in the evening, for a habituation night during which they completed a full night of sleep monitored with polysomnography (PSG). Roughly a week later, participants returned to complete the experimental session. Between these two visits, each participant followed a constant sleep/wake schedule (according to their own rhythm +/- 1 h) during at least 3 days before the experimental session (compliance assessed with sleep diaries and wrist actigraphy, Acti-Graph wGT3X-BT, Pensacola, FL). Sleep quality and quantity for the night preceding the experiment was assessed with the St. Mary's sleep questionnaire[58] (sleep data are presented in Table S5).

During the experiment, volunteers participated in two fMRI sessions referred to as pre-night (around 9pm) and post-night (around 8.30am) sessions which took place at the MRI facility as well as an experimental night (EEG) session, which took place in the sleep lab between the two fMRI sessions (between 11pm-7am). At the beginning of each fMRI session, vigilance was measured objectively and subjectively using the Psychomotor Vigilance Task (PVT)[59] and Stanford Sleepiness Scale (SSS)[60], respectively (vigilance data are reported in Table S5). The pre-night MRI session started with a resting state scan (RS1). Then participants performed two motor tasks: (1) the random version of the serial reaction time task (SRTT, see below for details) to measure baseline performance / general motor execution (not scanned) and (2) the training and a short post-training test on the sequential SRTT probing motor sequence learning (both scanned). During the sequential SRTT, participants learned to perform three different motor sequences. Each sequence was associated to a different sound. Two of the three sounds associated to the learned material were replayed during the post-learning night – using auditory closed-loop stimulation – on the peak (up-state) or the trough (down-state) of slow oscillations detected online during the subsequent sleep episode (see Polysomnography and TMR section for detection algorithm details). The other sequence served as a no-reactivation control condition. The combination between the 3 reactivation conditions

(referred to as up-reactivated, down-reactivated and not-reactivated conditions), the 3 sounds and the 3 sequences was randomized across participants. After task completion, a post-task RS scan (RS2) and a structural scan were acquired. Following this first MRI session, participants were transferred to the sleep lab. During the experimental night, participants' brain activity was recorded with PSG and CL-TMR was applied during the NREM2-3 stages of the first 3 h of sleep (i.e. 3 h from the first stimulation). The post-night session took place at the MRI facility and started with a RS (RS3) that was followed by the retest on the sequential SRTT. The session was concluded with the last RS (RS4) and the random SRTT (not scanned). Note that the RS data are not reported in the present manuscript.

### Stimuli and tasks

**Motor task.** A bimanual serial reaction time task (SRTT)[61,62] was implemented in Matlab Psychophysics Toolbox version 3[63]. During the task, eight squares were presented on the screen, each corresponding to one of the eight keys on a specialized MR-compatible keyboard and to one of the 8 fingers of both hands excluding the thumbs. The color of the outline of the squares alternated between red and green, indicating rest and practice blocks, respectively. During the practice block, participants had to press as quickly as possible the key corresponding to the location of a green filled square that appeared on the screen. After a response, the next square changed to green following either a pseudo-random (see below for details) or sequential order depending on the task variant. After 20 presses, the practice block automatically turned into a rest block and the outline of the squares changed from green to red. The rest interval was 10 s.

In the sequential version of the SRTT, participants learned 3 different 5-element sequences (sequence A: 4 7 2 8 3, sequence B: 1 6 3 5 2, and sequence C: 7 3 8 4 6, where 1 is the left little finger and 8 is the right little finger) that were pseudo-randomly assigned to the three conditions. Sequence practice was organized by block (one sequence practiced per block) and each sequence was repeated 4 times per block (20 key presses per block). The order of the sequences was fixed within participants but pseudo-randomized across participants. The participants were explicitly instructed that the visual cues would appear following a sequential pattern and that there would be three different motor sequences to perform. The pre-night training session consisted of 63 practice blocks (21 blocks per sequence) immediately followed by a post-training test of 9 practice blocks (3 blocks per sequence). These two sessions were separated by 5 min in order to allow fatigue effects to dissipate and minimize its confounding effect on end-of-training performance[64]. Note that the three sets of EPI images using reversed phase-encoding polarity were acquired during this break (see fMRI data acquisition below). The post-night session consisted of 63 practice blocks (21 blocks per sequence). During all task practice sessions, three different 100-ms auditory cues (see below) pseudo-randomly assigned to sequences A, B, or C were played before the beginning of each sequence.

For the pseudo-random version of the SRTT, 12 patterns of 5-elements were created by randomly selecting 5 keys out of 8 for each of the 12 iterations which resulted in 60 key presses per block. The same random procedure was used for each block of random practice.

For both versions of the SRTT, participants were instructed to keep their fingers still and look at the squares on the screen during the rest blocks and to respond as quickly and as accurately as possible to the visual cues during the practice blocks.

**Acoustic stimulation.** Three different 100-ms sounds (previously used for our earlier research[14,65]) were pseudo-randomly assigned to the three conditions (up-reactivated, down-reactivated, and not-reactivated), for each participant. The three synthesized sounds consisted of (1) a tonal harmonic complex created by summing a sinusoidal wave with a fundamental frequency of 543 Hz and 11 harmonics with linearly

decreasing amplitude (i.e. the amplitude of successive harmonics is multiplied by values spaced evenly between 1 and 0.1); (2) a white noise band-passed between 100 and 1000 Hz; and (3) a tonal harmonic complex created with a fundamental frequency of 1480 Hz and 11 harmonics with linearly increasing amplitude (i.e. the amplitude of successive harmonics is multiplied by values spaced evenly between 0.1 and 1). A 10-ms linear ramp was applied to the onset and offset of the sound files so as to avoid earphone clicks. Before the start of the training session, a dummy MRI acquisition was launched to adjust the volume of the three different sounds. Sounds were played via MR-compatible, electrostatic headphones (MR-Confon, Magdeburg, Germany). An experimenter adjusted the volume of the sounds until the participant reported they could hear it above and beyond the scanner noise but still comfortably. The sound level determined for each of the three sounds was then used during task practice. During the reactivation session taking place during the experimental night in the sleep lab, sounds were played via ER3C air tube insert earphones (Etymotic Research). Before turning the light offs for the night, auditory detection thresholds were determined by performing a transformed 1-down 1-up procedure[66,67] separately for each of the three sounds. Subsequently, the sound pressure level was set to 2db above the individual auditory threshold, thus limiting the risk of awakening during the night. The three sounds were then presented to the participants at the intensity mentioned above to confirm that they could hear them distinctively. Before the start of the night episode, participants were instructed that they may or may not receive auditory stimulations during the night.

### EEG data acquisition and closed-loop TMR

Both habituation and experimental nights were monitored with a digital sleep recorder (V-Amp, Brain Products, Gilching, Germany; bandwidth: DC to Nyquist frequency) and were digitized at a sampling rate of 1000 Hz. Standard electroencephalographic (EEG) recordings were made from Fz, C3, Cz, C4, Pz, Oz, A1 and A2 according to the international 10–20 system (note that Fz, Pz and Oz were omitted during habituation). A2 was used as the recording reference and A1 as a supplementary individual EEG channel. An electrode placed on the middle of the forehead was used as the recording ground. Bipolar vertical and horizontal eye movements (electrooculogram: EOG) were recorded from electrodes placed above and below the right eye and on the outer canthus of both eyes, respectively. Bipolar submental electromyogram (EMG) recordings were made from the chin. Electrical noise was filtered using a 50 Hz notch.

The CL-TMR device (Elemind Technologies), that was used for the online detection and stimulation of the SOs (see detection methods below), required another set of electrodes for which the signal was recorded from FPz at a sampling rate of 500 Hz on average (ground and reference electrodes placed behind the right ear). This device has been developed to instantaneously compute the phase of the filtered signal (0.1–4.5 Hz) with the EndPoint Corrected Hilbert Transform (which allows to correct for the Gibbs phenomenon during the discrete Fourier transform of the signal[68]). During the experimental night, an experienced researcher performed online visual scoring of the polysomnography (PSG) data in order to detect NREM2-3 sleep. When these stages were reached, the phase detection algorithm was launched (see below). The auditory stimulation was presented in a blocked design with 3-min long intervals that alternated between up- and down-SO detection/stimulation. Each stimulation interval was separated by a no-stimulation rest period of a minimal duration of 1 min (see below for details on the actual duration of the no-stimulation intervals). The stimulation was manually stopped when the experimenter detected REM sleep, NREM1 or wakefulness. The CL-TMR ended 3 h after the first stimulation was sent (about 2 sleep cycles). The sounds associated to the up(down)-reactivated sequence was then played on the peak(trough) of the SOs within these alternating

intervals. The algorithm for online SO stimulation consisted of a two-step process for trough and peak detection and validation. First, the online detection algorithm used a fast-moving average filter with a window of 50 samples. A down-phase was detected when signal went below a specific threshold adapted for biological sex according to Rosinvil et al.[69] and of −41 μV in females and 39.5 μV in males. The up-phase of a SO was identified when, in addition to the criterion described for down-phase detection above, peak-to-peak signal amplitude reached 77 μV in females and 74 μV in males. After a cue was delivered, the stimulation algorithm was paused until the next zero crossing (from up to down phase) to avoid stimulating the same SO multiple times but to allow the stimulation of consecutive SOs. As a result, 90.83 % [95CI: 87.13–94.53] of the auditory cues were sent at a frequency that corresponds to the SO frequency range (i.e., 0.1 to 4.5 Hz). Importantly, as the down-phase detection relied on fewer criteria than the up-phase detection, the likelihood to send down-stimulation was higher than up-stimulation. To avoid an imbalanced number of stimulations between conditions, a second step after online detection consisted in the implementation of an online validation algorithm to match the number of accurately stimulated SO between conditions. To do so, the signal was filtered using a FIR filter between 0.1 and 4.5 Hz (order 500) and whenever a stimulation was sent, the validation algorithm assessed whether it fell within an SO (−40 μV trough and 75 μV peak to peak amplitude criteria). The online validation algorithm therefore allowed to track the number of true positives for both stimulation conditions and stimulation was dynamically adjusted depending on this count. Specifically, any time the total count of stimulation was greater for the down as compared to the up condition, the algorithm temporarily paused during the following down-detection interval. For example, if at the end of the 3 min-long up stimulation interval, the number of true positives in the up condition was lower than in the down condition, the next down-stimulation interval was silenced which resulted in a rest interval duration of 5 min (1 min rest + 3 min silenced down condition + 1 min rest) before the next up stimulation interval started. Note that the duration of the stimulation intervals could also be reduced in case the experimenter paused stimulation based on the online scoring during data collection (e.g., in case of arousals, wake, NREM1 or REM sleep). This eventually resulted in overall similar stimulation / no-stimulation interval durations and a similar number of analyzed stimuli between conditions (see section 4.6.2 below for the number of analyzed stimuli after offline validation). Specifically, the average interval duration was 2.30 min [95CI: 2.22–2.39] for the up-, 2.38 min [95CI: 2.29–2.47] for the down- and 2.58 min [95CI: 2.25–2.91] for the not-stimulated conditions. The rmANOVA on interval duration with Condition as within-subject factor (up vs. down vs. not-stimulated intervals) showed no significant condition effect ($F_{(2,58)} = 18.56$, $p = 0.070$ (0.093 sphericity corrected), $\eta^2 = 0.088$). Finally, to compute the accuracy of the stimulation in terms of both percentage of true positives stimulated and phase of the stimulation, an offline validation algorithm was applied using additional criteria that could not be implemented online due to time constraints pertaining to real-time EEG analyses. Specifically, for each stimulation sent, the offline validation algorithm assessed whether the stimulation fell within a 180° range around 90° for the up-stimulation and around 270° for the down-stimulation during a true slow oscillation defined by a trough reaching −41 μV in females and 39.5 μV in males and a peak-to-peak amplitude reaching 77 μV in females and 74 μV in males. Offline validation analyses showed that stimulation accuracy was 82.2% [95CI: 79.6–84.8] across stimulation conditions (see Supplementary Table S5) and the mean stimulation phase was 93.39° [95CI: 93.89–92.89] for the up condition and 286.51° [95CI:286.13–286.89] for the down condition (Fig. 1c). The number of true SOs stimulated was not different between conditions (up-stimulated = 622.9 [95CI: 495.2–750.7], down-stimulated = 600.5 [483.3–717.7], $t = 1.65$, $df = 29$, $p = 0.11$, Cohen's $d = 0.3$). Nonetheless,

exploratory correlation analyses were performed to test whether there was a relationship between the number of cues (total number of cues and number of true positive cues) and offline changes in performance in each condition. Results are reported in the supplements and did not show any significant correlation (see section 11 in Supplementary information).

## fMRI data acquisition

MRI data were acquired on a Philips Achieva 3.0 T MRI system equipped with a 32-channel head coil. Task-related fMRI data were acquired during the training and overnight retest sessions using an ascending gradient EPI pulse sequence for T2*-weighted images (TR = 2000 ms; TE = 29.8 ms; multiband factor 2; flip angle = 90°; 54 transverse slices; slice thickness = 2.5 mm; interslice gap = 0.2 mm; voxel size = 2.5 × 2.5 × 2.5 mm³; field of view = 210 × 210 × 145.6 mm³; matrix = 84 × 82) for each participant (max. 1200 dynamic scans). The average scan times were of 25.4 min [95CI: 24.3–26.5] for the pre-night training, of 3.5 min [95CI: 3.4–3.7] for the pre-night test, and of 22.9 min [95CI: 21.8–23.9] for the post-night training. Resting-state fMRI data were also collected prior and immediately after the training and overnight retest sessions with the same EPI sequence as above (data not reported here). Additionally, field maps (TR = 1500 ms; TE = 3.5 ms; flip angle = 90°; 42 transverse slices; slice thickness = 3.75 mm; interslice gap = 0 mm; voxel size = 3.75 × 3.75 × 3.75 mm³; field of view = 240 × 240 × 157.5 mm³; matrix = 64 × 64) were collected between the SRTT Training and Test together with three sets of EPI images using reversed phase-encoding polarity (TR = 2000 ms; TE = 29.8 ms; multiband factor 2; flip angle = 90°; 54 transverse slices; slice thickness = 2.5 mm; interslice gap = 0.2 mm; voxel size = 2.5 × 2.5 × 2.5 mm³; field of view = 210 × 210 × 145.6 mm³; matrix = 84 × 82, 6 dynamic scans). Note that these sequences were not included in the final analysis pipeline[#] (see #1 in Table S4 of the Supplementary information). High-resolution T1-weighted structural images were acquired with a MPRAGE sequence (TR = 9.5 ms, TE = 4.6 ms, TI = 858.1 ms, FA = 9°, 160 slices, FoV = 250 × 250 mm², matrix size = 256 × 256 × 160, voxel size = 0.98 × 0.98 × 1.20 mm³) for each participant.

## Analyses
### Behavioral data
**Preprocessing.** For each block of practice, motor performance on both the random and sequential SRTT was measured in terms of speed (median response time on correct trials, in ms) and accuracy (% of correct responses, with a trial classified as "correct" if the key pressed by the participants matches the visual cue). Note that correct trials were excluded from the analyses if they were outlier trials based on John Tukey's method of leveraging the Interquartile Range[#] (see #2 in Supplementary Table S4). The total number of discarded trials (inaccurate and outlier) was 10.5% across sessions and conditions (see Supplementary Table S6 for details). Corresponding analyses are presented in section 11 of the Supplementary information and showed that the number of analyzed trials did not differ between conditions. Consistent with our pre-registration, our primary analyses focused on performance speed (but see section 12 in the Supplementary information for results related to the accuracy).

The offline changes in performance on the sequential SRTT were computed as the relative change in speed between the end of the training of the pre-night session (namely during the 3 blocks of the pre-night test) and the beginning of the post-night session (3 first blocks of practice) separately for the up-reactivated, the down-reactivated, and the not-reactivated sequences. A positive offline change in performance therefore reflects an increase of absolute performance from the pre-night test to the post-night test. Additionally, we computed a TMR index which consisted of the difference in offline gains in performance speed between up-reactivated and not-reactivated

sequences (TMR index$_{up}$) as well as down-reactivated and not-reactivated sequences (TMR index$_{down}$), separately. A positive TMR index reflects higher offline changes in performance speed for the reactivated sequences as compared to the not-reactivated, control, one.

**Statistics.** Behavioral statistical analyses were performed with the open-source software R[70,71]. Statistical tests were considered significant for $p < 0.05$. When necessary, corrections for multiple comparisons were conducted with the False Discovery Rate[72] (FDR) procedure within each family of hypothesis tests. Greenhouse-Geisser corrections was applied in the event of the violation of sphericity. F and t statistics and corrected $p$-values were therefore reported for ANOVAs and student tests, respectively. Effect sizes are reported using Cohen's d for Student $t$ tests and $\eta^2$ for rmANOVAs using G*power[57].

In our confirmatory analysis, we tested whether offline changes in performance on the sequential SRTT differed between reactivation conditions after a night of sleep. To do so, a one-way rmANOVA was performed on the offline changes in performance speed (main text) and accuracy (Supplementary information and Fig. S9) with condition (up- vs. down- vs. not-reactivated) as within-subject factor. Post-hoc analysis on the 3 possible pair comparisons were performed using Student $t$ tests and FDR correction was applied[72].

We describe in the Supplementary information the control analyses that were collectively performed to verify that the pattern of behavioral results emerged from our experimental manipulation on motor memory consolidation processes rather than from various potential confounding factors listed below (see section 11 in the supplements). First, we tested whether vigilance during each behavioral session was similar using a one-way rmANOVAs on both the median RTs of the PVT and the Stanford Sleepiness Scale scores with Session as two-level factor (pre- and post-night, see Table S5). Second, we tested whether the three movement sequences were learned to the same extent during the pre-night session using two-way rmANOVAs on performance speed and accuracy measures with sequence (A vs. B vs. C) and blocks (21 for training and 3 for post-training test) as within-subject factors (Fig. S7 in the Supplementary information). Third, we performed the same analysis using condition (up-reactivated vs. down-reactivated vs. not-reactivated) – as opposed to sequence (A, B and C) – and block (21 for training and 3 for post-training test) as within-subject factors (Fig. 2a). Fourth, we tested whether the three different sounds differently impacted motor memory performance. To do so, a one-way rmANOVA was performed on the offline changes in performance speed with the 3-level factor sound type (low-frequency tone vs. white noise vs. high-frequency tone) as within-subject factor (see Fig. S8 in the supplements). Last, to highlight that improvement in movement speed was specific to the learned sequences as opposed to general improvement of motor execution, we computed the overall performance change for both the sequential SRTT (first 4 blocks of the pre-night training vs. 4 last blocks of post-night training collapsed across sequences) and the pseudo-random version of the SRTT (4 blocks pre-night session vs. 4 blocks post-night session).

**Electrophysiological data**
**Offline sleep scoring.** Offline sleep scoring was performed by a certified sleep technologist - blind to the stimulation periods - according to criteria defined in the guidelines from the American Academy of Sleep Medicine[73,74] using the software SleepWorks (version 9.1.0 Build 3042, Natus Medical Incorporated, Ontario, Canada). Data were visually scored in 30 s epochs and band pass filters were applied between 0.3 and 35 Hz for EEG signals, 0.3 and 30 Hz for EOG, and 10 and 100 Hz for EMG. A 50 Hz notch filter was also used. Sleep characteristics resulting from the offline sleep scoring as well as the distribution of auditory cues across sleep stages are shown in Table S5 of the Supplementary information. Briefly, results indicate that

participants slept 7.5 h on average (sleep efficiency: 83.3 %) and that cues were accurately presented in NREM sleep (stimulation accuracy mean: 98.5% [95CI: 97.6–99.3]; up-reactivated cues: 98.8 % [95CI: 98.1–99.4]; down-reactivated cues: 98.2% [95CI: 97.2–99.3]).

**Preprocessing.** EEG data preprocessing was carried out using functions supplied by the fieldtrip toolbox[75]. EEG was re-referenced to an average of A1 and A2 and filtered between 0.1–30 Hz. Specifically, data were cleaned by manually screening each 30-s epoch. Data segments contaminated with muscular activity or eye movements were excluded. Independent component analysis was used to remove cardiac artifacts.

**Event-related analyses.** We performed both the pre-registered auditory-cue-locked analyses as well as exploratory SO-trough-locked analyses. Trough-locked analyses were provided to complement the pre-registered analyses in order to reduce the influence of the phase of the SO on activity evoked by the stimulation (see details in section 1 of the Supplementary information). In both cases, the EEG analyses were performed on preprocessed cleaned data down-sampled to 100 Hz.

For the pre-registered cue-locked analyses, event-related potentials were computed for both up and down cues (up- and down-stimulated) as well as for sham (up- and down-sham) cues derived from non-stimulated intervals to perform exploratory comparisons (up-stimulation vs. up-sham and down-stimulation vs. down-sham). For the up and down-stimulation cues, only correct stimulation (i.e. sent during SOs classified as true positives based on the offline validation algorithm described above) were considered for the analyses. Sham cues were retrospectively derived for the non-stimulated SOs from the silent intervals. For each SO detected during the no-stimulation interval, the sham cue time-sample was defined as the time point of a randomly picked phase at which a true positive cue was sent during the up- and the down-stimulation intervals. Namely, each not-stimulated SO was assigned an up-sham cue and a down-sham cue at a phase of the corresponding condition. The cue-locked potentials were obtained by segmenting the data into epochs time-locked to the true or sham cues from −2.5 to 2 s and averaged across all trials in each condition separately (up-stimulated, up-sham, down-stimulated, and down-sham). The average number of artifact-free cue-locked trials by condition was 622.7 [95% CI: 495.0–750.5] for the up-stimulation, 599.8 [95% CI: 482.4–717.2] for the down-stimulation, and 667.5 [95% CI: 532.7–802.3] for both up- and down-sham conditions (up-stimulation vs. up-sham: t(29) = −1.71, $p = 0.10$, Cohen's d = 0.13; down-stimulation vs. down-sham: t(29) = −2.47, $p = 0.020$, Cohen's d = 0.20; up-stimulation vs. down-stimulation: t(29) = 1.69, $p = 0.10$, Cohen's d = 0.070).

For the trough-locked analyses, the trough time-sample of each stimulated SO classified as true positive based on the offline validation algorithm described above (but see Supplementary Fig. S10 for analyses based on all events irrespective of accuracy) was extracted from both the up- and down-stimulation intervals. We also extracted the trough time-sample of each offline-detected SO occurring during the silent intervals (referred to as not-stimulated SO). Trough-locked responses were obtained by segmenting the data into epochs time-locked to the trough of the SOs offline-detected on FPz (from −2 to 2 s) for the up-, the down- and the not-stimulated SO and averaged across all trials in each condition separately. The average number of artifact-free trials by condition was of 622.9 [95% CI: 495.1–750.7] for the up-, 599.7 [95% CI: 482.4–717.1] for the down-, and 664.3 [95% CI: 528.7–800.0] for the not-stimulated conditions. The rmANOVA on the number of trials with condition as within-subject factor (up vs. down vs. not-stimulated) showed a significant condition effect (F(2,58) = 3.95, $p = 0.025$, $\eta^2 = 0.12$). Post hoc analyses on the 3 possible paired-comparisons showed, however, that this effect did not survive correction for multiple comparisons (up vs. down: t = 1.70, df = 29,

$p = 0.10$ (0.13 FDR-corrected), Cohen's $d = 0.070$; up vs. not-stimulated: $t = -1.58$, $df = 29$, $p = 0.13$ (0.13 FDR-corrected), Cohen's $d = 0.12$; down vs. not-stimulated: $t = -2.35$, $df = 29$, $p = 0.026$ (0.077 FDR-corrected), Cohen's $d = 0.19$). To analyze oscillatory activity, we computed TFRs of the power spectra locked to either the auditory cues or the SO-troughs (see above) for each experimental condition and per channel. To this end, we used an adaptive sliding time window of five cycles length per frequency ($\Delta t = 5/f$; 20-ms step size), and estimated power using the Hanning taper/FFT approach between 5 and 30 Hz. Individual TFRs were converted into change of power relative to power from −2.5 to −2 s relative to the cue for the cue-locked analyses and relative to the entire period around the SO trough (from −2 s to 2 s relative to trough) for SO-trough-locked analyses. Note that statistical analyses were performed on a more conservative −1.5 s to 1.5 s relative to the cue or to the SO-trough to avoid border effects.

For both cue- and trough-locked analyses, ERP and TF were analyzed using nonparametric CBP tests[76] implemented in fieldtrip toolbox. We used paired $t$ test between conditions and cluster-based correction to account for multiple comparisons across time and space for the ERP analyses, and time, frequency and space for the TF analyses. All time-space (ERP analyses) and time-frequency-space (TF analyses) samples whose t-values exceeded a threshold of alpha cluster of 0.01 were considered as candidate members of clusters, i.e., samples clustered in connected sets on the basis of time and space adjacencies for ERP analyses and on the basis of time, frequency and space adjacencies for the TF analyses. The sum of t-values within every cluster, that is, the 'cluster size', was calculated as test statistics. These cluster sizes were then tested against the distribution of cluster sizes obtained for 500 partitions with randomly assigned conditions within each individual. The clusters were considered significant at $p < 0.05$. For CBP contrast analyses, Cohen's d is reported. Corrections for three comparisons, i.e., $p < 0.0083$, was conducted with Bonferonni procedure within each family of hypothesis tests[#] (see #3 in Table S4 of the Supplementary information).

Given that the results of the cue-locked analyses are highly confounded by the phase of SO in the current study, the main text focused on the results of the SO-trough-locked analyses while the results of the cue-locked analyses (as well as interpretations and comparison with previous literature) are reported in the Supplementary information (Fig. S1). Further, note that event-related phase amplitude coupling analyses were also pre-registered but eventually not performed as these analyses were redundant with the SO-trough locked analyses[#] (see #4 in Table S4 of the Supplementary information) described above.

Additional control analyses were performed to test whether the nature of the different sounds influenced sleep oscillations. To do so, we analyzed the EEG data with the same procedure as described above for SO-trough-locked analyses but using sounds (low-frequency tone vs. white noise vs. high-frequency tone) instead of condition as within-subject factor. Corresponding results are reported in section 11 of the Supplementary information and show that the nature of the sound did not influence the pattern of EEG results presented in the main text.

**Sleep events detection.** Sleep spindles and SOs were detected on all EEG channels automatically a posteriori in NREM sleep epochs during the reactivation period by using the YASA open-source Python toolbox[77]. This analysis included all detected sleep events in intervals of stimulated and not-stimulated intervals. Preprocessed cleaned data were down-sampled to 500 Hz and were transferred to the python environment. Concerning the spindle detection, the algorithm is inspired from the A7 algorithm described in Lacourse et al.[78] with spindle characteristics described in Purcell et al.[79]. The relative power in the spindle frequency band (8–18 Hz) with respect to the total power in the broadrunn-band frequency (1–30 Hz) is estimated based on

Short-Time Fourier Transforms with 2-s windows and a 200-ms overlap. Next, the algorithm uses a 300 ms window with a step size of 100 ms to compute the moving root mean squared (RMS) of the filtered EEG data in the sigma band. A moving correlation between the broadband signal (1–30 Hz) and the EEG signal filtered in the spindle band is then computed. Sleep spindles are detected when the three following thresholds are reached simultaneously: (1) the relative power in the sigma band (with respect to total power) is above 0.2, (2) the moving RMS crosses the $RMS_{mean} + 1.5\ RM_{SSD}$ threshold, and (3) the moving correlation described is above 0.65. Additionally, detected spindles shorter than 0.3 s or longer than 3 s were discarded. Spindles occurring in different channels within 500 ms of each other were assumed to reflect the same spindle. In these cases, the spindles are merged together. Concerning the SO detection, the algorithm used is a custom adaptation from[80,81]. Specifically, data were filtered between 0.3 and 2 Hz with a FIR filter using a 0.2 Hz transition resulting in a −6 dB points at 0.2 and 2.1 Hz. Then all the negative peaks with an amplitude between −40 and −200 μV and the positive peaks with an amplitude comprised between 10–150 μV are detected in the filtered signal. After sorting identified negative peaks with subsequent positive peaks, a set of logical thresholds are applied to identify the true slow oscillations: (1) duration of the negative peak between 0.3 and 1.5 sec, (2) duration of the positive peak between 0.1 and 1 s, (3) amplitude of the negative peak between 40 and 300 μV, (4) amplitude of the positive peak between 10 and 200 μV, and (5) PTP amplitude between 75 and 500 μV.

We extracted the frequency and the amplitude of spindles as well as the density of both spindles and SOs[#] (see #5 in Table S4 of the Supplementary information for considerations about SO amplitude measures). On these variables of interest, we performed one-way rmANOVAs with condition (events occurring during up- vs. down- vs. not-stimulated intervals) as within-subject factor using the software R[70,71] and Greenhouse-Geisser corrections was applied in the event of the violation of sphericity. Statistical tests were considered significant for $p < 0.05$. When a condition effect was detected, post-hoc analysis on the 3 possible paired-comparisons were performed using Student $t$ test and FDR correction was applied[72].

**fMRI data.** Statistical parametric mapping (SPM12; Welcome Department of Imaging Neuroscience, London, UK) was used for the pre-processing of the functional images and the statistical analyses of the BOLD data.

**Preprocessing.** Preprocessing included the realignment of the functional time series using rigid body transformations, iteratively optimized to minimize the residual sum of squares between each functional image and the first image of each session separately in a first step and with the across-session mean functional image in a second step (mean of maximum movement in the three dimensions: 1.49 mm [95CI: 0.91–2.07] for the pre-night training session and 0.96 mm [95CI: 0.73–1.18] for the post-night training session). Movement was considered as excessive when exceeding more than 2 voxels in either of the three dimensions for both the pre- and post-night sessions (one individual was excluded from data analyses because of such excessive movement). The pre-processed functional images were then co-registered to the structural T1-image using rigid body transformation optimized to maximize the normalized mutual information between the two images. The anatomical image was segmented into gray matter, white matter, cerebrospinal fluid (CSF), bone, soft tissue and background and the individuals' forward deformation fields were used for the normalization step. All functional and anatomical images were normalized to the MNI template (resampling size of $2 \times 2 \times 2$ mm). Functional images were spatially smoothed using an isotropic 8 mm fullwidth at half-maximum [FWHM] Gaussian kernel.

**Activation-based analyses.** The analysis of the task-based fMRI data, based on a summary statistics approach, was conducted in two serial steps accounting for intra-individual (fixed effects) and inter-individual (random effects) variance, respectively. Changes in brain regional responses was estimated for each participant with a model including responses to the three motor sequences (up- vs. down- vs. not-reactivated) and their linear modulation by performance speed (median response time on correct key presses per block) for each task run (pre-night training, pre-night test, post-night training). The rest blocks occurring between each block of motor practice served as the baseline condition modeled implicitly in the block design. These regressors consisted of boxcars convolved with the canonical hemodynamic response function. Movement parameters derived from realignment as well as erroneous key presses were included as covariates of no interest. High-pass filtering was implemented in the design matrix using a cutoff period of 128 s to remove slow drifts from the time series. Serial correlations in the fMRI signal was estimated using an autoregressive (order 1) plus white noise model and a restricted maximum likelihood (ReML) algorithm. Linear contrasts were generated at the individual level to test for (1) the main effect of practice (across sequences) and its linear modulation by performance, (2) the main effect of practice for each sequence (up-, down-, and not-reactivated) and (3) the difference in brain responses between sequences (up- vs. down- vs. not-reactivated). These contrasts were written within each of the two training runs (pre-night and post-night training)[#] (see #6 in Table S4 of the Supplementary information) as well as between these runs. The resulting contrast images were further spatially smoothed (Gaussian kernel 6 mm Full Width at Half Maximum (FWHM)). The resulting contrast images were entered in a second level analysis for statistical inference at the group level (one sample $t$ tests), corresponding to a random effects model accounting for inter-subject variance.

**Connectivity-based analyses.** Task-related functional connectivity was examined using psychophysiological interaction (PPI) analyses with a data-driven seed selection approach. Specifically, we assessed connectivity of three seed regions (right caudate ($x = 10$, $y = 14$, $z = 12$), right putamen ($x = 18$, $y = 12$, $z = -2$), and right hippocampus ($x = 32$, $y = -38$, $z = -6$)) revealed by the univariate analyses and showing a main effect of session across multiple conditions (see Fig. S11 in Supplementary information). Note that peak maxima for significant clusters in the hippocampus, putamen and caudate were located in the right hemisphere across the large majority of tested contrasts (see Supplementary Table S2-1 and 2) and that this laterality effect was not hypothesized a priori. For each individual, the first eigenvariate of the signal was extracted using Singular Value Decomposition of the time series across the voxels included in a 10 mm-radius sphere centered on these coordinates. Linear models were generated, at the individual level, with a first regressor representing the practice of the motor sequence (pre- and post-night sessions in each of the three reactivation conditions), a second regressor corresponding to the BOLD signal in the seed and a third regressor representing the interaction between the first (psychological) and second (physiological) regressors. To build this regressor, the underlying neuronal activity was first estimated by a parametric empirical Bayes formulation, combined with the psychological factor, and subsequently convolved with the hemodynamic response function[82]. The individual linear contrasts testing for the interaction between the psychological and physiological regressors within and between the different runs mentioned above were then further spatially smoothed (Gaussian kernel 6 mm FWHM). The resulting contrast images were entered in a second level analysis for statistical inference at the group level (one sample $t$ tests), corresponding to a random effects model accounting for inter-subject variance.

**Regression analyses.** We performed regression analysis between the individuals' brain maps showing between session changes in activity/connectivity within each condition and the individuals' TMR index (for each condition separately, i.e. TMRindex$_{up}$ and TMRindex$_{down}$). [See also Supplementary Fig. S5 for results of the same regression analyses but using maps showing between session changes in activity/connectivity between conditions]. These regressions were performed in a second level analysis for statistical interference at the group level (one sample $t$ test), corresponding to a random effects model accounting for inter-subject variance. Finally, we performed exploratory regression analyses between the individuals' brain maps showing between session changes in activity/connectivity within each condition and the EEG sigma power as well as the peak amplitude of the SOs. For these analyses, the significant clusters from the event-related potentials and the oscillatory activity analyses of the up- vs. down-stimulated contrasts were used (Fig. 2). The amplitude and power for each individual were averaged across all channels between 0.32–0.64 s post-trough (peak amplitude) and between 0.25–0.4 s post-trough and 12–17 Hz (sigma power). These regression analyses were performed separately for each sleep feature (i.e., SO amplitude and sigma power) in a second level analysis for statistical interference at the group level (one sample $t$ test), corresponding to a random effects model accounting for inter-subject variance.

**Statistics.** The set of voxel values resulting from each second level analysis described above (activation, functional connectivity and regression analyses) constituted maps of the t statistic [SPM(T)], thresholded at $p < 0.005$ (uncorrected for multiple comparisons). The goal of the fMRI analyses was to examine brain patterns elicited in specific motor-related regions. Therefore, statistical analyses were performed across all the voxels of a large mask including a set of task-relevant brain regions involved in motor sequence learning processes[22,83–87] and consisting of the primary motor cortex (M1), the supplementary motor cortex (SMA), the premotor cortex (PMC), the anterior part of the superior parietal lobule (aSPL), the hippocampus, the putamen and the caudate nucleus (and see section 15 in the supplements for additional information about the functional role of these brain regions in motor sequence learning and sleep-related motor memory consolidation). These brain areas were defined with the brainnetome atlas as follows. M1 contained the upper limb and hand function regions of Brodmann area (BA) 4. The premotor cortex (PMC) was defined as the dorsal (A6cdl; dorsal PMC) and ventral (A6cvl; ventral PMC) part of BA 6. aSPL was defined to include the rostrocaudal areas of inferior parietal lobule (39rd and 40rd), as well as the intraparietal area 7 (A7ip) and the lateral area 5 of the superior parietal lobe (A5l). The SMA was defined as part A6m of the superior frontal gyrus and area 4 of the paracentral lobule (a4ll). The probability maps of these cortical areas were thresholded at 50% for binarization. The hippocampus mask included both rostral and caudal parts of the hippocampus. The caudate mask included both dorsal and ventral parts of the caudate. Finally, the putamen mask included both ventromedial and dorsolateral part of the putamen. The probability maps of these subcortical areas were thresholded at 5% for binarization.

Statistical inferences were performed at a threshold of $p < 0.05$ after family-wise error (FWE) correction for multiple comparisons over small volumes[88] (SVC, 10 mm radius) located in the structures of interest reported by published work (see Table S7 in Supplementary information) investigating motor sequence learning[89–94] and motor memory consolidation[23,94,95], using both implicit[23,89,91] and explicit[89,90,92–95] versions of motor sequence learning such as the finger tapping task[90,93–95] or the serial reaction time task[23,89,91,92] (see Table S8 in Supplementary information for more information about tasks and processes examined). All results reported and discussed in the main text survived SVC.

**Reporting summary**

Further information on research design is available in the Nature Portfolio Reporting Summary linked to this article.

## Data availability

The raw data collected in this study have been deposited in the publicnEUro database (https://doi.org/10.70883/qxbh2876) under accession code PN000004: Closed Loop Sleep Motor Memory using a brain imaging data structure (BIDS) format. These data are available under restricted access to adhere to the European General Data Protection Regulation. Unlimited access to the data can be provided after signing a Data User Agreement. Users then receive a link to access the data with a 72-h validity. Source data files are provided with this paper.

## Code availability

The source code is available at https://github.com/judithnicolas/Closed-Loop-Sleep-Motor-Memory.

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

## Acknowledgements

This work was supported by the Belgian Research Foundation Flanders (FWO; G0D7918N, G.A., S.P.S. and B.R.K.), The Fond de Recherche en santé du Québec en sciences naturelles (RRQNT-2018–264146, J.C. and J.D.), Healthy Brain for Healthy Lives Discovery Grant Program from the Canada First Research Excellence Fund (J.D .and J.C.) and internal funds from KU Leuven (C12/18/007, G.A.). G.A. also received support from FWO (G0B1419N, G099516N, 1524218 N, G.A.) and Excellence of Science (EOS, 30446199, MEMODYN, G.A. and S.P.S.). Financial support for authors J.N. and B.R.K. was provided by the European Union's Horizon 2020 research and innovation program under the Marie Skłodowska-Curie grant agreement (#887955 for J.N. and #703490 for B.R.K.).

## Author contributions

J.D., J.C., G.A., B.R.K., S.P.S., and J.N. contributed to the funding acquisition. J.D., J.C., G.A., B.R.K., D.L., D.W., L.L., N.G., and J.N. contributed to the design and the methodology. J.N., G.A., and G.L. contributed to the data acquisition and curation. J.N., D.L., L.L., B.R.K., and G.A. contributed to the formal analysis. J.D., J.C., B.R.K., S.P.S., J.N., and G.A. contributed to the project administration, supervision. J.N., B.R.K., and G.A. contributed to writing the original draft, reviewing and editing. J.D., J.C., G.A., B.R.K., D.L., D.W., L.L., N.G., G.L., and J.N. reviewed and edited the final draft.

## Competing interests

N.G. and D.W. are the applicants and co-inventors of the patent of the device used in this experiment. N.G. and D.W. assigned their rights to Massachusetts Institute of Technology (MIT) and NuVu Studio, respectively, who were the original patent applicants. The patent application (No.: US 2017/0020447 A1 US 20170020447A1) has been granted in the U.S., U.S. Patent No. 2017002O447A1 and the current co-owners of the patent are MIT and Elemind Technologies. D.W. is the co-founder of Elemind Technologies, the company which made the device used for the study. As a co-founder, he has shares in Elemind. Elemind's responsibility was to provide hardware and technical support. D.W. did not participate in the study design, experimentation or data analysis and he did not influence the outcome of the study. Elemind currently has no plan to commercialize the research described in this manuscript. The benefits from this work for Elemind Technologies are: 1. to learn about the possibilities the technology enables; 2. to be acknowledged for the technological contribution; 3. to advance the field of research in phase-based stimulation technologies. The other authors declare no competing interests.
