## [Transparent Peer Review file · Nature Communications]

Unraveling the Neurophysiological Correlates of Phase-Specific Enhancement of Motor Memory Consolidation via Slow-Wave Closed-Loop Targeted Memory Reactivation

Corresponding Author: Professor Geneviève Albouy

Version 0:

Reviewer comments:

Reviewer #1

(Remarks to the Author)

In this study Nicolas and colleagues investigated whether applying targeted memory reactivation at specific phases of slow oscillations (up vs. down vs. no reactivation) would influence behavioral and neural aspects of motor memory consolidation. The authors report that up- (as compared to down-) state cueing resulted in greater performance improvement, while up-stimulated slow oscillations exhibited higher amplitude and greater peak-nested sigma power. fMRI results revealed that up-state cueing elevated activity in striato-motor and hippocampal circuits. This is in general an interesting and timely study. However, some features of the manuscript seem problematic and need to be addressed by the authors.

Comment 1: The authors report that memory performance did not significantly differ between the up- and not-reactivated sequences, while down-state TMR had a detrimental effect on memory (as compared to up-state and not-reactivated). In light of these results, I would strongly encourage the authors to tune down their claim that memory consolidation was specifically boosted by SO-up-phase TMR. This would be true if up-state TMR would have led to superior performance compared to not-reactivated sequences. As briefly discussed by the authors, the pattern of results indeed seems to suggest that down-state TMR might have had a detrimental effect rather than up-state TMR having a beneficial one.

Comment 2: Do I understand correctly that EEG results (i.e., ERPs and TFRs) are based on all SOs detected during the simulation periods and not necessarily SOs that triggered TMR (in case of up- and down-state TMR? If that is the case, please re-run these analyses taking only relevant SOs (TMR targeted) into account. In case that only relevant SOs were taken into account, how do the authors interpret the finding that differences in spindle power between up- and down-state TMR (Fig. 3a) already emerged before the time window for up-state stimulation? In addition, while locking the data to SO throughs is a clever idea, please also add a figure comprising the same analyses for trigger-locked data (when comparing up- and down-state TMR). This would allow easier comparison with earlier studies (e.g., Ngo et al., 2022). Furthermore, these results would indicate whether up- and down-state TMR would differ in their capacity to entrain SOs.

Comment 3: One systematic difference between the up- and the down-state TMR conditions is that according to table S5 the algorithm accuracy for down-state TMR (74.9 %) is lower as compared to up-state TMR (89.5 %). Hence due to the implemented 'online validation algorithm', which was administered to match the number of accurately stimulated SO between conditions, the overall number of down-state TMR cues was significantly higher (791 vs 700). Does the number of cue presentations correlate with memory performance? In addition, please show ERPs and TFRs not only for 'true positive' events but also when incorporating all events.

Comment 4: Somehow related to above. Why was the threshold for down-state detection set to around $-40 \mu\text{V}$? As visible in Fig. 3 down-states peaked around $-80 \mu\text{V}$. Would this threshold explain why the accuracy of targeting down-states was rather modest?

Comment 5: It seems surprising that spindle density was lower during both up- and down-stimulated as compared to not-stimulated blocks. I would be curious how the authors interpret this finding. Relatedly spindle densities seem very low in general (Fig. S2b). Could it be that the applied algorithm is too conservative?

Comment 6: Please integrate the circular plot indicating the stimulation phase (Fig. S7) in Fig. 3. Also show phase angles across all trials (for up- and down-state cues, respectively).

Comment 7: The authors report that

... the overnight increase in striatal activity was greater for the up-reactivated sequence as compared to the down-reactivated sequence, which in turn was greater than for the not-reactivated sequence (Figure 4b)...

Was this effect restricted to the right caudate or also evident in the putamen and M1? Please add the right caudate to Fig. 4a (overnight increase in activity within condition) to allow for excluding that condition specific differences in overnight activity increases were driven by baseline effects.

Comment 8: For correlational analyses with behavior (e.g., Fig. 4c and 5c) the authors computed a TMR index which consisted of the difference in offline gains in performance between up-reactivated and not-reactivated sequences, as well as down-reactivated and not-reactivated sequences. Is this metric based on accuracies? As shown in Fig. S5 TMR did not have any effect on accuracies, which makes me wonder why this would be a useful proxy for TMR success.

Minor comments:

- Please include the proportion of female/male among the N=31
- How were stimulation phases (Figure S7) computed?
- Did the sounds (low-frequency tone, white noise, high-frequency tone) differ in their capacity to impact sleep oscillations and memory performance?
- Please include line numbers

Reviewer #2

(Remarks to the Author)

The authors report a study on the phase specific effects of closed-loop targeted memory reactivation during sleep on a serial reaction time task. The authors' report that cueing during the up-phase of SOs leads to greater performance improvement compared to cueing during the down-phase. Up-stimulated SOs also had higher amplitude than down- or non-stimulated SOs and greater sigma power than down-stimulated SOs. fMRI analyses within the different stimulation conditions showed a variety of correlations between behavior and changes in BOLD responses both locally in the striatum and the hippocampus and in connectivity between the hippocampus, striatum, and motor cortex. Overall, this is an interesting study looking at phase-specific TMR and the authors' efforts in bringing closed-loop TMR to motor memory domain is appreciated. However, there are concerns with the analysis methods, results curation, and interpretation of results.

While there is novelty in describing changes in the motor memory related functional networks before and after sleep and their relation to sleep oscillations and motor performance, this study offers limited conceptual advances or solid empirical contributions to the field. There are major methodological and data-related limitations that challenge the conclusions drawn in the study. The concluding claim that "this study shows a phase-specific beneficial effect of slow-oscillation closed-loop TMR on consolidation such that motor memories reactivated during the up-phase of slow oscillations exhibited greater performance....." is unwarranted as the data, as collected and presented, do not adequately support these conclusions. Behavioral analysis. The authors do not report a significant consolidation effect between post-sleep training (trials 1-3) and the test trials for either speed or accuracy across any condition. As the task is not consolidated by participants in the study, it is difficult to draw any conclusions about memory consolidation by analyzing the behavioral data. Previous work has reported sleep-dependent improvements in reaction time in similar SRTT tasks. Other work using TMR with a similar SRTT task also found an improvement in uncued but not cued sequences (e.g. Cousins 2016). This discrepancy is not addressed in the discussion.

When examining the differences in reaction time changes between testing and post-sleep training, a rm-anova was used to report a significant decrease in the downstate cued sequence vs. the upstate and non-cued sequence. However, when examining the data in figure 2, the medians of the group don't seem to be significantly different, with the mean differences appearing to be driven by a few participants with very large decreases in performance in the down-cued sequence. A non-parametric or robust statistical approach may be warranted with this distribution of data.

It would be useful to characterize the artifactual blocks that are removed from analysis more completely. The mean percent removed from each TMR condition should be compared, as well as if the removed blocks differed in the PM-training, testing, or the AM-testing. Additionally, it would be useful to clarify how calculations were performed if the blocks used to calculate the over-sleep changes were removed as artifactual.

Based on these considerations, the conclusion 'Our behavioral results indicate that TMR applied at the up-phase of the SO resulted in greater gains in motor performance than when administered at the down-phase of the SO.' (pp. 13, line 383) is not supported.

Electrophysiological Analysis. Different SO's are stimulated in the down-reactivated vs. up-reactivated stimulation conditions because of the additional peak-to-peak amplitude threshold with the up-reactivated. This leads to different stimulation numbers for the up vs. down. We appreciate the authors' addressing this issue with online validation algorithm and offline validation algorithm. However, this difference still leads to issues with understanding the effect of stimulation as in the down-reactivation some actual SOs are not being stimulated and more non-SOs are being stimulated. Removing the 'false-positive' stimulation events when the down-state and peak-to-peak amplitude thresholds were not reached removes a

significant number of stimulation events (~ 10% of the up-reactivated cues and ~24% of the down-reactivated cue data based on Table S5). Presumably these stimulation events also affected participant electrophysiology. Not including these in the EEG analysis nor the correlation analysis between BOLD/connectivity changes and EEG measures compromises the interpretability of the stimulation results.

Using trough-locked analysis does not allow an evaluation of the acute effects of stimulation. This analysis also makes it difficult to see if, on average, the detected oscillation that is trough-locked is an evoked oscillation with a sound stimulus presented in the 1.5 second pre-trough period. However, due to the stimulation protocol, this might be difficult to address with previous analysis methods in the field.

The authors showed that up-stimulated SOs showed higher sigma power compared to down-stimulated SOs, but the spindle density did not differ between the two conditions. It is possible that the increased power detected is a result of heightened arousal/noise. The discussion of spindles, ripples and SOs being important in memory consolidation does not accurately reflect the finding as sigma power does not equal sleep spindles.

MRI Analysis. The analysis methodology and selective presentation of results makes it difficult to evaluate imaging results. The ROI selection is not well-motivated. The authors masked the ROI regions and then found significant clusters using voxelwise analyses before correcting with SVC. SVC is a correction that can be applied when the a priori hypothesis does not apply to the entire brain, but only to specific areas that can be encompassed within a small volume. This highlights the importance of building a strong a priori hypothesis that details the separate functional roles of each ROI selected. Moreover, the cluster size for the statistically significant ROIs was not included in the results.

Seed Regions. For Connectivity-based analysis, the justification for using only right hemisphere seeds 'in order to limit the number of PPI analyses' (pp. 30, line 886) is not scientifically justified. The additional restriction to limit it to the right side was also based on a data driven difference as compared to an a priori hypothesis.

The seed region for the hippocampus is a small section of the hippocampus, please justify why the ROI should not include the entire hippocampus or a subregion defined a priori. This significant cluster also includes white matter within activated cluster of voxels (in figure S8 lateral to the right hippocampus). Additionally, the seed regions are quite large for the caudate with small significant clusters.

MRI Correlation with Behavior. As there does not appear to be consolidation with any of the conditions and TMR does not appear to improve consolidation, the TMR index is not an interpretable measure of performance. A more interpretable correlation analysis between activity and behavior would be the Post-Pre BOLD change with the over-sleep performance improvement. Additionally, for the Post-Pre Bold changes, the scanning for the entire task performance was used, however for the correlated behavior, only the testing and first three AM training trials were used. It would be more meaningful to compare behavioral and MRI data collected over the same period.

Minor issues.

Figure S7. The caption does not match the figure for the mean stimulation angle.

Imaging. MRI Scan time is not stated.

Reviewer #3

(Remarks to the Author)

Sleep helps to consolidate novel information for long-term. A process that is mediated by specific brain rhythms (and a specific interplay thereof) and relies on a reactivation of said information during sleep. Here, Nicolas et al. take advantage of a variety of techniques to examine sleep-dependent consolidation of motor memory: They perform targeted memory reactivation in real-time to reactivate motor traces in slow oscillation up- and down-states. This is further combined with fMRI pre- and post-sleep to assess changes in brain activity. Their data suggests that it is indeed possible to manipulate consolidation of motor memory, which is accompanied by activation changes within striatal and hippocampal networks.

All in all, this project must have been a difficult undertaking. Based on a clear study design and methodological approach, Nicolas et al. have gathered compelling evidence for the feasibility of real-time TMR within the procedural memory domain. I only have a few comments, I was hoping the authors could address to clarify a few aspects.

Major comments

Stimulation-locked evoked potentials

To assess the immediate effect of TMR, the authors use a different approach than other publications. Instead of averaging the signal time-locked to the stimulation onset, they use the previous online detected trough. I saw that the authors motivated this approach in the supplementary material, but I don't agree with that. They say this approach removes influence of the phase and I understand the idea is that it allows a direct comparison between all three conditions, however, the phase is exactly the targeted here. This approach also disadvantages the up-condition because the time between trough and peak should fluctuate across stimulation, which causes a temporal smearing/jitter of evoked responses. This in turn affects the up-down comparison. The authors should either derive control no-stim intervals for both up and down and compare each to their stimulation counterpart, or if they want to compare up vs. down, they could time-lock to the stimulus onset and ignore anything prior to the stimulation.

Behavioural results

For one, did the authors examine the behavioural data for outliers? It seems the offline change for the down-condition is driven by some negative outliers. This additional point is not really major, but the direction of an improvement confused me. While in Figure 2a a smaller reaction time is better, the opposite direction is expected for Figure 2B. The authors do mention that in the methods section; however, it would help to highlight that earlier on as well.

Overview of results

In a similar vein to my previous comment, given the sheer number of results, it was very difficult for me to keep an overview. For instance, while behaviourally, they only found an effect for the down condition (see later comment), physiological results show effects for both up and down or only down. This is perhaps asking too much, but it would really help to have some kind of graphical summary for the different results.

Figure 4 and ROIs

Something that confused me as well was, why the authors used so many different ROIs for their analysis, e.g., in Figure 4. Why are regression analysis performed on different ROIs than found previously when looking at differences between conditions? Can the authors please motivate this as it seems I am missing something. Related to this, while the TMR index is derived by the difference between cued and uncued conditions, has the same been done for the BOLD response? I did not find that information.

Behavioural interpretation

In the first sentence of the 2nd paragraph of the discussion, the authors write “Our behavioral results indicate that TMR applied at the up-phase of the SO resulted in greater gains in motor performance than when administered at the down-phase of the SO.” While it is true that the up-condition outperforms the down-condition, this is not the whole story. If we consider the uncued condition, we see that up-stimulation had no real benefit but rather down-stimulation lead to an impairment of memory. Another interpretation could be that up-state stimulation reactivated both the up- and no-stim sequences. Can the authors please clarify their statement or rephrase it.

Minor comments

- Page 23, line 593 (Motor task): Out of curiosity, was there a pause between the training and testing blocks before sleep?
- Page 24, line 646 (Closed-loop TMR) Was there a pause after a cue was delivered?
- Page 24, line 674 (Closed-loop TMR): I might have missed it, but what was the number of detected no-stimulation events? I assume those numbers were much lower compared to the up- and down-condition considering it was performed in only 1-min intervals. If that is the case, how was that considered in the data analysis?
- Page 30, line 902 (Regression analysis): I know that some do not put too much weight on regression assumptions, but I do find it important to show that at least homoscedasticity and normally distributed residuals are present. Otherwise, the regression models are not fully interpretable. Did the authors perform those assumption checks?

Version 1:

Reviewer comments:

Reviewer #1

(Remarks to the Author)

The authors addressed all my concerns. I have nothing more to add.

Reviewer #2

(Remarks to the Author)

The authors comprehensively addressed all of our comments and the results stand. No further concerns.

Reviewer #3

(Remarks to the Author)

I thank the authors for providing such a thorough review addressing all my concerns (and those of the other reviewers as well). It is great that the authors toned down their conclusion regarding the up-state stimulation and provided the time-locked analysis. Hence, I do not have any further comments.

REVIEWER 1

General comments: In this study Nicolas and colleagues investigated whether applying targeted memory reactivation at specific phases of slow oscillations (up vs. down vs. no reactivation) would influence behavioral and neural aspects of motor memory consolidation. The authors report that up- (as compared to down-) state cueing resulted in greater performance improvement, while up-stimulated slow oscillations exhibited higher amplitude and greater peak-nested sigma power. fMRI results revealed that up-state cueing elevated activity in striato-motor and hippocampal circuits. This is in general an interesting and timely study. However, some features of the manuscript seem problematic and need to be addressed by the authors.

Author Response: We are grateful to the reviewer for these positive remarks about our work and for the time devoted to our manuscript. We have addressed the points raised by the reviewer in the detailed response below and we hope they find our responses and the corresponding updates to the manuscript (highlighted in yellow in the revised manuscript) satisfactory.

Point #1: The authors report that memory performance did not significantly differ between the up- and not-reactivated sequences, while down-state TMR had a detrimental effect on memory (as compared to up-state and not-reactivated). In light of these results, I would strongly encourage the authors to tune down their claim that memory consolidation was specifically boosted by SO-up-phase TMR. This would be true if up-state TMR would have led to superior performance compared to not-reactivated sequences. As briefly discussed by the authors, the pattern of results indeed seems to suggest that down-state TMR might have had a detrimental effect rather than up-state TMR having a beneficial one.

Author Response: We thank the reviewer for pointing this out and replaced all the instances in the manuscript where SO-up-phase TMR was described to boost memory consolidation by statements that are closer to the data and specifying the contrasts supporting this conclusion. Accordingly, we now always specify that memory consolidation was boosted by SO-up-phase TMR **as compared to SO-down-phase TMR**. These instances can be found in the abstract and the introduction of the revised manuscript. Please see below for two representative examples:

- Page 2: “Sleep electrophysiological data indicated that up- **(as compared to down-)** stimulated SOs...”
- Page 4: “Our main results showed that consolidation¹⁸, SO amplitude^{17,19,21}, sigma band power^{17,19} and task-related brain responses in hippocampo- and striato-cortical networks^{12,22} were boosted by SO-up-phase TMR **as compared to SO-down-phase TMR.**”

In line with our original manuscript and the reviewer’s comment, we certainly agree that providing potential alternative interpretations of the behavioral results is important. The revised manuscript includes the possibility that down-stimulation induced disruption of performance (as suggested by the reviewer), as well as the possibility of a general, non-specific, acoustic-stimulation-induced improvement in performance on the non-reactivated sequence (see below for exemplar text):

- Page 21: “...one could therefore argue that a more plausible interpretation of the data is that down-stimulation actively disrupted the motor memory consolidation process...”
- Page 21: “...it is possible that, due to our within-subject design, overall acoustic stimulation during post-learning sleep might also have boosted performance on the control (not-reactivated) condition...”

Point #2: Do I understand correctly that EEG results (i.e., ERPs and TFRs) are based on all SOs detected during the simulation periods and not necessarily SOs that triggered TMR (in case of up- and down-state TMR)? If that is the case, please re-run these analyses taking only relevant SOs (TMR targeted) into account. In case that only relevant SOs were taken into account, how do the authors interpret the finding that differences in spindle power between up- and down-state TMR (Figure 3a) already emerged before the time window for up-state stimulation? In addition, while locking the data to SO troughs is a clever idea, please also add a figure comprising the same analyses for trigger-locked data (when comparing up- and down-state TMR). This would allow easier comparison with earlier studies (e.g., Ngo et al., 2022). Furthermore, these results would indicate whether up- and down-state TMR would differ in their capacity to entrain SOs.

Author Response: We thank the reviewer for these interesting remarks.

With respect to the first comment of the reviewer regarding SO included in the analyses for the up and down TMR conditions, we indeed only included relevant / TMR targeted SOs, i.e., SOs that received stimulation at the targeted phase and referred to as true positive stimulations in the manuscript. This is now clarified in the revised manuscript as follows:

- Page 31: “**For the trough-locked analyses, the trough time-sample of each stimulated SO classified as true positive based on the *offline validation algorithm* described above (but see supplemental Figure S10 for analyses based on all events irrespective of accuracy) was extracted from both the up- and down-stimulation intervals.**”

Also note that in response to point #3 below as well as point #4 of reviewer #2, we also now provide in the supplemental information the analyses including all stimulated SOs, irrespective of stimulation accuracy (note that the pattern of results remained similar as in the main analyses, see Figure S10 in the supplemental information and detailed response to point #3 below).

The reviewer makes an interesting second comment related to the onset of the difference in spindle power between up and down conditions (from 0.25 to 0.4 sec relative to the trough onset) that emerges before the onset of the up stimulation (average onset up stimulation of 0.53 sec [95% CI: 0.51 – 0.55]). These results might reflect the impact of our repetitive stimulation design on oscillatory brain activity. Specifically, auditory stimulations were delivered in the current study using a blocked design with 3-min long intervals during which each stimulation type was sent repetitively (e.g., 3 consecutive minutes of up-stimulation followed by minimum 1 minute of no stimulation and 3 consecutive minutes of down-stimulation, etc.). Moreover, our algorithm only enforced a silent (no stimulation) period until the next zero-crossing (to avoid stimulating the same SO multiple times) which potentially allowed for the stimulation of consecutive SOs. Accordingly, 90.83 % [95CI: 87.13 – 94.53] of the auditory cues were sent at a frequency that corresponds to

the SO frequency range (i.e., 0.1 to 4.5 Hz). Based on earlier work showing oscillatory entrainment by rhythmic acoustic stimulation (Haegens & Zion Golumbic, 2018; Lakatos et al., 2019), we suggest that our stimulation design particularly favored the entrainment of ongoing sleep oscillations resulting, with repetition, in power modulation prior to cue onset. Specifically, we propose that the phase-specific modulation of induced responses observed prior to cue onset reflects the effect of CL-TMR on ongoing top-down (Chen et al., 2012; David et al., 2006; Tallon-Baudry & Bertrand, 1999) (here cortico-hippocampal (Staresina, 2024)) processes. This is in line with research in the attentional domain showing that top-down control of attentional processes are reflected by amplitude modulation of gamma and alpha oscillations preceding the appearance of the target (Bonfond et al., 2017; Haegens et al., 2011; Jensen & Mazaheri, 2010; Rihs et al., 2009; Tallon-Baudry, 2003; Thut, 2006). Altogether, our data suggest that SO-up-TMR (vs. SO-down-TMR) reinforced ongoing SO-sigma coupling and the associated spindle-mediated top-down mechanisms facilitating memory reactivation (Staresina, 2024); which in turn enhanced memory performance. We thank the reviewer for giving us the opportunity to reflect on these considerations that have been added to the discussion of the revised manuscript:

- Page 21-22: “**Interestingly, our results also indicate that these up-stimulation-induced modulations in sigma power emerged prior to auditory cue onset (i.e., prior to SO up-state). This result might be related to our repetitive stimulation paradigm which potentially allowed for the stimulation of consecutive SOs. Based on earlier work showing oscillatory entrainment by rhythmic acoustic stimulation ^{36,37}, we suggest that our stimulation design particularly favored the entrainment of ongoing sleep oscillations resulting, with repetition, in power modulation prior to cue onset. Specifically, we propose that the phase-specific modulation of induced responses observed prior to cue onset reflects the effect of CL-TMR on ongoing top-down ³⁸⁻⁴⁰ (here cortico-hippocampal ⁴¹) processes. Further, we propose that the reinforcement of ongoing SO-sigma coupling favored spindle-mediated top-down mechanisms supporting memory reactivation and in turn enhanced memory performance ⁴¹.**”

With respect to the third comment, we are grateful to the reviewer for their positive comment about our SO-trough locking analysis strategy and we fully agree that being able to compare our findings to earlier studies is important. Therefore, we performed Event-Related Potentials (ERP) analyses, around the auditory cues sent at the up and down phases of the SO with procedures similar to earlier research (Göldi et al., 2019; Ngo & Staresina, 2017). As in previous reports (Ngo & Staresina, 2022) and to also address point #1 of reviewer #3, sham “cues” were also created in the not-stimulated intervals. This then allows us to compute ERPs on the control / not stimulated condition for comparison purposes. Specifically, for SOs detected during the no-stimulation intervals, the time-sample of each sham-cue used for the ERP analyses was matched to the phase of one true up or down cue event. To do so, for each SO detected during the no-stimulation interval, a stimulated phase was randomly picked from the list of true cue phases extracted from up and down stimulation intervals and was applied to the non-stimulated SO. The ERPs were then obtained by segmenting the data into epochs time-locked to the (true or sham) cues from -2.5 to 2 sec and averaged across all trials in each condition separately (up-stimulated, up-sham, down-stimulated and down-sham). To analyze oscillatory activity, we computed Time-Frequency Representations (TFR) of the power spectra per experimental condition and per channel. To this

end, we used an adaptive sliding time window of five cycles length per frequency ($\Delta t = 5/f$; 20-ms step size), and estimated power using the Hanning taper/FFT approach between 5 and 30 Hz. Individual TFRs were converted into change of power relative to the entire period around the true or sham cue (from -2.5 to -2 sec relative to cue). Note that statistical analyses were performed on a more conservative -1.5s to 1.5s window relative to cue to avoid border effects. We performed nonparametric cluster-based permutation tests using paired t-test between conditions and cluster-based correction to account for multiple comparisons across time and space for the ERP analyses, and time, frequency and space for the TFR analyses. The cluster threshold was set at $\alpha = 0.01$ whereas the alpha p-value for each candidate cluster was set at $\alpha = 0.05$. Corrections for two-tailed and three comparisons were conducted with Bonferonni procedure within each family of hypothesis tests, i.e., $p < 0.0083$. To address the point of the reviewer, as well as point #4 of reviewer #2 and point #1 of reviewer #3, we compared, for both ERP and TFR analyses, up-sham vs. up-stimulation, down-sham vs. down-stimulation, and up-stimulation vs. down-stimulation conditions.

ERP analyses: The results of the ERP analyses are presented in Figure R1a below and show that up-stimulation, as compared to up-sham, did *not* elicit an increase in response amplitude *during cue presentation* but was followed by a deflection of greater amplitude (significant spatio-temporal clusters are shown with the horizontal magenta lines in Figure R1a). This is partially in line with earlier observations, as Ngo and Staresina (Ngo & Staresina, 2022) reported a deeper deflection after cue offset that was similar to that observed in the current study. However, they also reported a prolonged up-state during cue presentation that we did not observe. The discrepancy in results during cue presentation might be related to the difference in sound duration between the two studies. The current study used a much shorter auditory cue (duration 100ms as compared to 500ms in Ngo and colleagues) that might have induced less pronounced effect *during* sound exposure. Interestingly, we report greater amplitude modulation for the up-stimulated SO (as compared to sham) **before cue-onset** while no such effect was reported in Ngo and Staresina (Ngo & Staresina, 2022). Although it is difficult to compare results between studies because no statistical analyses were performed prior to cue-onset in (Ngo & Staresina, 2022), this earlier work shows less pronounced modulations of SO amplitude during this window. As discussed above, we suggest that this discrepancy may be attributed to the difference in stimulation designs. Specifically, in this earlier research, four different stimulation conditions were used (i.e., up-associated, up-unassociated, down-associated and down-unassociated) and the corresponding sounds were presented randomly with an 8 sec. pause between each stimulation. In contrast, stimulation in our research was performed using a blocked design that allowed the stimulation of consecutive SOs [see above, 90.83 % [95CI: 87.13 – 94.53] of the auditory cues were sent at a frequency that corresponds to the SO frequency range (i.e., 0.1 to 4.5 Hz)] in a sustained manner during 3-min epoch for each stimulation condition. Similar to the pre-cue power modulation discussed above, the cue-locked ERP results suggest that our stimulation design particularly favored the entrainment of ongoing sleep oscillations. Interestingly, while the 500-ms acoustic stimulation in (Ngo & Staresina, 2022) consistently elicited a second SO cycle after cue-offset, no such effect was observed in our study with a shorter sound duration (100ms). In contrast, entrainment in our study is reflected by the pre-cue modulations discussed above.

The comparison between down-stimulation and down-sham conditions (Figure R1a, significant spatio-temporal clusters are shown with the horizontal blue lines) indicated that stimulation resulted in a decrease in SO trough amplitude during and up to 400ms after cue exposure. Similar to the up condition, significant amplitude modulation was observed pre-cue whereby SO amplitude was greater for down-stimulation as compared to sham which might also reflect oscillatory entrainment as discussed above. Similar as above, while the 500-ms down acoustic stimulation in (Ngo & Staresina, 2022) consistently elicited a second SO cycle after cue-offset, no such effect was observed in our study in which entrainment in the down condition was also reflected by pre-cue amplitude modulations.

Finally, the direct comparison between up- and down-stimulation conditions (significant spatio-temporal clusters are shown with the horizontal black lines in Figure R1a) showed that the amplitude of up-stimulated SOs was significantly higher than down-stimulated SOs from cue presentation to 400 ms after cue offset. These results are in line with findings reported in (Ngo & Staresina, 2022) but the longer sound duration used in this earlier work resulted in the synchronization of up- and down-stimulated SOs into a second SO cycle 400ms after cue onset (time from which amplitude differences were no longer observed between up- and down-stimulated SOs in (Ngo & Staresina, 2022)).

TF analyses: The results of the TF analyses are presented in Figure R1b-d and indicate that power in the sham conditions was greater than in the stimulated conditions (both up- and down-) between 5 and 10 Hz (up comparison: from -0.9 sec to 0.5 sec, strongest at -0.3 sec relative to cue; down comparison: from -0.4 sec to 1.3 sec, strongest at -0.1 sec relative to cue) and between 18 and 22 Hz (up comparison: 0.6 sec post-cue onset; down comparison: 1.2 sec post-cue onset; Figure R1c but see also R1b for TFRs in each condition). Note that in both conditions, these power modulations are in phase with the SO negative peak. We invite the reviewer to read our response to their point #5 for an interpretation of sigma power modulation. As expected, when directly comparing the two stimulation conditions (see Figure R1d), we observed that the power modulations in the 5-10 Hz band and in the sigma band alternated with the phase of the SOs (see details in caption of Figure R1). As up- and down-stimulated SOs in the current study are never in phase (in contrast to (Ngo & Staresina, 2022) where conditions synchronize 400ms after cue onset into a second SO cycle), the comparison of cue-locked TFRs between conditions brings limited insight into power modulation in our study (beyond phase-related differences shown in Figure R1d). Instead, when phase-related modulations are controlled for (see trough-locked analyses presented in the main text), our results indicate a difference in sigma power between up and down conditions similar to the one reported in (Ngo & Staresina, 2022) on the second (synchronized) SO cycle elicited by the 500-ms acoustic stimulation.

These analyses are now reported in the revised submission. However, as the results of the cue-locked analyses are highly confounded by the phase of SO in the current study, we opted to keep SO-trough-locked analyses as the main analyses in the revised manuscript and report the results of the cue-locked analyses (as well as interpretations and comparison with previous literature) in the supplements (see section 1 of the supplemental information and Figure S1). These results are called on page 8 of the main text and the corresponding methods are described on page 31.

a. Cue locked event related potentials

b. Time-frequency representations of power modulation in the up- and down-stimulation and sham conditions

c. Difference in power modulation between the stimulated and sham

d. Difference in power modulation between up- and down-stimulation

Figure R1: Results of the cue-locked electrophysiological analyses. a. Event-related potentials. ERP grand-average illustrated at Fz (magenta (yellow): true (sham) stimulation in the up condition; blue (cyan): true (sham) stimulation in the down condition). Horizontal lines represent the adjacent time points of the significant spatio-temporal clusters showing a difference in ERP amplitude between conditions. The solid vertical line represents the cue onset and the dashed line represents the cue offset (100ms sound duration). (i) The up-stimulation vs. up-sham contrast (magenta) revealed four clusters whereby SO amplitude was greater in the stimulation as compared to the sham condition. Two clusters showed greater negative ERP amplitude in the stimulation as compared to sham condition (from -1.46 to -0.94 sec and from 0.41 to 0.73 sec relative to cue onset) and two clusters showed greater positive ERP amplitude in the stimulation as compared to the sham condition (from -0.86 to -0.4 sec and from -0.19 to -0.02sec sec); (ii) The down-

stimulation vs. down-sham contrast (blue) revealed three significant clusters pre-cue onset whereby SO amplitude was greater in the stimulation as compared to the sham condition (greater positive amplitude from -0.27 to -0.05 sec and from -1.31 to -0.81 sec; greater negative amplitude from -0.73 to -0.32 sec). An additional cluster was observed at cue onset whereby the amplitude of the negative peak was lower in the stimulation as compared to sham condition (from 0.02 to 0.46 sec); The (iii) up-stimulation vs. down-stimulation contrast (black) revealed four significant clusters in which SO amplitude differed between conditions according to the phase difference between conditions (from -1.5 to -1.08 sec, -1.01 to -0.48 sec, -0.45 to 0.06 sec and -0.04 to 0.44 sec). **b-d. Time-frequency representations.** TFRs of the power modulation (as compared to baseline) illustrated at Cz on which the grand-average of the SOs at Fz is super-imposed (same color code as in a). **b. Power modulation within each condition.** **c. Difference in power modulations between stimulation and sham conditions.** TFRs of the difference in power modulation for the three contrasts of interest. The area highlighted in the TFR represents the adjacent time-frequency points of the significant spatio-temporal-frequency cluster showing a difference in power between conditions. The results indicate that power in the sham conditions was greater than in the stimulation condition for both up and down conditions between 5 and 10 Hz (up comparison: from -0.95 to 0.3 sec - strongest at -0.25 sec - relative to cue; down comparison: from -1.5 to -0.68 sec, from -0.42 to 0.4 sec - strongest at 0.1 sec -, and from 0.55 to 0.85 sec relative to cue onset) and between 18 and 22 Hz in the up condition (from -1.28 to -1.24 sec and from 0.54 to 0.65 sec relative to cue onset). These power modulations are in phase with the SO negative peak across conditions. **d. Difference in power modulations between up- and down-stimulation.** Results show that power modulations in the 5-10 Hz band and in the 12-18 Hz band alternate with the phase of the SOs. All presented clusters are significant after permutations and after correction for multiple comparisons (cluster p-values < 0.0083).

Point #3: One systematic difference between the up- and the down-state TMR conditions is that according to table S5 the algorithm accuracy for down-state TMR (74.9 %) is lower as compared to up-state TMR (89.5 %). Hence due to the implemented 'online validation algorithm', which was administered to match the number of accurately stimulated SO between conditions, the overall number of down-state TMR cues was significantly higher (791 vs 700). Does the number of cue presentations correlate with memory performance? In addition, please show ERPs and TFRs not only for 'true positive' events but also when incorporating all events.

Author Response: We performed the correlation analyses suggested by the reviewer and there was no correlation between the offline changes in performance and the total number of cues in any of the conditions in the present study (up-stimulation: $r(25) = 0.014$, $p = 0.55$; down-stimulation: $r(25) = 0.00018$, $p = 0.95$). Note that similar results were observed for the correlation between offline changes in performance and the number of true positive cues only (up-stimulation: $r(25) = 0.014$, $p = 0.55$; down-stimulation: $r(25) = 0.00017$, $p = 0.95$). This is in line with an earlier TMR meta-analysis (Hu et al., 2020) and with our own previous work (Nicolas et al., 2022, 2023) indicating a general trend in TMR studies for an absence of relationship between the number of reactivation cues sent during sleep and subsequent memory performance. These results are now presented in the supplemental information and mentioned in the main text as follows:

- Page 28: "Offline validation analyses showed that stimulation accuracy was 82.2% [95CI: 79.6 – 84.8] across stimulation conditions (see Supplemental Table S5) and the mean stimulation phase was 93.39° [95CI: 93.89 - 92.89] for the up condition and 286.51° [95CI:286.13 - 286.89] for the down condition (Figure 1c). The number of true SOs stimulated was not different between conditions (up-stimulated = 622.9 [95CI: 495.2 – 750.7], down-stimulated = 600.5 [483.3 - 717.7], $t = 1.65$, $df = 29$, p -value = 0.11, Cohen's $d = 0.3$). Nonetheless, **exploratory correlation analyses were performed to test whether**

there was a relationship between the number of cues (total number of cues and number of true positive cues) and offline changes in performance in each condition. Results are reported in the supplements and did not show any significant correlation (see section 11 in supplemental information)."

As suggested by the reviewer, we also ran the primary (i.e., trough-locked) electrophysiological analyses including all events, i.e., all the stimulations presented during NREM2-3 sleep regardless of whether they were true positive or not. These analyses therefore also included the cues that fell outside the 180° range around 90° for the up-stimulation and around 270° for the down-stimulation (see definition of true positive in the main manuscript). Results remained largely unchanged, albeit less robust for the comparison between up and down conditions (i.e., differences were significant after permutation but not after correction for multiple comparisons, see caption of Figure R2 for a detailed description). Considering this similarity and because these additional analyses represent a deviation from our pre-registered procedures (*"Only correct stimulation (i.e. auditory cues sent during true SW as defined by Carrier et al. (2011) and refined later in Rosinvil et al. (2020) will be considered for the analyses"*), we opted to keep the original analysis in the revised manuscript. However, we report the results of these analyses in the supplemental information, section 13, Figure S10 (and pasted below for convenience).

Figure R2: Electrophysiological results of the analyses including only the true positive stimulation (left column and Figure 3 of the main manuscript) and including all the NREM2-3 sleep stimulations (right column). a. Up- vs down-stimulated contrasts. Time-frequency representations (TFR) of the difference in power modulation illustrated at Cz around the trough of the up- and down-stimulated SOs on which the grand-average of the SOs illustrated on Fz is super-imposed (magenta(blue): up(down)-stimulated SO). Horizontal lines represent the adjacent time points of the significant spatio-temporal clusters showing a difference in SO amplitude between the two trough-locked ERPs (horizontal grey lines show cluster significant after permutations [cluster p -value < 0.05] and horizontal black lines after permutation and

correction for multiple comparisons [cluster p -value < 0.0083]). Results show that the up-stimulated SO presented greater amplitude at their peak (**left**: 0.33 to 0.64 sec post-trough; **right**: 0.47 to 0.62 sec post-trough) followed by a deeper deflection (**left**: from 0.78 to 1.03 sec post-trough; **right**: from 0.78 to 1.02 sec post-trough). Further, the area highlighted in the TFR represents the adjacent time-frequency points of the significant spatio-temporal-frequency cluster showing a difference in power between conditions (grey contour shows cluster significant after permutations [cluster p -value < 0.05] and black contour after permutation and correction for multiple comparisons [cluster p -value < 0.0083]). Sigma power nested in the ascending phase of the up-stimulated SOs was greater than for the down-stimulated SOs (**left**: from 0.25 to 0.4 sec post-trough and from 12 to 17 Hz; **right**: from 0.25 to 0.37 sec post-trough and from 13.5 to 19 Hz). **b. Up- vs not-stimulated contrasts.** Same as panel a but for the up- and the not-stimulated trough-locked SO potential (magenta and green, respectively) and power modulation. Results show that the amplitude of the up-stimulated SOs was greater than the not-stimulated SOs from -0.61 to 0.61 sec (**left**) / from -0.53 to 0.60 sec (**right**) while it reversed from 0.72 to 1.14 sec (**left**) / from 0.72 to 1.14 sec (**right**) relative to the trough onset. Power in the 5-17.5 Hz (**left**) / 5-19 Hz (**right**) frequency range was lower in the up- as compared to the not-stimulated condition in the descending phase of the SOs (**left**: from -0.45 to 0.24 sec relative to the SO trough; **right**: from -0.48 to 0.27 sec relative to the SO trough). **c. Down- vs not-stimulated contrasts.** Same as panel a but for the down- and the not-stimulated trough-locked SO potential (blue and green, respectively) and power modulation. Results show that the amplitude of the down-stimulated SOs was greater than the not-stimulated SOs from -0.60 to 0.42 sec (**left**) / from -0.60 to 0.44 sec (**right**) while it reversed from 1.16 to 1.50 sec (**left**) / (**right**) relative to the trough onset. Power was lower in the 5-18 Hz (**left**) / 5-19 Hz and 17-22 Hz (**right**) frequency range in the down- as compared to the not-stimulated condition during the descending phase of SOs (**left**: from -0.49 to 0.24 sec relative to the SO trough; **right**: from -0.52 to 0.27 sec relative to the SO trough) and greater in the 5-10 Hz from 0.76 to 1.50 sec (**left**) / 5-12 Hz from 0.70 to 1.50 sec (**right**) relative to trough onset.

Point #4: Somehow related to above. Why was the threshold for down-state detection set to around -40 μ V? As visible in Fig. 3 down-states peaked around -80 μ V. Would this threshold explain why the accuracy of targeting down-states was rather modest?

Author Response: The down-state detection at -40 μ V was chosen as this is the most commonly used threshold for SO detection across different laboratories and algorithms (Carrier et al., 2011; Rosinvil et al., 2020; Vallat & Walker, 2021). The modest accuracy for down-state detection is unlikely to be related to this amplitude criteria but is rather due to the nature of the online detection approach. As in previous closed-loop stimulation paradigms (Göldi et al., 2019; Ngo & Staresina, 2022), down-state detection was based on an amplitude criterion, as additional criteria that could increase the accuracy of detection (and used for example for up-state detection) come after the down-state (e.g., amplitude of the following positive peak) and can therefore not be used for the online detection of down-states. For example, down-state stimulation was sent every time the amplitude of the signal dropped below -40 μ V while up-stimulation was sent based on this criterion **and** a minimum peak-to-peak amplitude of 75 μ V following the negative peak. Down stimulation is therefore usually quite lenient - and significantly more than up stimulation - and results in lower accuracy. Importantly, the -40 μ V detection threshold cannot explain the difference in accuracy between the up- and the down-stimulation as the same threshold was used to detect both phases. Instead, as mentioned above, the difference in accuracy is due to a difference in the number of criteria to meet for detection between the two conditions.

Point #5: It seems surprising that spindle density was lower during both up- and down-stimulated as compared to not-stimulated blocks. I would be curious how the authors interpret this finding.

Relatedly spindle densities seem very low in general (Fig. S2b). Could it be that the applied algorithm is too conservative?

Author Response: We thank the reviewer for pointing this out. Spindles characteristics can indeed fluctuate between studies depending on the parameters used for detection. In the current study, we used detection parameters provided by default in YASA (Vallat & Walker, 2021) and these parameters were indeed more conservative than in other studies. Specifically, the spindle frequency range was set between 12-16 Hz and the minimum and maximum spindle durations set to 0.5 and 3 sec, respectively. Detection with these rather conservative criteria indeed resulted in lower spindle density compared to earlier reports (i.e., around 0.5 spindles / min of analyzed NREM2-3 in our study). For example, Purcell et al. (Purcell et al., 2017) - who characterized sleep spindles in 11,630 individuals with more lenient parameters - report an average density of 2.19 spindles per minute of NREM2 and 1.45 spindles per minute of NREM3 (which is the sleep stage that was mainly stimulated in our study). To address the reviewer's comment and examine the influence of these parameters on spindle characteristics, we ran the spindle detection analysis with the parameters used in Purcell et al. (Purcell et al., 2017), i.e. a broader frequency range set between 8-18 Hz and a broader time window set between 0.3 and 3 sec. As expected, with these more lenient parameters, the mean spindle density increased in our sample to a similar density as reported in Purcell and colleagues (Purcell et al., 2017) (i.e., 2.67 [95CI: 2.31 – 3.03] spindles per minute of NREM2-3).

Importantly, the effect of stimulation on spindle characteristics remained unchanged using this more lenient approach. Specifically, as in our original analyses, we observed that spindle frequency and amplitude were unaffected by the stimulation while spindle density was lower during both up- and down-stimulated as compared to not-stimulated intervals. It therefore appears unlikely that these effects were driven by the more conservative approach used in the current study. To allow for a more direct comparison between our work and earlier studies, we elected to report the results of the spindle analyses using the more widely used parameters described in Purcell et al. instead of the default parameters of YASA. The corresponding changes can be found in the revised methods page 32 (Sleep events detection section) and updated results are presented in the Supplemental Figure S3.

The current results showing lower spindle density in both stimulation conditions as compared to the non-stimulation condition are indeed contradictory with earlier studies showing that sound presentation during sleep increases spindle activity (e.g., Antony et al., 2012; Cairney et al., 2018). However, they are in line with evidence from the animal literature showing that sounds can disrupt spindle activity (Aksamaz et al., 2024; Britvina & Eggermont, 2008). The invasive recordings in these animal studies suggest that auditory stimulation during sleep can result in specific inter-regional sleep oscillation coupling modulations that can, in some cases, induce spindle activity disruption. The present findings are also in line with our earlier research (Nicolas et al., 2022) showing a similar stimulation-induced decrease in spindle *frequency*. More importantly, our earlier work (Nicolas et al., 2022) and the present data highlight the importance of performing fine-grained examination of spindle activity modulation in relation to the phase of the SO, as these modulations might

support different mnemonic processes. Specifically, our previous research showed that while TMR resulted in an **increase** in sigma power in the *ascending* phase of the SO (that was related to subsequent motor performance enhancement), it induced a significant **decrease** in sigma power in the descending phase of the SO as compared to control stimulation. It was proposed that sigma oscillations may play different roles in memory reactivation and memory protection during sleep depending on their precise temporal coordination with slow oscillations (up- vs. down-phase, respectively) (Nicolas et al., 2022). The current results are in line with these previous observations as up- (as compared to down-) stimulation resulted in an **increase** in sigma power at the peak of the SO which is thought to support memory reinstatement processes (Nicolas et al., 2022). In contrast, general auditory stimulation (irrespective of the condition and as compared to no stimulation) induced a **decrease** in sigma power at the trough of the SO as well as a decrease in spindle density which might reflect a decrease in gating for sounds to which a memory trace is associated (Nicolas et al., 2022). As the nature of these reflections is rather speculative, we decided to not add these considerations to the revised discussion.

Point #6: Please integrate the circular plot indicating the stimulation phase (Fig. S7) in Fig. 3. Also show phase angles across all trials (for up- and down-state cues, respectively).

Author Response: We thank the reviewer for this suggestion. The circular plot showing phase angles across all trials (irrespective of their accuracy) and across all participants has been added to the revised manuscript. However, we opted to present this panel in Figure 1 instead of Figure 3 so that the information about the accuracy of the detection algorithm is provided earlier in the manuscript and is paired with the methods and design considerations already contained within Figure 1.

Point #7: The authors report that “... the overnight increase in striatal activity was greater for the up-reactivated sequence as compared to the down-reactivated sequence, which in turn was greater than for the not-reactivated sequence (Figure 4b) ...”. Was this effect restricted to the right caudate or also evident in the putamen and M1? Please add the right caudate to Fig. 4a (overnight increase in activity within condition) to allow for excluding that condition specific differences in overnight activity increases were driven by baseline effects.

Author Response: The gradient in overnight increase in activity described above was indeed restricted to the caudate nucleus in which significant differences were observed among the three different conditions (see Supplemental Table S2-1). In the putamen, the overnight increase in activity was only significantly different between the *up* and *down* conditions; and in M1, the difference was only observed between *down* and *not* conditions. We thank the reviewer for the suggestion to display the parameter estimates of the caudate in Figure 4a. We have now included the violin plots representing BOLD responses extracted from clusters overlapping between conditions in the right caudate. As shown in the updated Figure 4a of the main manuscript (and pasted below for convenience as Figure R3), the effects described above are not driven by baseline differences in caudate activity.

a. Overnight increase in task-related brain activity within conditions

Figure R3: Phase-specific modulation of overnight increase in activity within condition. Brain activity increased overnight in a set of cortico-striatal regions for up- (right putamen: $x = 24, y = 18, z = -10, pSVC < 0.001$; right caudate: $x = 10, y = 4, z = 12, pSVC < 0.001$; left M1: $x = -38, y = -20, z = 52, pSVC < 0.001$), down- (right putamen: $x = 26, y = -8, z = -4, pSVC = 0.001$; right caudate: $x = 10, y = 4, z = 16, pSVC = 0.004$; left M1: $x = -38, y = -24, z = 54, pSVC < 0.001$), and not-reactivated sequences (right putamen: $x = 28, y = -12, z = -6, pSVC = 0.002$; right caudate: $x = 10, y = 6, z = 12, pSVC < 0.001$; left M1: $x = -38, y = -26, z = 56, pSVC = 0.004$). Violin plots represent BOLD responses extracted from clusters overlapping between conditions (1. Right putamen: $x = 26, y = -8, z = -4$; 2. Right caudate: $x = 10, y = 0, z = 14$; 3. Left M1: $x = -38, y = -20, z = 52$).

Point #8: For correlational analyses with behavior (e.g., Fig. 4c and 5c) the authors computed a TMR index which consisted of the difference in offline gains in performance between up-reactivated and not-reactivated sequences, as well as down-reactivated and not-reactivated sequences. Is this metric based on accuracies? As shown in Fig. S5 TMR did not have any effect on accuracies, which makes me wonder why this would be a useful proxy for TMR success.

Author Response: We apologize that this information was not clear in the original manuscript. The TMR index was indeed based on the performance *speed* but not the performance *accuracy* measure. As the offline changes in performance *speed* - from which the TMR index was derived - were effectively modulated by the experimental conditions (up vs. down vs. not), the TMR index based on performance *speed* is indeed considered a useful proxy for TMR success. This information was clarified throughout the manuscript. For example:

- **Page 12:** “Importantly, the between-session increase in striato-motor activity reported above was correlated with the TMR index (i.e., the difference in offline changes in performance **speed** between the reactivated vs. the not-reactivated sequences) for both the up- and down-reactivated sequences (Figure 4c; see Table S2-3).”
- **Page 29:** “Additionally, we computed a TMR index which consisted of the difference in offline gains in performance **speed** between up-reactivated and not-reactivated sequences (TMR index_{up}) as well as down-reactivated and not-reactivated sequences (TMR index_{down}), separately.”

We also clarified that the offline changes in performance measures from which the TMR index was derived were also based on performance *speed*. For example:

- **Page 7:** “the offline changes in performance **speed**”

- Page 23: “In summary, this study shows a phase-specific beneficial effect of slow-oscillation closed-loop TMR on consolidation such that motor memories reactivated during the up-phase of slow oscillations exhibited greater performance **speed** compared to memories reactivated at the down-phase.”

Minor comments:

Point #9: Please include the proportion of female/male among the N=31

Author Response: We thank the reviewer for pointing out this omission. Fifteen females were recruited for the study. We now have included this information in the main manuscript and in Table S5 of the supplemental information.

- Page 3: “Briefly, in a within-subject design, 31 young healthy participants (**15 females**) ...”

Point #10: How were stimulation phases (Figure S7) computed?

Author Response: The stimulation phases were extracted using the EndPoint Corrected Hilbert Transform (ECHT) from the stimulation device. ECHT allows to instantaneously compute the phase of a discrete analytic signal (here the filtered signal between 0.1-4.5 Hz) by correcting for the Gibbs phenomenon distortion during the discrete Fourier transform of the signal (Schreglmann et al., 2021). We thank the reviewer for pointing out this omission and added this information in the revised manuscript as follows:

- Page 27: “The CL-TMR device (Elemind Technologies), that was used for the online detection and stimulation of the SOs (see detection methods below, required another set of electrodes for which the signal was recorded from FPz at a sampling rate of 500 Hz on average (ground and reference electrodes placed behind the right ear). **This device has been developed to instantaneously compute the phase of the filtered signal (0.1 - 4.5 Hz) with the EndPoint Corrected Hilbert Transform (which allows to correct for the Gibbs phenomenon during the discrete Fourier transform of the signal ⁷⁰).**”

Point #11: Did the sounds (low-frequency tone, white noise, high-frequency tone) differ in their capacity to impact sleep oscillations and memory performance?

Author Response: We thank the reviewer for raising this important point. To address it, we first tested whether the nature of the sound influenced memory performance, and we ran the same rmANOVA initially conducted for our main behavioural analysis on offline performance changes but with Sound type (3: low-frequency tone vs. white noise vs. high-frequency tone) instead of Condition (3: up- vs. down- vs. not-reactivated) as within-subject factor. We found no significant main effect of the Sound type factor on offline changes in performance ($F(2,54) = 30.82$, $p = 0.48$ (0.43 sphericity corrected), $\eta^2 = 0.029$; Figure R4).

Figure R4: Offline changes in performance speed as a function of the sound attributed to the sequence. (% change between the average of the three blocks of pre-night test and the first three blocks of post-night training) averaged across participants for the low pitch (red), the white noise (green), and the high pitch (blue) sounds. Violin plots: median (horizontal bar), mean (diamond), the shape of the violin plots depicts the kernel density estimate of the data. Black points represent individual data, jittered in arbitrary distances on the x-axis within the respective violin plot to increase perceptibility. For each individual, performance on the different sound conditions are connected with a line between violin plots. n.s.: non-significant.

Second, we tested whether the nature of the sound influenced sleep oscillations, and we again ran the same analysis as the main EEG analysis but with Sound type (3: low-frequency tone vs. white noise vs. high-frequency tone) as within-subject factor. We did not find any significant positive or negative clusters neither in the SO amplitude nor in the oscillatory analyses comparing the three different sounds (all p-values > 0.070).

These results are now included in the supplemental information (Section 11, Figure S8) and mentioned in the Materials and Methods of the main manuscript (copied below for completeness).

- Page 30: “We describe in the supplemental information the control analyses that were collectively performed to verify that the pattern of behavioral results emerged from our experimental manipulation on motor memory consolidation processes rather than from various potential confounding factors listed below (see section 11 in the supplements). Fourth, we tested whether the three different sounds differently impacted motor memory performance. To do so, a one-way rmANOVA was performed on the offline changes in performance speed with the 3-level factor sound type (low-frequency tone vs. white noise vs. high-frequency tone) as within-subject factor (see Figure S8 in the supplements).”
- Page 32: “Additional control analyses were performed to test whether the nature of the different sounds influenced sleep oscillations. To do so, we analyzed the EEG data with same procedure as described above for SO-trough-locked analyses but using sounds (low-frequency tone vs. white noise vs. high-frequency tone) instead of condition as within-subject factor. Corresponding results are reported in section 11 of the supplemental information and show that the nature of the sound did not influence the pattern of EEG results presented in the main text.”

Point #12: Please include line numbers

Author Response: Line numbers are now included.

REVIEWER #2

General comments: The authors report a study on the phase specific effects of closed-loop targeted memory reactivation during sleep on a serial reaction time task. The authors' report that cueing during the up-phase of SOs leads to greater performance improvement compared to cueing during the down-phase. Up-stimulated SOs also had higher amplitude than down- or non-stimulated SOs and greater sigma power than down-stimulated SOs. fMRI analyses within the different stimulation conditions showed a variety of correlations between behavior and changes in BOLD responses both locally in the striatum and the hippocampus and in connectivity between the hippocampus, striatum, and motor cortex. Overall, this is an interesting study looking at phase-specific TMR and the authors' efforts in bringing closed-loop TMR to motor memory domain is appreciated. However, there are concerns with the analysis methods, results curation, and interpretation of results.

While there is novelty in describing changes in the motor memory related functional networks before and after sleep and their relation to sleep oscillations and motor performance, this study offers limited conceptual advances or solid empirical contributions to the field. There are major methodological and data-related limitations that challenge the conclusions drawn in the study. The concluding claim that "this study shows a phase-specific beneficial effect of slow-oscillation closed-loop TMR on consolidation such that motor memories reactivated during the up-phase of slow oscillations exhibited greater performance....." is unwarranted as the data, as collected and presented, do not adequately support these conclusions.

Author Response: We thank the reviewer for the time devoted to our manuscript. The reviewer raised interesting points that we have addressed on a point-by-point basis below. Several comments that relate to the empirical contribution of our work are linked to a lack of clarity about certain aspects of our research approach (e.g., behavioral and EEG data curation, correction for multiple comparisons for the MRI data). We made a particular effort to increase the clarity of our procedures to better substantiate that the data adequately support the conclusions of the paper. It is our opinion that the current research offers substantial conceptual advances to the field for the following reasons. First, this study integrates concepts and models of memory consolidation across different memory systems and species. Second, it deepens our understanding of the neurophysiological processes associated with memory reactivation during sleep. Specifically, our results shed light into how the precise temporal coordination of brain oscillations during post-learning sleep is related to memory consolidation processes at both the behavioral and neural levels. Last, our findings highlight the potential of closed-loop targeted memory consolidation to induce phase-specific modulations of motor performance. Accordingly, the results of this research offer new avenues for the future development of procedures designed to speed the recovery of motor function following e.g. brain disease or injury.

We kindly invite the reviewer to read our detailed response below (including responses to the general points made in the remarks above) and we hope our responses and corresponding updates to the manuscript (highlighted in yellow in the revised manuscript) will alleviate the reviewer's concerns.

Point #1: Behavioral analyses. The authors do not report a significant consolidation effect between post-sleep training (trials 1-3) and the test trials for either speed or accuracy across any condition. As the task is not consolidated by participants in the study, it is difficult to draw any conclusions about memory consolidation by analyzing the behavioral data. Previous work has reported sleep-dependent improvements in reaction time in similar SRTT tasks. Other work using TMR with a similar SRTT task also found an improvement in uncued but not cued sequences (e.g. Cousins 2016). This discrepancy is not addressed in the discussion.

Author Response: We are not entirely sure which null effect the reviewer is referencing when they mention that there was no significant consolidation between pre- and post-sleep trials across conditions in the current study. For the purposes of this response, we assume that the reviewer is referring to the absolute changes in performance within each condition and whether they were significantly different from zero. We would first like to clarify that such statistical analyses were not performed in the original manuscript as the amplitude of offline changes in performance is heavily influenced by computational and/or methodological choices (e.g., number of blocks selected, presence/absence of immediate post-training test dissipating fatigue). These issues – largely led by work from the lab of Dr. Tim Rickard - have been discussed at length in previous experimental papers and commentaries (Brawn et al., 2010; Cai & Rickard, 2009, 2009; Rickard et al., 2008; Rickard & Pan, 2017). Accordingly, there is a general consensus in the field that statistically comparing offline performance changes to a test value of zero offers little insight into the motor memory consolidation process. Rather, the comparison of offline changes in performance between conditions and/or groups submitted to the same experimental design provides a more direct and valid assessment (King et al., 2017). Even though examining the absolute changes in performance within conditions / groups comes with substantial limitations for the assessment of offline consolidation processes, we respectfully disagree that ‘maintenance’ in performance between sessions (e.g., no significant changes between pre- to post-sleep sessions / changes not significantly different from zero) reflects a lack of consolidation. Putting aside the methodological considerations mentioned above, one possible outcome of the consolidation process is indeed the maintenance of performance, i.e. the process by which performance (and the associated memory trace) does not go back to the level observed during the first exposure to the task (e.g., during block 1). This has been observed in previous motor sequence learning literature (Albouy, Fogel, et al., 2013; Albouy et al., 2015) and analogous assessments are common with visuomotor adaptation tasks (Albouy, Vandewalle, et al., 2013; Brashers-Krug et al., 1996; Krakauer, 2009; Krakauer et al., 1999, 2005). Indeed, the proxy of consolidation can include a variety of behavioral outcomes such as gains in performance (Albouy et al., 2006, 2008; Debas et al., 2010; Doyon et al., 2009; Maquet et al., 2000; Nettersheim et al., 2015; Walker et al., 2005), performance maintenance (Walker, Brakefield, Seidman, et al., 2003; Walker et al., 2002), or resistance to interference (also called stabilization) (Nettersheim et al., 2015; Walker, Brakefield, Allan Hobson, et al., 2003). The present data suggest that all three conditions (up, down and not) exhibited maintenance overnight and thus were consolidated. More importantly, the magnitude of consolidation differed between conditions. Specifically, behavioral results showed that offline changes in performance were significantly *greater* for the up-, *as compared to* the down-, reactivated sequence, which supports the conclusion that SO-up-phase TMR enhanced performance as compared to SO-down-phase TMR.

Concerning the work of Cousins and colleagues (2016), a discussion - and a comparison between studies - of absolute changes in performance within conditions / groups should be avoided because of the methodological and computational concerns mentioned above. Importantly, Cousins and colleagues reported *greater* performance for the sequence that was reactivated *as compared to* the not-reactivated sequence. These findings and the current results are in line with the general idea that TMR influences motor performance (e.g., Antony et al., 2012; Cousins et al., 2016; Nicolas et al., 2022; Schönauer et al., 2014; and see Hu et al., 2020 for the results of a meta-analysis confirming these effects). Here, we provide novel evidence that these TMR effects differ depending on the SO-phase in which the cues were delivered. As the goal of the current study was to examine whether the phase of the SO at which TMR is administered influences the consolidation process, we focused the discussion of the manuscript on earlier work using closed-loop stimulation paradigms targeting similar SO-phases as in the current study. Accordingly, earlier studies using open-loop TMR paradigms as in Cousins et al. were not discussed in depth in our manuscript.

In summary, our analyses followed current recommendations in the field for a more direct and valid assessment of offline motor memory consolidation processes. The corresponding behavioral results support the main conclusion of the paper, i.e., that SO-up-phase TMR enhanced performance as compared to SO-down-phase TMR.

Point #2: When examining the differences in reaction time changes between testing and post-sleep training, a rm-anova was used to report a significant decrease in the downstate cued sequence vs. the upstate and non-cued sequence. However, when examining the data in figure 2, the medians of the group don't seem to be significantly different, with the mean differences appearing to be driven by a few participants with very large decreases in performance in the down-cued sequence. A non-parametric or robust statistical approach may be warranted with this distribution of data.

Author Response: We verified the following assumptions for the ANOVA performed on offline changes in performance. First, we examined the distribution of the residuals of the ANOVA and performed a Fligner-Killeen test with the null hypothesis that the variance of the residuals is similar between conditions. Below we provide the Q-Q plot (with the quantile-to-quantile band) showing that residuals are normally distributed (Figure R5) with only three residuals outside the range. Furthermore, the Fligner-Killeen test show homoscedasticity of the residuals (chi-squared = 1.85, df = 2, p-value = 0.40) indicating that assumptions for the ANOVA were not violated. Accordingly, we proceeded with the parametric statistical approach.

Figure R5: Q-Q plot of the residuals of the behavioral main analysis model with experimental data quantiles on the y-axis and normal theoretical quantiles on the x-axis.

However, as noted by the reviewer, we acknowledge that there were some individuals in our sample showing more extreme values for the offline changes in performance than the rest of the population (in particular in the down condition). Note that these individuals were not classified as outliers in the current study, as outliers were identified - per our pre-registered procedures - based on initial training performance (i.e., “...magnitude of [initial] learning was more than three standard deviations below the group average”). Nonetheless, to address the reviewer’s comment, we tested whether there were any outliers on the offline change measures (i.e., any individual showing an offline change in performance that was more than three standard deviations away from the offline change averaged across participants and conditions). This procedure revealed that one individual presented outlier offline changes in performance in the down condition. Inspection of the data of this individual did not reveal any problematic behavioral data points (see Figure R6) or any deviation from the group with respect to demographics, characteristics, sleep and vigilance data. Therefore, we believe that there is no justification to deviate from our pre-registered procedures and to exclude this participant from our sample. For the sake of completeness, we nevertheless ran the rMANOVA on the offline changes in performance speed when excluding this more extreme participant and results show a trend towards a significant effect of condition (Condition effect ($F(2,52) = 2.73, p = 0.074$ (0.074 sphericity corrected)) and a similar effect size as reported in the original analyses ($\eta^2 = 0.13$). This information is now added to the caption of Figure 2 in the revised text.

- **Page 8: “Note that the main effect of condition is marginally significant when excluding the participant showing the extreme datapoint in the down condition (Condition effect ($F(2,52) = 2.73, p = 0.074, \eta^2 = 0.13$). This data point was not excluded from the primary analysis as it did not meet any criterion for outlier exclusion as defined in our pre-registration.”**

Figure R6: Behavioral results of participant showing outlier offline changes in performance on the down condition. Participant's median reaction time in ms plotted as a function of blocks of practice during the pre- and post-night for the up-reactivated (magenta circles), the down-reactivated (blue empty circles), and the not-reactivated (green diamonds) sequences and for the random serial reaction time task performed at the start and end of the experiment (black overlay).

Point #3: It would be useful to characterize the artifactual blocks that are removed from analysis more completely. The mean percent removed from each TMR condition should be compared, as well as if the removed blocks differed in the PM-training, testing, or the AM-testing. Additionally, it would be useful to clarify how calculations were performed if the blocks used to calculate the over-sleep changes were removed as artifactual. Based on these considerations, the conclusion 'Our behavioral results indicate that TMR applied at the up-phase of the SO resulted in greater gains in motor performance than when administered at the down-phase of the SO.' (pp. 13, line 383) is not supported.

Author Response: We are unsure what the reviewer means by "artifactual blocks" that were removed from the analyses. It is our opinion that perhaps a lack of clarity with respect to data exclusion in the original manuscript has led to a misunderstanding. We provide more detail here in this response and then corresponding changes have been made to the manuscript.

Practice **blocks** (some of which were indeed used to compute offline changes in performance, see below) comprised 20 key presses (or trials), i.e. four repetitions of the 5-element sequence. No blocks were removed from any of the behavioral analyses. However, based on the pre-registered procedures also outlined in #2 Supplemental Table S4 ("The averaged response times for correct key presses [...] will be computed for each block" and "Individual trials [...] will be excluded from the analyses if they are outlier trials based on John Tukey's method of leveraging the Interquartile"), we removed from the analyses both *incorrect* and *outlier trials* (i.e., individual key responses). As shown in Table R1 below, the percentage of discarded trials was relatively low across sessions and conditions. As such, there were no instances where the number of removed trials was high enough that a full practice block (including those used for the computation of offline changes in performance) had to be discarded. This information is now clarified in the revised manuscript and paste below for convenience.

We agree with the reviewer that it is important to compare the mean percent of removed trials among the three different conditions across the different sessions (pre-night training, pre-night testing and post-night training, see Table R1 below for mean percent). We specifically tested whether the number of excluded trials was different among conditions in blocks of practice used to (1) perform the MRI contrasts (i.e., pre-night training vs. post-night training blocks) and (2) compute offline changes in performance (i.e., pre-night test blocks vs. first 3 blocks of post-night training). The rmANOVA on the percentage of discarded trials (incorrect and outlier) with condition (up- vs. down- vs. not-reactivated) and MRI sessions (pre-night training vs. post-night training) as within-subject factors revealed no significant main effects nor any interaction (Session effect: $F(1,27) = 0.83$, $p = 0.37$, $\eta^2 = 0.0030$; Condition effect: $F(2,54) = 0.37$, $p = 0.70$ (0.69 sphericity corrected), $\eta^2 = 0.013$; session by condition interaction: $F(2,54) = 1.81$, $p = 0.17$ (0.18 sphericity corrected), $\eta^2 = 0.063$). The rmANOVA performed on the percentage of discarded trials with condition and “offline computation” session (pre-night test vs. 3 first blocks of post-night training) as within-subject factors revealed no significant main effects nor any interaction (Session effect: $F(1,27) = 0.097$, $p = 0.76$, $\eta^2 = 0.0036$; Condition effect: $F(2,54) = 0.83$, $p = 0.44$ (0.43 sphericity corrected), $\eta^2 = 0.0030$; session by condition interaction: $F(2,54) = 0.57$, $p = 0.57$ (0.55 sphericity corrected), $\eta^2 = 0.021$). These analyses confirm that the number of analyzed trials did not differ among conditions.

Table R1: Mean percent of discarded trials in the behavioral analysis and 95% confidence interval.

	Pre-night training	Pre-night testing	Post-night (3 first blocks)	Post-night training (all blocks)
Up-reactivated	10.9 [7.9 - 13.9]	9.6 [6.3 - 13]	9.9 [6.7 - 13.1]	9.9 [7.4 - 12.4]
Down-reactivated	10.9 [8.8 - 13.1]	12.6 [8.7 - 16.4]	11 [8.3 - 13.6]	11.4 [9.4 - 13.3]
Not-reactivated	10.5 [8.5 - 12.5]	9.8 [7.7 - 11.8]	10.1 [6.9 - 13.3]	9.5 [7.5 - 11.6]

To increase the clarity of our procedures and to better substantiate that our conclusions are supported by our behavioral analyses, we now report the analyses comparing mean percent of discarded trials described above in Table S6 of section 11 the supplemental information and in the main text as follows:

- **Page 29:** “For each block of practice, motor performance on both the random and sequential SRTT was measured in terms of speed (median response time on correct trials, in ms) and accuracy (% of correct responses, with a trial classified as “correct” if the key pressed by the participants matches the visual cue). Note that correct trials were excluded from the analyses if they were outlier trials based on John Tukey’s method of leveraging the Interquartile Range# (see #2 in supplemental Table S4). **The total number of discarded trials (inaccurate and outlier) was 10.5% across sessions and conditions (see supplemental Table S6 for details). Corresponding analyses are presented in section 11 of the supplemental information and showed that the number of analyzed trials did not differ between conditions.**”

Point #4: Electrophysiological analysis. Different SO's are stimulated in the down-reactivated vs. up-reactivated stimulation conditions because of the additional peak-to-peak amplitude threshold with the up-reactivated. This leads to different stimulation numbers for the up vs. down. We appreciate the authors' addressing this issue with online validation algorithm and offline validation algorithm. However, this difference still leads to issues with understanding the effect of stimulation as in the down-reactivation some actual SOs are not being stimulated and more non-SOs are being stimulated. Removing the 'false-positive' stimulation events when the down-state and peak-to-peak amplitude thresholds were not reached removes a significant number of stimulation events (~ 10% of the up-reactivated cues and ~24% of the down-reactivated cue data based on Table S5). Presumably these stimulation events also affected participant electrophysiology. Not including these in the EEG analysis nor the correlation analysis between BOLD/connectivity changes and EEG measures compromises the interpretability of the stimulation results. Using trough-locked analysis does not allow an evaluation of the acute effects of stimulation. This analysis also makes it difficult to see if, on average, the detected oscillation that is trough-locked is an evoked oscillation with a sound stimulus presented in the 1.5 second pre-trough period. However, due to the stimulation protocol, this might be difficult to address with previous analysis methods in the field.

Author Response: We thank the reviewer for raising these important points. With respect to the suggestion to include false positive cues in the EEG analyses, we kindly invite the reviewer to read our response to point #3 of reviewer #1 for a detailed answer. Briefly, we re-ran all the EEG analyses with all cues delivered during NREM sleep (irrespective of their accuracy) and the corresponding results (presented in the right column of Figure R2 above) are similar to our initial observations including only true positive stimulations (Figure R2, left column); and thus the original conclusions remain largely unchanged. Considering that the results of these analyses are nearly identical to the original ones and because these additional analyses represent a deviation from our pre-registered procedures (" *Only correct stimulation (i.e. auditory cues sent during true SW as defined by Carrier et al. (2011) and refined later in Rosinvil et al. (2020) will be considered for the analyses*"), we opted to keep the original analysis in the revised manuscript. However, we report the results of the analyses including all cues in the supplemental information for completeness (section 13 in the supplements and Figure S10).

Regarding the suggestion of the reviewer to perform cue-locked (in addition to the trough-locked) EEG analyses, we kindly refer the reviewer to our response to point #2 of reviewer #1. In short, we performed Event-Related Potentials (ERP) analyses around the auditory cues sent at the up and down phases of the SO with procedures similar to earlier research (Göldi et al., 2019; Ngo & Staresina, 2017). Further, sham cues were derived from the not-stimulated intervals to compute ERPs on the control / not stimulated condition. Nonparametric cluster-based permutation tests were performed on a -1.5s to 1.5s window relative to cue onset using paired t-test between conditions. Altogether, our results (detailed in the caption of Figure R1 above) partially replicated the results of earlier studies and discrepancies could largely be related to differences in experimental design. Specifically, auditory stimulations in the current study were of shorter duration (100ms vs. 500ms in earlier research (Ngo & Staresina, 2022)) and were delivered using a blocked design with 3-min long intervals during which each stimulation type was sent repetitively which allowed the stimulation of consecutive SOs. The use of a much shorter auditory cue duration in the current study might have induced less pronounced effect *during* sound exposure and did not

allow up- and down-conditions to synchronize in contrast to earlier research (Ngo & Staresina, 2022) where conditions synchronize 400ms after cue onset into a second SO cycle. However, our rhythmic and sustained stimulation paradigm particularly favored the entrainment of ongoing sleep oscillations as shown by the strong pre-cue modulations in amplitude (see Figure R1 and response to point #2 of reviewer #1 for a detailed discussion of this point).

With respect to the time-frequency analyses, it is important to note that as up- and down-stimulated SOs are never in phase in the current study, the comparison of cue-locked time-frequency representations between conditions brings limited insight into power modulation in our study (beyond phase-related differences shown in Figure R1d). Instead, when phase-related modulations are controlled for (see trough-locked analyses presented in the original analyses of the main text), our results indicate a difference in sigma power between up and down conditions similar to the one reported in (Ngo & Staresina, 2022) on the second (synchronized) SO cycle elicited by the 500-ms acoustic stimulation.

These analyses are now reported in the revised manuscript. However, as the results of the cue-locked analyses are highly confounded by the phase of SO in the current study, we opted to keep SO-trough-locked analyses as the main analyses in the revised manuscript and report the results of the cue-locked analyses (as well as interpretations and comparison with previous literature) in the supplements (see section 1 of the supplemental information and Figure S1). These results are called on page 8 of the main text and the corresponding methods are described on page 31.

Point #5: The authors showed that up-stimulated SOs showed higher sigma power compared to down-stimulated SOs, but the spindle density did not differ between the two conditions. It is possible that the increased power detected is a result of heightened arousal/noise. The discussion of spindles, ripples and SOs being important in memory consolidation does not accurately reflect the finding as sigma power does not equal sleep spindles.

Author Response: We thank the reviewer for raising this interesting point with respect to arousal. To examine whether the increased sigma power in the up- as compared to the down-stimulated condition could be a result of heightened arousal, we quantified the number of arousals occurring within a two-second window post-SO-trough (i.e., the window in which sigma power modulation was observed after the SO trough) for each stimulation condition. In total, 2.2 [95CI: 1.3 – 3.2] arousals were observed across all these 2-s intervals for up stimulation and 2.6 [95CI: 1.5 – 3.7] for down stimulation and the number of arousals was not significantly different between conditions ($t = -0.76$, $df = 29$, p -value = 0.46, Cohen's $d = 0.14$). Therefore, it is unlikely that a difference in arousal caused the difference in sigma power between the up- and the down-stimulated SOs. Instead, we suggest that up-stimulation specifically strengthened ongoing SO-sigma coupling and reinforced memory consolidation processes during sleep.

The lack of modulation of *spindle event* characteristics in the current study (as opposed to the significant effect observed on sigma power) is in line with our earlier work¹² and suggests that sigma oscillations might represent a proxy of spindle activity that is more sensitive to memory-related experimental interventions than spindle events per se. This is in line with recent evidence that has brought forward the idea that spindle event detection in general is less sensitive than the

study of the sigma rhythm as a whole (Dimitrov et al., 2021; Nicolas et al., 2022). The increase in up-phase stimulation-induced *sigma power* is in line with earlier CL-TMR research in which these modulations are described to reflect a strengthening of SO-sigma coupling that reinforces memory consolidation processes (Göldi et al., 2019; Ngo & Staresina, 2022). We thank the reviewer for giving us the opportunity to further clarify these aspects in the revised discussion.

- Page 21-22: “Our electrophysiological data show phase-specific modulations of sleep oscillations such that up-stimulated SOs exhibited higher amplitude and presented greater sigma power nested at their peak as compared to down-stimulated SOs. **In contrast, the different stimulation conditions did not modulate spindle event characteristics.** [...]”

In line with our findings, these earlier studies have collectively shown that stimulating SO at the up-phase of the slow-oscillation enhanced the amplitude of ongoing SOs^{17-21,26-28,30,34}, sigma oscillation power during the ascending phase of the SO¹⁸⁻²⁰ and memory consolidation^{17-21,27,28}. **It is worth noting though that we did not observe up-stimulation-induced modulations of spindle event characteristics.** However, these null results do not rule out the involvement of spindle activity in TMR-related motor memory consolidation processes. Recent evidence has brought forward the idea that spindle event detection in general is less sensitive than the study of the sigma rhythm as a whole^{14,35}. In line with these observations, our results show that sigma oscillation properties – as opposed to spindle events – were modulated by the phase of the stimulation. Interestingly, our results also indicate that these up-stimulation-induced modulations in sigma power emerged prior to auditory cue onset (i.e., prior to SO up-state). This result might be related to our repetitive stimulation paradigm which potentially allowed for the stimulation of consecutive SOs. Based on earlier work showing oscillatory entrainment by rhythmic acoustic stimulation^{36,37}, we suggest that our stimulation design particularly favored the entrainment of ongoing sleep oscillations resulting, with repetition, in power modulation prior to cue onset. Specifically, we propose that the phase-specific modulation of induced responses observed prior to cue onset reflects the effect of CL-TMR on ongoing top-down³⁸⁻⁴⁰ (here cortico-hippocampal⁴¹) processes. Further, we propose that the reinforcement of ongoing SO-sigma coupling favored spindle-mediated top-down mechanisms supporting memory reactivation and in turn enhanced memory performance⁴¹.”

We do agree with the reviewer that “sigma power” and “spindles” cannot be used interchangeably. We carefully reviewed the manuscript to edit any of these instances and modified the sentence introducing our electrophysiological results in the main manuscript as follows:

- Page 8: “EEG data collected during the TMR episode were analyzed to test whether (the phase of the) stimulation modulated the characteristics of electrophysiological markers critically involved in motor memory consolidation and reactivation during sleep, i.e., SOs, **spindle events, as well as sigma oscillations, taken as a proxy of spindle activity.**”
- Page 21: “**Slow oscillations, sleep spindles (events and sigma oscillations taken as a proxy of spindle activity)** and sharp wave ripples are part of the three cardinal NREM sleep oscillations playing a critical role in memory consolidation during sleep^{1,6,32}.”

Point #6: MRI Analysis. The analysis methodology and selective presentation of results makes it difficult to evaluate imaging results. The ROI selection is not well-motivated. The authors masked the ROI regions and then found significant clusters using voxelwise analyses before correcting with SVC. SVC is a correction that can be applied when the a priori hypothesis does not apply to the entire brain, but only to specific areas that can be encompassed within a small volume. This highlights the importance of building a strong a priori hypothesis that details the separate functional roles of each ROI selected. Moreover, the cluster size for the statistically significant ROIs was not included in the results.

Author Response: We thank the reviewer for giving us the opportunity to better describe (in this document and the revised manuscript) the functional role of the regions selected *a priori* for the analysis mask and for the correction for multiple comparisons. Our statistical analyses were indeed performed across all the voxels of a large mask including a set of task-relevant brain regions known to play a critical role in motor sequence learning (Berlot et al., 2020; Dolfen et al., 2022; Yokoi et al., 2018; Yokoi & Diedrichsen, 2019) and consisting of M1, PMC, SMA, aSPL, striatum and hippocampus. Note that this motor-network mask approach was chosen instead of a whole-brain approach to limit the number of identified clusters to *a priori*-defined task-relevant brain regions.

There is extensive evidence that motor sequence learning is associated with modulation of BOLD signal in these brain regions (see Albouy, King, et al., 2013; Dayan & Cohen, 2011; Doyon et al., 2003; Hikosaka et al., 2002; Penhune & Steele, 2012 for different reviews). These different models converge toward the general idea that parallel circuits operate during motor sequence learning to support the development of different features of the motor sequences. The learning of the spatial coordinates of the task recruits hippocampo-parietal circuits as well associative striatal areas whereas learning of the motor coordinates relies on M1-sensorimotor-striatum networks. Evidence reviewed in these models suggest that the transformation between these two coordinate systems is mediated by the SMA and the premotor cortex. Importantly, recent neuroimaging work has provided greater insight into the type of information represented in these brain regions (e.g., finger vs. sequence representation in M1, sequence vs. time representation in the PMC, movement-position binding in the striatum, response-goal-position binding in hippocampo-parietal networks; Berlot et al., 2020; Dolfen et al., 2024; Yokoi & Diedrichsen, 2019). Importantly, these regions have also been shown to play a critical role in the (wake and sleep-related) consolidation process (King et al., 2017). Specifically, sleep differently influences brain activity depending on the nature of the information supported by these different brain regions. Abstract (spatial) representations encoded in hippocampo-cortical regions have been shown to require sleep to consolidate while the consolidation of motoric representations is thought to occur over both wake and sleep intervals (Albouy et al., 2015). It is proposed that the between-representation transformation maps developed during learning in the SMA and PMC undergo sleep-dependent consolidation and facilitate the interaction between the spatial and motor representations of learning over sleep intervals to optimize motor behavior (King et al., 2017). Based on the literature reviewed above, we expected that up- (as compared to both down- and not-) stimulation would specifically result in strengthening task-related brain activity (at retest as compared to training) in hippocampo-parietal networks. We also expected that activity in striato-motor networks would increase after the consolidation period in concert with (e.g., via functional interaction with)

hippocampal-cortical networks for the up condition as compared to both the down and not conditions.

The functional role of the regions selected *a priori* for the analysis mask and for the correction for multiple comparisons is now presented in section 15 of the supplemental information and in the main text as follows:

- **Page 35:** “The goal of the fMRI analyses was to examine brain patterns elicited in specific **motor-related** regions. Therefore, statistical analyses were performed across all the voxels of a large mask including a set of task-relevant brain regions involved in motor sequence learning processes ^{22,84–88} and consisting of the primary motor cortex (M1), the supplementary motor cortex (SMA), the premotor cortex (PMC), the anterior part of the superior parietal lobule (aSPL), the hippocampus, the putamen and the caudate nucleus (and see section 15 in the supplements for additional information about the functional role of these brain regions in motor sequence learning and sleep-related motor memory consolidation). These brain areas were defined with the brainnetome atlas as follows.”

We also expanded the justification related to the specific coordinates that were used for the correction for multiple comparisons step. Indeed, the correction for multiple comparisons was performed using small volume correction approaches (Poldrack, 2007) across a smaller number of voxels located in spheres centered around coordinates of interest selected from the literature. This approach has been used extensively in the literature (e.g., Barakat et al., 2013; Fischer et al., 2005; Gann et al., 2021; Graydon et al., 2005; King et al., 2016; Peigneux et al., 2006; Rasch et al., 2007; Thielen et al., 2015). The list of coordinates and corresponding references are presented in the supplemental Table S7. The coordinates used for correction were reported in papers investigating both motor sequence learning (Gann et al., 2021; Lehericy et al., 2005, 2006; Schendan et al., 2003; Van Der Graaf et al., 2004) and motor memory consolidation (Albouy et al., 2008; Dolfen et al., 2021; Penhune & Doyon, 2002). As the task used in this research is the classical SRTT but was administered with explicit instructions, we included coordinates from studies examining motor sequence learning including both implicit (Albouy et al., 2008; Schendan et al., 2003; Van Der Graaf et al., 2004) and explicit (Dolfen et al., 2021; Gann et al., 2021; Lehericy et al., 2005, 2006; Schendan et al., 2003) versions of motor sequence learning, and using either the finger tapping task (Dolfen et al., 2021; Lehericy et al., 2005, 2006) or the serial reaction time task (Albouy et al., 2008; Gann et al., 2021; Schendan et al., 2003; Van Der Graaf et al., 2004). This approach was chosen as it is overall more conservative than classical ROI-based analyses where activity is averaged across voxels within each ROI.

To better justify the selection of coordinates from the literature, we added a new supplementary Table (Table S8) in which we provide a short description of the goal of the study / type of task used for each reference listed in the supplementary Table S7 presenting the coordinates used for small volume correction. This information is also summarized in the methods of the main manuscript as follows:

- **Page 36:** “Statistical inferences were performed at a threshold of $p < 0.05$ after family-wise error (FWE) correction for multiple comparisons over small volumes ⁸⁹ (SVC, 10 mm radius) located in the structures of interest reported by published work (see Table S7 in

supplemental information) investigating motor sequence learning^{90–95} and motor memory consolidation^{25,95,96}, using both implicit^{25,90,92,93} and explicit^{90,91,94–96} versions of motor sequence learning such as the finger tapping task^{91,94–96} or the serial reaction time task^{25,90,92,93} (see Table S8 in supplements for more information about tasks and processes examined). All results reported and discussed in the main text survived SVC.”

Last, as requested by the reviewer, we now report the size of all clusters presented in the tables of the supplemental information (Tables S1-3).

We hope the changes listed above clarify our approach for the *a priori* selection of regions of interest in our MRI analyses and alleviate the reviewer’s concerns.

Point #7: Seed Regions. For Connectivity-based analysis, the justification for using only right hemisphere seeds ‘in order to limit the number of PPI analyses’ (pp. 30, line 886) is not scientifically justified. The additional restriction to limit it to the right side was also based on a data driven difference as compared to an a priori hypothesis. The seed region for the hippocampus is a small section of the hippocampus, please justify why the ROI should not include the entire hippocampus or a subregion defined a priori. This significant cluster also includes white matter within activated cluster of voxels (in figure S8 lateral to the right hippocampus). Additionally, the seed regions are quite large for the caudate with small significant clusters.

Author Response: The reviewer is correct that seed region selection (and their laterality) was **not** based on *a priori* hypotheses for the connectivity analyses. The procedure for seed selection was **data-driven** as per our pre-registration (“PPI analyses will be computed to test task-related functional connectivity of seed regions based on the results of the activation-based analyses”). Seed selection was therefore entirely based on the results of the activation-based analyses. As the activation-based analyses showed that *peak maxima* for significant clusters in the hippocampus, putamen and caudate were located in the right hemisphere across the large majority of tested contrasts (see supplemental Table S2-1 and -2), right hippocampus, right caudate and right putamen were chosen as seed regions for the PPI analyses. This laterality effect was not hypothesized *a priori*. It is worth noting however that there is no known antero-posterior or hemispheric functional specialization of the hippocampus in the motor memory domain as compared to what is observed in the declarative memory domain (Poppenk et al., 2013; Strange et al., 1999). Therefore, in the PPI analyses performed in the current research, any sub-regional specificity of the seeds is purely data-driven and reflects the functional specificity of the BOLD responses associated to the experimental design. We apologize that the seed selection procedure was unclear and we have developed this point in the revised methods:

- Page 34: “Task-related functional connectivity was examined using psychophysiological interaction (PPI) analyses **with a data-driven seed selection approach**. Specifically, we assessed connectivity of three seed regions (right caudate (x = 10, y = 14, z = 12), right putamen (x = 18, y = 12, z = -2), and right hippocampus (x = 32, y = -38, z = -6)) revealed by the univariate analyses and showing a main effect of session across multiple conditions (see Figure S11 in supplemental information). **Note that peak maxima for significant clusters in**

the hippocampus, putamen and caudate were located in the right hemisphere across the large majority of tested contrasts (see supplemental Table S2-1 and 2) and that this laterality effect was not hypothesized a priori.”

With respect to the size of the spheres used to extract the time series from the seeds to perform the connectivity analyses, we also followed the pre-registered procedures and used 10mm-radius spheres to extract the seed’s signal. We acknowledge that for some regions, the resulting sphere includes non-grey matter tissues such as white matter or CSF (as shown in supplemental Figure S11). However, the procedure used to extract the time series is designed to minimize the influence of voxels that contribute little variance to the seed’s signal within the sphere. Specifically, the first eigenvariate was extracted using singular value decomposition of the time series across the voxels of the sphere. This procedure averages signal across voxels that explain most of the variance of the signal in the sphere. This method, as opposed to extracting the mean value across voxels, is more robust to heterogeneity of responses within the sphere and provides a weighted mean where atypical voxels (such as white matter or CSF) are down-weighted. The extracted time-series are thus considered as “a representative timeseries” for the region. For the reasons described above, this procedure is therefore not very sensitive to the size of the sphere used. To demonstrate so, we extracted the time-series from all three seed regions (caudate, hippocampus, and putamen) in all participants using a smaller sphere (6mm-radius) and correlated the extracted timeseries with the one obtained from a 10mm-radius sphere as used in the current study. As expected, the 6-mm and the 10-mm radius time series extracted in each participant and each seed regions were highly correlated (all p-values < 0.001 and all rho-values > 0.95). It is therefore unlikely that - with the methods used in the current research - smaller spheres would influence the pattern of connectivity results reported in the manuscript.

Point #8: MRI Correlation with Behavior. As there does not appear to be consolidation with any of the conditions and TMR does not appear to improve consolidation, the TMR index is not an interpretable measure of performance. A more interpretable correlation analysis between activity and behavior would be the Post-Pre BOLD change with the over-sleep performance improvement. Additionally, for the Post-Pre Bold changes, the scanning for the entire task performance was used, however for the correlated behavior, only the testing and first three AM training trials were used. It would be more meaningful to compare behavioral and MRI data collected over the same period.

Author Response: Concerning the reviewer’s concern about the lack of consolidation, we kindly invite the reviewer to refer to our response to their point #1 that emphasizes the importance of between-condition comparisons. With respect to the interpretability of the TMR index, in line with evidence discussed above and reported in the original manuscript, the TMR index was significantly different between up and down conditions ($t = 2.32$, $df = 27$, $p\text{-value} = 0.028$, Cohen’s $d = 0.44$). This suggests that (i) the TMR index is a metric that is sensitive to our experimental intervention and that (ii) the phase at which TMR was administered significantly influenced the behavioral proxy of motor memory consolidation. We therefore respectfully disagree with the reviewer that the TMR index is not an interpretable measure of performance. Accordingly, we opted to not deviate from our pre-registered procedures and used the TMR index rather than over-sleep performance enhancement for the brain-behavior regression analyses.

We thank the reviewer for raising the issue about the mismatch between the brain and behavioral data used in the regression analyses. While the behavioral data is indeed selected to reflect *offline* memory processes (i.e., a subset of blocks pre- and post-consolidation interval as is routinely done in the literature, e.g., Borragán et al., 2015; Ertelt et al., 2012; Robertson et al., 2004; Spencer et al., 2007), the MRI data reflects brain responses related to more extended practice pre- and post-consolidation interval. This is a recurrent problem in the field of offline memory consolidation as the different options available to match the behavioral and brain imaging contrasts all present some important caveats.

One possibility is to only analyze the 3 MRI blocks during pre- and post-night sessions corresponding to the behavioral blocks used for the computation of the offline change in performance. This option is usually not preferred as the signal-to-noise ratio obtained from only 3 MRI blocks pre- and post-sleep would not be sufficient to extract reliable brain activity and connectivity metrics. Another possibility is to compute the behavioral proxy of consolidation using all blocks of the pre- and post-sleep sessions (21 blocks in each session) that are used for the MRI contrast. While this is feasible, the behavioral marker does not longer reflect *offline* memory processes as online improvements in performance across the 21 practice blocks are also included in this metric (as well as potential fatigue and reactive inhibition effects (Rickard & Pan, 2017)). We nevertheless performed such computation and re-ran the corresponding behavioral (Figure R7) and brain-behavior regressions (Figure R8) analyses. Concerning the behavioral data, the results of the rmANOVA testing for a main effect condition including all behavioral blocks are similar as the results derived from the original and pre-registered analyses, although they exhibit stronger effect sizes. Specifically, between-session changes in performance speed from the 21 training blocks to the 21 retest blocks differed depending on the phase of the stimulated SO (Condition effect ($F(2,54) = 5.60$, $p = 0.0062$ (0.0075 sphericity corrected), $\eta^2 = 0.17$). Follow-up comparisons show that between-session changes in performance speed were greater for both the up- and not-reactivated sequences as compared to the down-reactivated sequence (up vs. down: $t = 3.50$, $df = 27$, $p\text{-value} < 0.001$ (0.0025 FDR-corrected), Cohen's $d = 0.66$; down vs. not: $t = -2.36$, $df = 27$, $p\text{-value} = 0.013$ (0.019 FDR-corrected), Cohen's $d = 0.45$). In contrast, between-session changes in performance speed did not significantly differ between the up- and not-reactivated sequences (up vs. not: $t = 0.27$, $df = 27$, $p\text{-value} = 0.40$ (0.59 FDR-corrected), Cohen's $d = 0.051$; see Figure R7). In line with the above and similar to our initial analyses, the TMR index computed over all practice blocks was significantly greater for the up than the down condition (TMR_{up} vs. TMR_{down} , $V = 338$, $p\text{-value} < 0.001$, Cohen's $d = 0.54$; and see below for regression analyses using these TMR indices).

Figure R7: Changes in performance speed between the pre- and post-night full training (% change between the average of the 21 blocks of pre-night training and the 21 blocks of post-night training) averaged across participants for the up-reativated (magenta), the down-reativated (blue) and the not-reativated (green) sequences. Results show a main effect of Condition (*: p -value < 0.05) whereby offline changes in performance speed were greater for the up- and not-reativated as compared to the down-reativated sequence (note that improvement in performance from pre- to post-night sessions is reflected by a positive change). Violin plots: median (horizontal bar), mean (diamond), the shape of the violin plots depicts the kernel density

estimate of the data. Colored points represent individual data, jittered in arbitrary distances on the x-axis within the respective violin plot to increase perceptibility. For each individual, performance on the different conditions are connected with a line between violin plots. n.s.: non-significant.

Concerning the brain-behavior regressions, new analyses were performed using the TMR index computed over all practice blocks. These new analyses revealed a pattern of results that is overall very similar to the results presented in the original manuscript. Specifically, this analysis replicated the correlations observed between the TMR index and (i) brain activity in the striatum for the up condition (Figure R8a-b and Figure 4c in the revised manuscript), (ii) brain connectivity in striato-hippocampal networks for both the up and down conditions (Figure R8c-d and Figure 6c, 7c in the revised manuscript) and (iii) brain connectivity in striato-motor networks for the down condition (Figure R8e and Figure 7c in the revised manuscript). The brain activity-behavior regressions initially observed in the striatum in the down condition (Figure R8b and Figure 4c in the revised manuscript) was reproduced only with a more permissive threshold ($p_{\text{uncorrected}} = 0.006$).

Altogether, the behavioral and brain-behavior regression analyses using all blocks yielded similar results as the analyses presented in the main manuscript. Given this consistency in results and because this measure using all blocks is not an accurate reflection of offline memory processes *per se*, we opted to not deviate from our pre-registered procedures and keep the initial behavioral and regression analyses in the revised manuscript. We hope the reviewer will find this decision satisfactory.

Figure R8: Overlay of the results derived from the regression analyses with the TMR index computed over a subset of blocks (initial TMR index) and the TMR index computed over all practice blocks (new TMR index). In each plot, the significant clusters of the initial (magenta or blue for up- or down-reactivated conditions, respectively) are overlaid with the new cluster (green or light blue for up- or down-reactivated conditions, respectively) derived from the regression analyses using the new TMR index computed as the percentage of change between the average of the 21 blocks of pre-night training and the 21 blocks of post-night training. Activations maps are displayed on a T1-weighted template image with a threshold of $p < 0.005$ (except for (b) where $p = 0.006$) uncorrected. **a-b.** Regressions between overnight increase in brain activity for the up-reactivated sequence and TMR_{Up} index (a) and between the overnight increase in brain activity for the down-reactivated sequence and the TMR_{Down} index (b). **c.** Regression between overnight decrease in connectivity for the up-reactivated sequence and the TMR_{Up} index. **d-e.** Regression between overnight increase (d) and decrease (e) in connectivity for the down-reactivated sequence and the TMR_{Down} index.

Minor issues

Point #9: Figure S7. The caption does not match the figure for the mean stimulation angle.

Author Response: We thank the reviewer for their thorough reading of our manuscript. The typo in the caption of Figure S7 (in the former version of the manuscript) has been corrected and this information is now reported in Figure 1 (panel c) of the main manuscript in response to point #6 of reviewer #1.

Point #10: Imaging. MRI Scan time is not stated.

Author Response: We apologize for this omission. Scan time was not constant across participants as it depended on the individuals' performance speed to perform the motor task. The mean scan time (and the 95% confidence interval) is now reported in the methods and is pasted below for convenience.

- Page 28: "Task-related fMRI data were acquired during the training and overnight retest sessions using an ascending gradient EPI pulse sequence for T2*-weighted images (TR = 2000 ms; TE = 29.8 ms; multiband factor 2; flip angle = 90°; 54 transverse slices; slice thickness = 2.5 mm; interslice gap = 0.2 mm; voxel size = 2.5 × 2.5 × 2.5 mm³; field of view = 210 × 210 × 145.6 mm³; matrix = 84 × 82) for each participant (max. 1200 dynamic scans). The average scan times were of 25.4 min (95CI: [24.3 - 26.5]) for the pre-night training, of 3.5 min (95CI: [3.4 - 3.7]) for the pre-night test, and of 22.9 min (95CI: [21.8 - 23.9]) for the post-night training."

REVIEWER #3

Remarks to the Author

Sleep helps to consolidate novel information for long-term. A process that is mediated by specific brain rhythms (and a specific interplay thereof) and relies on a reactivation of said information during sleep. Here, Nicolas et al. take advantage of a variety of techniques to examine sleep-dependent consolidation of motor memory: They perform targeted memory reactivation in real-time to reactivate motor traces in slow oscillation up- and down-states. This is further combined with fMRI pre- and post-sleep to assess changes in brain activity. Their data suggests that it is indeed possible to manipulate consolidation of motor memory, which is accompanied by activation changes within striatal and hippocampal networks.

All in all, this project must have been a difficult undertaking. Based on a clear study design and methodological approach, Nicolas et al. have gathered compelling evidence for the feasibility of real-time TMR within the procedural memory domain. I only have a few comments, I was hoping the authors could address to clarify a few aspects.

Author Response: We are grateful to the reviewer for the time devoted to our manuscript and for the positive remarks about our work. We address the reviewer's comments below and hope that our responses and the corresponding updates made to the manuscript (highlighted in yellow in the revised manuscript) are satisfactory.

Major comments

Point #1: Stimulation-locked evoked potentials: To assess the immediate effect of TMR, the authors use a different approach than other publications. Instead of averaging the signal time-locked to the stimulation onset, they use the previous online detected trough. I saw that the authors motivated this approach in the supplementary material, but I don't agree with that. They say this approach removes influence of the phase and I understand the idea is that it allows a direct comparison between all three conditions, however, the phase is exactly the targeted here. This approach also disadvantages the up-condition because the time between trough and peak should fluctuate across stimulation, which causes a temporal smearing/jitter of evoked responses. This in turn affects the up-down comparison. The authors should either derive control no-stim intervals for both up and down and compare each to their stimulation counterpart, or if they want to compare up vs. down, they could time-lock to the stimulus onset and ignore anything prior to the stimulation.

Author Response: We thank the reviewer for suggesting these alternative analyses. Following a similar point made by reviewer #1, we ran the stimulation-locked evoked potential analyses suggested by the reviewer. We kindly refer the reviewer to our response to point #2 (third comment) of reviewer #1 for detailed information. In short, we performed Event-Related Potentials (ERP) analyses around the auditory cues sent at the up and down phases of the SO with procedures similar to earlier research (Göldi et al., 2019; Ngo & Staresina, 2017). Further, and consistent with the recommendation of the reviewer, sham cues were derived from the not-stimulated intervals to compute ERPs on the control / not stimulated condition. Nonparametric cluster-based permutation tests were performed on a -1.5s to 1.5s window relative to cue onset

using paired t-test between conditions. Altogether, our results (detailed in the caption of Figure R1 above) partially replicated the results of earlier studies and discrepancies are likely related to differences in experimental design. Specifically, auditory stimulations in the current study were of shorter duration (100ms vs. 500ms in earlier research (Ngo & Staresina, 2022)) and were delivered using a blocked design with 3-min long intervals during which each stimulation type was sent repetitively which allowed the stimulation of consecutive SOs. The use of much shorter auditory cue duration in the current study might have induced less pronounced effect *during* sound exposure and did not allow up- and down-conditions to synchronize in contrast to earlier research (Ngo & Staresina, 2022) where conditions synchronized 400ms after cue onset into a second SO cycle. However, our rhythmic and sustained stimulation paradigm particularly favored the entrainment of ongoing sleep oscillations as shown by the strong pre-cue modulations in amplitude (see Figure R1 and response to point #2 of reviewer #1 for a detailed discussion of this point).

With respect to the time-frequency analyses, it is important to note that as up- and down-stimulated SOs are never in phase in the current study, the comparison of cue-locked time-frequency representations between conditions brings limited insight into power modulation in our study (beyond phase-related differences shown in Figure R1d). Instead, when phase-related modulations are controlled for (see trough-locked analyses presented in the original analyses of the main text), our results indicate a difference in sigma power between up and down conditions similar as the one reported in (Ngo & Staresina, 2022) on the second (synchronized) SO cycle elicited by the 500-ms acoustic stimulation.

We agree with the reviewer that using such analytical approach is warranted and is necessary to be able to compare our findings to earlier studies. These analyses are therefore now reported in the revised manuscript. However, as the results of the cue-locked analyses are highly confounded by the phase of SO in the current study, we opted to keep SO-trough-locked analyses as the main analyses in the revised manuscript and report the results of the cue-locked analyses (as well as interpretations and comparison with previous literature) in the supplements (see section 1 of the supplemental information and Figure S1). These results are called page 8 of the main text and the corresponding methods are described on page 31.

Point #2: Behavioural results: For one, did the authors examine the behavioural data for outliers? It seems the offline change for the down-condition is driven by some negative outliers. This additional point is not really major, but the direction of an improvement confused me. While in Figure 2a a smaller reaction time is better, the opposite direction is expected for Figure 2B. The authors do mention that in the methods section; however, it would help to highlight that earlier on as well.

Author Response: Outlier behavioral data were identified as described in our pre-registration, i.e., based on initial training performance (i.e., “...magnitude of [initial] learning was more than three standard deviations below the group average”). The individuals pointed out by the reviewer and showing more extreme negative offline changes in performance for the down-condition were not classified as outliers based on our pre-registered procedure mentioned above. However, to address the reviewer’s concern and a similar point made by reviewer #2 (see point #8), we examined whether participants showed outlier offline changes in performance (i.e., any individual

showing an offline change in performance that was more than three standard deviations away from the average offline change across participants and conditions). This procedure revealed that one individual presented outlier offline changes in performance in the down condition. Inspection of the data of this individual did not reveal any problematic behavioral data points (see Figure R6 presented in response to point #2 of reviewer #2) or any deviation from the group with respect to participant's demographics, characteristics, sleep and vigilance data. Therefore, we believe that there is no justification to deviate from our pre-registered procedures and to exclude this participant from our sample. For the sake of completeness, we nevertheless ran the rmANOVA on the offline changes in performance speed when excluding this more extreme participant and results highlighted a trend towards significance (Condition effect ($F(2,52) = 2.73, p = 0.074$ (0.074 sphericity corrected)) and a similar effect size as reported in the original analyses ($\eta^2 = 0.13$). This information is now added to the caption of Figure 2 in the revised text.

- **Page 8:** “Note that the main effect of condition is marginally significant when excluding the participant showing the extreme datapoint in the down condition (Condition effect ($F(2,52) = 2.73, p = 0.074, \eta^2 = 0.13$). This data point was not excluded from the primary analysis as it did not meet any criterion for outlier exclusion as defined in our pre-registration.”

Second, we thank the reviewer for pointing out the lack of clarity with respect to the direction of the effects in Figure 2. We now mention the direction of the improvement when the effects are first described in the Results section and in the caption of Figure 2 to improve readability.

- **Page 7:** We tested whether the stimulation conditions (up-, down- and not-reactivated) influenced the behavioral index of motor memory consolidation, i.e., the offline changes in performance speed observed between the pre-night test session and the beginning of the post-night training session (see Figure 2a, **note that in panel a, performance improvement is reflected by a decrease in response time - RT**). Results show that offline changes in performance speed differed depending on the phase of the stimulated SO (Condition effect ($F(2,54) = 3.88, p = 0.027$ (0.034 sphericity corrected), $\eta^2 = 0.13$; Figure 2b, $n = 28$; **note that in panel b, a positive offline change in performance reflects overnight performance improvement**).

Point #3: Overview of results: In a similar vein to my previous comment, given the sheer number of results, it was very difficult for me to keep an overview. For instance, while behaviourally, they only found an effect for the down condition (see later comment), physiological results show effects for both up and down or only down. This is perhaps asking too much, but it would really help to have some kind of graphical summary for the different results.

Author Response: We thank the reviewer for this suggestion. We understand that, given the multimodal nature of our study, a summary figure might be helpful for the reader to keep an overview of the various results. We now provide a summary figure in the revised discussion (Figure 8, pages 19-20) that is also presented in Figure R9 below for the reviewer's convenience. Note that the goal of this figure was to provide an overview of the most consistent findings across modalities rather than to highlight specific contrasts within each modality. As the most consistent results

across modalities were observed on the up versus down contrasts, the summary figure focuses on these particular comparisons.

Figure R9: Summary of the main findings. Slow-oscillation closed-loop targeted memory reactivation (CL-TMR) was applied during post-learning sleep so that 2 sounds corresponding to two different memory traces initially encoded during the pre-night training session (i.e., the road-runner motor sequence and the coyote motor sequence in the **Pre-night panel**) were replayed at the up- (magenta shades) or the down- (blue shade) phases of the SO. During post-learning sleep, CL-TMR at the SO-up-phase - as compared to the SO-down-phase - enhanced SO amplitude and resulted in a greater burst of sigma oscillations at the peak of the SO (**Night panel**). Motor performance collected during the

post-night retest (**Post-night panel**) indicated that up-state cueing resulted in greater performance improvement than down-state cueing (i.e., the road-runner motor sequence was executed faster than the coyote motor sequence). At the brain level, up stimulation - as compared to down stimulation - resulted in a greater overnight increase in activity in striato-motor regions and preserved hippocampal activity (see arrows within the brain regions of interest) as well as an overall decrease in hippocampo-striato-motor connectivity (represented by the dashed lines between brain regions). These up-state cueing strengthening in activity in – and segregation of – striato-motor and hippocampal networks were related to the beneficial effect of TMR on sleep features and performance. Down-stimulation (as compared to up-stimulation) resulted in reduced overnight changes in brain activity that were accompanied by increased connectivity in hippocampo-striato-motor networks. These overnight increases in connectivity were differently related to performance improvement suggesting that different networks may play distinct roles to compensate for the reduced plasticity induced by down-stimulation during sleep (see results for details). Turtle and rabbit adapted from La Fontaine. Cx: Cortex.

Point #4: Figure 4 and ROIs: Something that confused me as well was, why the authors used so many different ROIs for their analysis, e.g., in Figure 4. Why are regression analysis performed on different ROIs than found previously when looking at differences between conditions? Can the authors please motivate this as it seems I am missing something. Related to this, while the TMR index is derived by the difference between cued and uncued conditions, has the same been done for the BOLD response? I did not find that information.

Author Response: We apologize for the lack of clarity in the description of our procedures that might have led to a misunderstanding. Our statistical analyses were not performed using an ROI approach but across all the voxels of a large mask including a set of well-known task-relevant brain regions consisting of M1, PMC, SMA, aSPL, striatum and hippocampus (Berlot et al., 2020; Dolfen et al., 2024; Yokoi et al., 2018; Yokoi & Diedrichsen, 2019). This large anatomical mask was created with the brainnetome atlas (see methods) and included more than 7000 voxels. Cluster labelling was performed on the statistical maps displayed on the anatomical mask mentioned above at $p < 0.005$ uncorrected for multiple comparisons (see Figures 4-7 in the main text). This motor-network mask approach was chosen as it is overall more conservative than classical ROI-based analyses where activity is averaged across voxels within each ROI. Thus, all analyses (activation, connectivity and regression analyses) were performed on this large mask and highlight the clusters that are significant within this task-related mask. These clusters might therefore be different between e.g. activation-based and regression-based analyses.

We recognized that using the term “ROI” in the methods was confusing and we have now edited the revised text to better describe the masking procedure.

- Page 35: “The goal of the fMRI analyses was to examine brain patterns elicited in specific **motor-related** regions. Therefore, statistical analyses were performed across all the voxels of a large mask including a set of task-relevant brain regions involved in motor sequence learning processes ^{22,84–88} and consisting of the primary motor cortex (M1), the supplementary motor cortex (SMA), the premotor cortex (PMC), the anterior part of the superior parietal lobule (aSPL), the hippocampus, the putamen and the caudate nucleus (and see section 15 in the supplements for additional information about the functional role of these brain regions in motor sequence learning and sleep-related motor memory consolidation). These brain areas were defined with the brainnetome atlas as follows.”

With respect to the second point of the reviewer, the TMR index was indeed a difference between performance on the reactivated and the non-reactivated sequences as described in our pre-registration. The reviewer is also correct that in the regression analyses, the BOLD response was not subtracted between the reactivated and not-reactivated conditions (see page 35: “...we performed regression analysis between the individuals’ brain maps showing between session changes in activity/connectivity within each condition and the individuals’ TMR index (for each condition separately, i.e. $TMR_{index_{up}}$ and $TMR_{index_{down}}$)”). This approach was chosen in the sake of simplicity and to facilitate the interpretation of the complex regression models. However, we agree with the reviewer that this is an interesting suggestion and we therefore ran the corresponding analyses. Specifically, we performed regression analyses between the individuals’ brain maps showing between session (pre vs. post) **and** between condition (reactivated vs. not-reactivated) differences in activity/connectivity and the individuals’ TMR index.

For the activation-based contrasts, results show that the correlation reported in the original manuscript (see Figure 4c and 5c in revised manuscript) for both up and down conditions between the overnight changes in striato-motor and hippocampal activity and the TMR index was replicated with the between-condition contrasts (see Figure R10a-d below). Specifically, the new analyses show that the between-session increase in striato-motor activity that was greater in the reactivated (both up and down) as compared to the not-reactivated condition was related to a greater TMR index (Figure R10a,c-d). Additionally, the initial correlation between the overnight changes in hippocampal activity and the TMR index observed for both the up- and down-reactivated sequences (Figure 5c in the revised manuscript) was replicated with the between-condition contrast ‘up vs. not’. Here, the overnight decrease in activity – that was greater for the not-reactivated as compared to the up-reactivated condition - was related to a lower TMR index (Figure R10b).

For the connectivity-based contrasts, results show that the correlation reported in the original manuscript (see Figure 6c and 7d) for both up and down conditions between the overnight changes in striato-hippocampal connectivity and the TMR index was replicated with the between-condition contrasts (Figure R10e-f). Specifically, the new analyses show that the between-session decrease in striato-hippocampal connectivity - that was greater for the up- as compared to the not-reactivated sequence - was related to a greater TMR index (Figure R10e). Additionally, the initial correlation between the overnight increase in striato-hippocampal connectivity and the TMR index observed for the down-reactivated sequence (Figure 7d in the revised manuscript) was also replicated with the between-condition contrast. Here, a greater difference in connectivity increase between the down- and not-reactivated conditions was related to a greater TMR index (Figure R10f).

These analyses are now reported in the revised manuscript. However, as the regression analyses with the *between-condition* contrasts yielded similar results as the initial, pre-registered, analyses using the *within-condition* contrasts, we opted to keep the initial regression model as our main analyses in the revised manuscript and present the results of the regression analyses using the between-condition contrasts in the supplements (see supplemental Figure S5 that was pasted in the present document as Figure R10 for the reviewer’s convenience). The corresponding methods

are reported page 35 and the results are mentioned in the caption of each figure of the main text presenting regression analyses with the within-condition contrast.

Figure R10: Overlay of the initial (within-condition contrasts) and new (between-condition contrasts) regression analyses. In each plot, the significant clusters of the initial regression analyses (magenta or blue for up- or down-reactivated conditions, respectively) are overlaid with the results of the new regression analyses in which the TMR index is regressed against the difference in BOLD response between the reactivated and not-reactivated conditions between sessions. Activations maps are displayed on a T1-weighted template image with a threshold of $p < 0.005$ uncorrected. **a.** Regressions between the overnight increase in striatal activity for the up-reactivated [as compared to the not-reactivated] sequence and the TMR_{Up} index. **b.** Regressions between the overnight decrease in hippocampal activity for the up-reactivated [as compared to the not-reactivated] sequence and the TMR_{Up} index. **c-d.** Regressions between the overnight increase in striatal activity for the down-reactivated [as compared to the not-reactivated] sequence and the TMR_{Down} index. **e.** Regression between the overnight decrease in connectivity for the up-reactivated sequence [as compared to the not-reactivated] and the TMR_{Up} index. **f.** Regression between the overnight decrease in connectivity for the down-reactivated sequence [as compared to the not-reactivated] and the TMR_{Down} index.

Point #5: Behavioural interpretation: In the first sentence of the 2nd paragraph of the discussion, the authors write “Our behavioral results indicate that TMR applied at the up-phase of the SO resulted in greater gains in motor performance than when administered at the down-phase of the SO.” While it is true that the up-condition outperforms the down-condition, this is not the whole story. If we consider the uncued condition, we see that up-stimulation had no real benefit but rather down-stimulation lead to an impairment of memory. Another interpretation could be that up-state stimulation reactivated both the up- and no-stim sequences. Can the authors please clarify their statement or rephrase it.

Author Response: We agree with the reviewer that this is a possibility. This is discussed in the revised manuscript.

- Page 21: “Specifically, it is possible that, due to our within-subject design, overall acoustic stimulation during post-learning sleep might also have boosted performance on the control (not-reactivated) condition. This could have accentuated the difference in performance between down- and not-reactivated conditions and could also explain the lack of difference between up- and not-reactivated conditions in the current study. While this is hypothetical, it is in line with previous work using similar within-subject design showing no difference in performance between the up- and not-reactivated conditions ¹⁹.”

To stay closer to the data and in response to a similar point made by reviewer #1 (point #1), we replaced all the instances in the manuscript where SO-up-phase TMR was described to boost memory consolidation by statements that specify the contrasts supporting this conclusion. Accordingly, we now always specify that memory consolidation was boosted by SO-up-phase TMR **as compared to SO-down-phase TMR** (see point #1, reviewer #1).

Minor comments

Point #6: *Page 23, line 593 (Motor task): Out of curiosity, was there a pause between the training and testing blocks before sleep?*

Author Response: There was indeed a pause between the pre-night training and testing blocks to allow for fatigue to dissipate (Pan & Rickard, 2015) (see caption Figure 1 in the main text). We also made this information available in the revised Methods section as follows:

- Page 26: “The pre-night training session consisted of 63 practice blocks (21 blocks per sequence) immediately followed by a post-training test of 9 practice blocks (3 blocks per sequence). **These two sessions were separated by 5 minutes in order to allow fatigue effects to dissipate and minimize its confounding effect on end-of-training performance ²⁴. Note that the three sets of EPI images using reversed phase-encoding polarity were acquired during this break (see fMRI data acquisition below).**”

Point #7: *Page 24, line 646 (Closed-loop TMR) Was there a pause after a cue was delivered?*

Author Response: After a cue was delivered, stimulation was paused until the next zero crossing (from up to down phase). This prevented sending multiple stimulations in a single SO but permitted potentially stimulating consecutive SOs. We thank the reviewer for pointing out this omission and we now describe the pause procedure in the revised manuscript as follows:

- Page 27: “First, the online detection algorithm used a fast-moving average filter with a window of 50 samples. A down-phase was detected when signal went below a specific threshold adapted for biological sex according to Rosinvil et al. ²³ and of -41 μV in females and 39.5 μV in males. The up-phase of a SO was identified when, in addition to the criterion described for down-phase detection above, peak-to-peak signal amplitude reached 77 μV ”

in females and 74 μ V in males. After a cue was delivered, the stimulation algorithm was paused until the next zero crossing (from up to down phase) to avoid stimulating the same SO multiple times but to allow the stimulation of consecutive SOs. As a result, 90.83 % [95CI: 87.13 – 94.53] of the auditory cues were sent at a frequency that corresponds to the SO frequency range (i.e., 0.1 to 4.5 Hz)."

Point #8: Page 24, line 674 (Closed-loop TMR): I might have missed it, the number of detected no-stimulation events? I assume those numbers were much lower compared to the up- and down-condition considering it was performed in only 1-min intervals. If that is the case, how was that considered in the data analysis?

Author Response: We thank the reviewer for pointing out a lack of clarity in the description of the stimulation algorithm. The *minimum* duration of the rest intervals was indeed 1 min but the actual duration of each rest interval depended on the number of events reached in each condition. Indeed, as described on page 28 of the manuscript, an online validation algorithm was used to match the number of true positive stimulations between up and down conditions. This algorithm was designed to stop stimulating the condition with the more lenient / fewer criteria (i.e., the down condition) until the number of true positive stimulation was similar between the two conditions. For example, if at the end of the 3min-long up stimulation interval, the number of true positives in the up condition was lower than in the down condition, the next down-stimulation interval was silenced which resulted in a rest interval duration of 5min (1min rest + 3min silenced down condition + 1min rest) before the next up stimulation interval started. This resulted in a significant increase in the no-stimulation time intervals. Note that the duration of the stimulation intervals could also be reduced in case the experimenter paused stimulation based on the online scoring during data collection (e.g., in case of arousals or wake, NREM1 or REM sleep). This eventually resulted in overall similar stimulation / no-stimulation interval durations and, most importantly, a similar number of analyzed stimuli between conditions. Specifically, the average interval duration was 2.30 min [95CI: 2.22 -2.39] for the up-, 2.38 min [95CI: 2.29 – 2.47] for the down- and 2.58 min [95CI: 2.25 - 2.91] for the not-stimulated conditions. The rmANOVA on the number of events with Condition as within-subject factor (up vs. down vs. not-stimulated events) showed a significant effect of condition ($F(2,58) = 3.95$, $p = 0.025$, $\eta^2 = 0.12$; See Figure R11). Post-hoc analyses on the 3 possible paired-comparisons showed however that this effect did not survive correction for multiple comparisons (up vs. down: $t = 1.70$, $df = 29$, p -value = 0.10 (0.13 FDR-corrected), Cohen's $d = 0.070$; up vs. not-stimulated: $t = -1.58$, $df = 29$, p -value = 0.13 (0.13 FDR-corrected), Cohen's $d = 0.12$; down vs. not-stimulated: $t = -.235$, $df = 29$, p -value = 0.026 (0.077 FDR-corrected), Cohen's $d = 0.19$).

We thank the reviewer for giving us the opportunity to clarify these aspects in the revised manuscript. The description of the online validation algorithm and its effect on the duration of the no-stimulation intervals are now described in the revised text as follows:

- Page 5: "Three-min long up- and down-stimulation intervals alternated and were separated by a no-stimulation rest period of a minimal duration of 1min (see methods for details on the actual duration of the no-stimulation intervals). The sounds associated to the up (or down)-

reactivated sequence were then played on the peak (or trough) of the SOs within these alternating intervals. Online detection was performed on FPz.”

- Page 27: “The auditory stimulation was presented in a blocked design with 3-min long intervals that alternated between up- and down-SO detection/stimulation. **Each stimulation interval was separated by a no-stimulation rest period of a minimal duration of 1min (see below for details on the actual duration of the no-stimulation intervals).**”
- Page 28: “Specifically, any time the total count of stimulation was greater for the down as compared to the up condition, the algorithm temporarily paused during the following down-detection interval. **For example, if at the end of the 3 min-long up stimulation interval, the number of true positives in the up condition was lower than in the down condition, the next down-stimulation interval was silenced which resulted in a rest interval duration of 5 min (1 min rest + 3 min silenced down condition + 1 min rest) before the next up stimulation interval started.** Note that the duration of the stimulation intervals could also be reduced in case the experimenter paused stimulation based on the online scoring during data collection (e.g., in case of arousals, wake, NREM1 or REM sleep). This eventually resulted in overall similar stimulation / no-stimulation interval durations and a similar number analyzed stimuli between conditions (see section 4.6.2 below for the number of analyzed stimuli after offline validation). Specifically, the average interval duration was 2.30 min [95CI: 2.22 -2.39] for the up-, 2.38 min [95CI: 2.29 – 2.47] for the down- and 2.58 min [95CI: 2.25 - 2.91] for the not-stimulated conditions. The rmANOVA on interval duration with Condition as within-subject factor (up vs. down vs. not-stimulated intervals) showed no significant condition effect ($F(2,58) = 18.56$, $p = 0.070$ (0.093 sphericity corrected), $\eta^2 = 0.088$).”

Additionally, information about the total number of analyzed events in each condition is now reported in the Methods section of the main manuscript as follows.

- Pages 31-32: “The average number of artifact-free trials by condition was of 622.9 [95% CI: 495.1 – 750.7] for the up-, 599.7 [95% CI: 482.4 – 717.1] for the down-, and 664.3 [95% CI: 528.7 – 800.0] for the not-stimulated conditions. **The rmANOVA on the number of trials with Condition as within-subject factor (up vs. down vs. not-stimulated) showed a significant condition effect ($F(2,58) = 3.95$, $p = 0.025$, $\eta^2 = 0.12$).** Post-hoc analyses on the 3 possible paired-comparisons showed, however, that this effect did not survive correction for multiple comparisons (up vs. down: $t = 1.70$, , $df = 29$, $p\text{-value} = 0.10$ (0.13 FDR-corrected), Cohen’s $d = 0.070$; up vs. not-stimulated: $t = -1.58$, , $df = 29$, $p\text{-value} = 0.13$ (0.13 FDR-corrected), Cohen’s $d = 0.12$; down vs. not-stimulated: $t = -.235$, $df = 29$, $p\text{-value} = 0.026$ (0.077 FDR-corrected), Cohen’s $d = 0.19$).”

Related to the point raised by the reviewer above, it came to our attention during the revision process that the caption of Table S5 inaccurately stated that the number of true positives reported in the Table was derived from the *online* validation algorithm while they were in fact derived from the *offline* validation algorithm. This error is now fixed in the revised supplements. Also note that the number of *detected* true positives reported in Table S5 and the number of *analyzed* events for both the cued- and trough-locked analyses reported in the main text are slightly different due to the preprocessing methods for artefact rejection. This is also now specified in the caption of Table S5.

Figure R11: Number of analyzed events in the main (trough-locked) EEG analysis for the up-stimulated (magenta), the down-stimulated (blue), and the not-stimulated (green) conditions. Violin plots: median (horizontal bar), mean (diamond), the shape of the violin plots depicts the kernel density estimate of the data. Colored points represent individual data, jittered in arbitrary distances on the x-axis within the respective violin plot to increase perceptibility. For each individual, the number of trials for the different conditions are connected with a line between violin plots. n.s.: non-significant.

Violin plots: median (horizontal bar), mean (diamond), the shape of the violin plots depicts the kernel density estimate of the data. Colored points represent individual data, jittered in arbitrary distances on the x-axis within the respective violin plot to increase perceptibility. For each individual, the number of trials for the different conditions are connected with a line between violin plots. n.s.: non-significant.

Point #9: Page 30, line 902 (Regression analysis): I know that some do not put too much weight on regression assumptions, but I

do find it important to show that at least homoscedasticity and normally distributed residuals are present. Otherwise, the regression models are not fully interpretable. Did the authors perform those assumption checks?

Author Response: For each linear regression model between a brain metric (activity or connectivity) and a covariate (TMR index, SW amplitude and sigma power) presented in the manuscript, we provide below in Table R2 the QQ-plots of the residuals (theoretical quantiles in abscissa and standardized residuals in ordinate) and the results of the Breusch-Pagan test which tests for heteroscedasticity in a linear regression model (in this test, the null hypothesis is the homoscedasticity of the residuals' variance). In terms of normality of the residuals, we observe that the large majority of the linear regression analyses presented residuals within the quantile-quantile confidence band. Note that two models (#1 and #11 in Table R2) presented one residual outside of the quantile-quantile confidence and for two other models (#3 and #12), three residuals were outside the range. In these models, the homoscedasticity of the variances was not violated (Breusch-Pagan test p-values > 0.27). Homoscedasticity of the variances was observed in all but one linear regression (model #7, Breusch-Pagan test p = 0.04). Overall, these analyses indicate that regression assumptions were met and that the corresponding models are interpretable.

Table R2: Q-Q plot of the residuals of the activation- and connectivity-based regressions with experimental data quantiles on the y-axis and normal theoretical quantiles on the x-axis and results of the Breusch-Pagan test.

#1	Position in main manuscript	Independent variable	Condition	Coordinates
	Page 13, Figure 4c	TMR index	up	right caudate x = 20, y = 18, z = 12

#2	Page 13, Figure 4c	TMR index	down	right caudate x = 16, y = -2, z = 26
#3	Page 13, Figure 4d (top panel)	SO peak amplitude	up	left M1 x = -46, y = -16, z = 48
#4	Page 13, Figure 4d (top panel)	SO peak amplitude	down	left M1 x = -44, y = -16, z = 52

#5	Page 13, Figure 4d (bottom panel)	SO peak amplitude	up	right pallidum x = 20, y = -2, z = -6
				
#6	Page 14, Figure 5c	TMR index	up	right hippocampus x = 36, y = -38, z = -8
				
#7	Page 14, Figure 5c	TMR index	down	right hippocampus x = 20, y = -34, z = 4

#8	Page 15, Figure 6c	TMR index	up	hippocampus-right putamen x = 32, y = -8, z = 4
#9	Page 17, Figure 7c (left panel)	sigma power	down	Caudate-left hippocampus x = -24, y = -14, z = -8
#10	Page 17, Figure 7c (left panel)	sigma power	down	Putamen-left hippocampus x = -18, y = -10, z = -8

#11	Page 17, Figure 7c (right panel)	SO peak amplitude	down	Caudate-left aSPL x = -44, y = -44, z = 38
#12	Page 17, Figure 7c (right panel)	SO peak amplitude	down	Putamen-left aSPL x = -38, y = -42, z = 36
#13	Page 17, Figure 7d (left panel)	TMR index	down	Putamen-right M1 x = 26, y = -8, z = 44

#14	Page 17, Figure 7d (right panel)	TMR index	down	caudate-left hippocampus x = -16, y = -40, z = 6

REFERENCES

- Aksamaz, S., Mölle, M., Akinola, E. O., Gromodka, E., Bazhenov, M., & Marshall, L. (2024). Single closed-loop acoustic stimulation targeting memory consolidation suppressed hippocampal ripple and thalamo-cortical spindle activity in mice. *European Journal of Neuroscience*, *59*(4), 595-612.
<https://doi.org/10.1111/ejn.16116>
- Albouy, G., Fogel, S., King, B. R., Laventure, S., Benali, H., Karni, A., Carrier, J., Robertson, E. M., & Doyon, J. (2015). Maintaining vs. enhancing motor sequence memories : Respective roles of striatal and hippocampal systems. *NeuroImage*, *108*, 423-434.
<https://doi.org/10.1016/j.neuroimage.2014.12.049>
- Albouy, G., Fogel, S., Pottiez, H., Nguyen, V. A., Ray, L., Lungu, O., Carrier, J., Robertson, E., & Doyon, J. (2013). Daytime Sleep Enhances Consolidation of the Spatial but Not Motoric Representation of Motor Sequence Memory. *PLoS ONE*, *8*(1), e52805.
<https://doi.org/10.1371/journal.pone.0052805>
- Albouy, G., King, B. R., Maquet, P., & Doyon, J. (2013). Hippocampus and striatum : Dynamics and interaction during acquisition and sleep-related motor sequence memory consolidation: Hippocampus and Striatum and Procedural Memory Consolidation. *Hippocampus*, *23*(11), 985-1004. <https://doi.org/10.1002/hipo.22183>
- Albouy, G., Ruby, P., Phillips, C., Luxen, A., Peigneux, P., & Maquet, P. (2006). Implicit oculomotor sequence learning in humans : Time course of offline processing. *Brain Research*, *1090*(1), 163-171. <https://doi.org/10.1016/j.brainres.2006.03.076>
- Albouy, G., Sterpenich, V., Baiteau, E., Vandewalle, G., Deseilles, M., Dang-Vu, T., Darsaud, A., Ruby, P., Luppi, P.-H., Degueldre, C., Peigneux, P., Luxen, A., & Maquet, P. (2008). Both the Hippocampus and Striatum Are Involved in Consolidation of Motor Sequence Memory. *Neuron*, *58*(2), 261-272.
<https://doi.org/10.1016/j.neuron.2008.02.008>

- Albouy, G., Vandewalle, G., Sterpenich, V., Rauchs, G., Desseilles, M., Balteau, E., Degueldre, C., Phillips, C., Luxen, A., & Maquet, P. (2013). Sleep stabilizes visuomotor adaptation memory : A functional magnetic resonance imaging study. *Journal of Sleep Research*, *22*(2), 144-154.
<https://doi.org/10.1111/j.1365-2869.2012.01059.x>
- Antony, J. W., Gobel, E. W., O'Hare, J. K., Reber, P. J., & Paller, K. A. (2012). Cued memory reactivation during sleep influences skill learning. *Nature Neuroscience*, *15*(8), 1114-1116.
<https://doi.org/10.1038/nn.3152>
- Barakat, M., Carrier, J., Debas, K., Lungu, O., Fogel, S., Vandewalle, G., Hoge, R. D., Bellec, P., Karni, A., Ungerleider, L. G., Benali, H., & Doyon, J. (2013). Sleep spindles predict neural and behavioral changes in motor sequence consolidation : Sleep Spindles Predict Motor Consolidation. *Human Brain Mapping*, *34*(11), 2918-2928. <https://doi.org/10.1002/hbm.22116>
- Berlot, E., Popp, N. J., & Diedrichsen, J. (2020). A critical re-evaluation of fMRI signatures of motor sequence learning. *eLife*, *9*, e55241. <https://doi.org/10.7554/eLife.55241>
- Bonnefond, M., Kastner, S., & Jensen, O. (2017). Communication between Brain Areas Based on Nested Oscillations. *Eneuro*, *4*(2), ENEURO.0153-16.2017. <https://doi.org/10.1523/ENEURO.0153-16.2017>
- Borragán, G., Urbain, C., Schmitz, R., Mary, A., & Peigneux, P. (2015). Sleep and memory consolidation : Motor performance and proactive interference effects in sequence learning. *Brain and Cognition*, *95*, 54-61. <https://doi.org/10.1016/j.bandc.2015.01.011>
- Brashers-Krug, T., Shadmehr, R., & Bizzi, E. (1996). Consolidation in human motor memory. *Nature*, *382*(6588), 252-255. <https://doi.org/10.1038/382252a0>
- Brawn, T. P., Fenn, K. M., Nusbaum, H. C., & Margoliash, D. (2010). Consolidating the Effects of Waking and Sleep on Motor-Sequence Learning. *The Journal of Neuroscience*, *30*(42), 13977-13982.
<https://doi.org/10.1523/JNEUROSCI.3295-10.2010>

- Britvina, T., & Eggermont, J. J. (2008). Multi-frequency auditory stimulation disrupts spindling activity in anesthetized animals. *Neuroscience*, *151*(3), 888-900.
<https://doi.org/10.1016/j.neuroscience.2007.11.028>
- Cai, D. J., & Rickard, T. C. (2009). Reconsidering the role of sleep for motor memory. *Behavioral Neuroscience*, *123*(6), 1153-1157. <https://doi.org/10.1037/a0017672>
- Cairney, S. A., Guttesen, A. á V., El Marj, N., & Staresina, B. P. (2018). Memory Consolidation Is Linked to Spindle-Mediated Information Processing during Sleep. *Current Biology*, *28*(6), 948-954.e4.
<https://doi.org/10.1016/j.cub.2018.01.087>
- Carrier, J., Viens, I., Poirier, G., Robillard, R., Lafortune, M., Vandewalle, G., Martin, N., Barakat, M., Paquet, J., & Filipini, D. (2011). Sleep slow wave changes during the middle years of life : Changes in slow waves with age. *European Journal of Neuroscience*, *33*(4), 758-766.
<https://doi.org/10.1111/j.1460-9568.2010.07543.x>
- Chen, C.-C., Kiebel, S. J., Kilner, J. M., Ward, N. S., Stephan, K. E., Wang, W.-J., & Friston, K. J. (2012). A dynamic causal model for evoked and induced responses. *NeuroImage*, *59*(1), 340-348.
<https://doi.org/10.1016/j.neuroimage.2011.07.066>
- Cousins, J. N., El-Deredy, W., Parkes, L. M., Hennies, N., & Lewis, P. A. (2016). Cued Reactivation of Motor Learning during Sleep Leads to Overnight Changes in Functional Brain Activity and Connectivity. *PLOS Biology*, *14*(5), e1002451. <https://doi.org/10.1371/journal.pbio.1002451>
- David, O., Kilner, J. M., & Friston, K. J. (2006). Mechanisms of evoked and induced responses in MEG/EEG. *NeuroImage*, *31*(4), 1580-1591. <https://doi.org/10.1016/j.neuroimage.2006.02.034>
- Dayan, E., & Cohen, L. G. (2011). Neuroplasticity Subservicing Motor Skill Learning. *Neuron*, *72*(3), 443-454.
<https://doi.org/10.1016/j.neuron.2011.10.008>
- Debas, K., Carrier, J., Orban, P., Barakat, M., Lungu, O., Vandewalle, G., Tahar, A. H., Bellec, P., Karni, A., Ungerleider, L. G., Benali, H., & Doyon, J. (2010). Brain plasticity related to the consolidation of

- motor sequence learning and motor adaptation. *Proceedings of the National Academy of Sciences*, 107(41), 17839-17844. <https://doi.org/10.1073/pnas.1013176107>
- Dimitrov, T., He, M., Stickgold, R., & Prerau, M. J. (2021). Sleep spindles comprise a subset of a broader class of electroencephalogram events. *Sleep*, zsab099. <https://doi.org/10.1093/sleep/zsab099>
- Dolfen, N., King, B. R., Schwabe, L., Gann, M. A., Veldman, M. P., von Leupoldt, A., Swinnen, S. P., & Albouy, G. (2021). Stress Modulates the Balance between Hippocampal and Motor Networks during Motor Memory Processing. *Cerebral Cortex*, 31(2), 1365-1382. <https://doi.org/10.1093/cercor/bhaa302>
- Dolfen, N., Reverberi, S., Op De Beeck, H., King, B. R., & Albouy, G. (2022). *The hippocampus binds movements to their temporal position in a motor sequence.* <https://doi.org/10.1101/2022.12.20.521084>
- Dolfen, N., Reverberi, S., Op De Beeck, H., King, B. R., & Albouy, G. (2024). The hippocampus represents information about movements in their temporal position in a learned motor sequence. *The Journal of Neuroscience*, e0584242024. <https://doi.org/10.1523/JNEUROSCI.0584-24.2024>
- Doyon, J., Korman, M., Morin, A., Dostie, V., Tahar, A. H., Benali, H., Karni, A., Ungerleider, L. G., & Carrier, J. (2009). Contribution of night and day sleep vs. Simple passage of time to the consolidation of motor sequence and visuomotor adaptation learning. *Experimental Brain Research*, 195(1), 15-26. <https://doi.org/10.1007/s00221-009-1748-y>
- Doyon, J., Penhune, V., & Ungerleider, L. G. (2003). Distinct contribution of the cortico-striatal and cortico-cerebellar systems to motor skill learning. *Neuropsychologia*, 41(3), 252-262. [https://doi.org/10.1016/S0028-3932\(02\)00158-6](https://doi.org/10.1016/S0028-3932(02)00158-6)
- Ertelt, D., Witt, K., Reetz, K., Frank, W., Junghanns, K., Backhaus, J., Tadic, V., Pellicano, A., Born, J., & Binkofski, F. (2012). Skill Memory Escaping from Distraction by Sleep—Evidence from Dual-Task Performance. *PLoS ONE*, 7(12), e50983. <https://doi.org/10.1371/journal.pone.0050983>

- Fischer, S., Nitschke, M. F., Melchert, U. H., Erdmann, C., & Born, J. (2005). Motor Memory Consolidation in Sleep Shapes More Effective Neuronal Representations. *The Journal of Neuroscience*, *25*(49), 11248-11255. <https://doi.org/10.1523/JNEUROSCI.1743-05.2005>
- Gann, M. A., King, B. R., Dolfen, N., Veldman, M. P., Chan, K. L., Puts, N. A. J., Edden, R. A. E., Davare, M., Swinnen, S. P., Mantini, D., Robertson, E. M., & Albouy, G. (2021). Hippocampal and striatal responses during motor learning are modulated by prefrontal cortex stimulation. *NeuroImage*, *237*, 118158. <https://doi.org/10.1016/j.neuroimage.2021.118158>
- Göldi, M., van Poppel, E. A. M., Rasch, B., & Schreiner, T. (2019). Increased neuronal signatures of targeted memory reactivation during slow-wave up states. *Scientific Reports*, *9*(1), 2715. <https://doi.org/10.1038/s41598-019-39178-2>
- Graydon, F. X., Friston, K. J., Thomas, C. G., Brooks, V. B., & Menon, R. S. (2005). Learning-related fMRI activation associated with a rotational visuo-motor transformation. *Cognitive Brain Research*, *22*(3), 373-383. <https://doi.org/10.1016/j.cogbrainres.2004.09.007>
- Haegens, S., Handel, B. F., & Jensen, O. (2011). Top-Down Controlled Alpha Band Activity in Somatosensory Areas Determines Behavioral Performance in a Discrimination Task. *Journal of Neuroscience*, *31*(14), 5197-5204. <https://doi.org/10.1523/JNEUROSCI.5199-10.2011>
- Haegens, S., & Zion Golumbic, E. (2018). Rhythmic facilitation of sensory processing : A critical review. *Neuroscience & Biobehavioral Reviews*, *86*, 150-165. <https://doi.org/10.1016/j.neubiorev.2017.12.002>
- Hikosaka, O., Nakamura, K., Sakai, K., & Nakahara, H. (2002). Central mechanisms of motor skill learning. *Current Opinion in Neurobiology*, *12*(2), 217-222. [https://doi.org/10.1016/S0959-4388\(02\)00307-](https://doi.org/10.1016/S0959-4388(02)00307-0)

0

- Hu, X., Cheng, L. Y., Chiu, M. H., & Paller, K. A. (2020). Promoting memory consolidation during sleep : A meta-analysis of targeted memory reactivation. *Psychological Bulletin*, 146(3), 218-244.
<https://doi.org/10.1037/bul0000223>
- Jensen, O., & Mazaheri, A. (2010). Shaping Functional Architecture by Oscillatory Alpha Activity : Gating by Inhibition. *Frontiers in Human Neuroscience*, 4. <https://doi.org/10.3389/fnhum.2010.00186>
- King, B. R., Hoedlmoser, K., Hirschauer, F., Dolfen, N., & Albouy, G. (2017). Sleeping on the motor engram : The multifaceted nature of sleep-related motor memory consolidation. *Neuroscience & Biobehavioral Reviews*, 80, 1-22. <https://doi.org/10.1016/j.neubiorev.2017.04.026>
- King, B. R., Saucier, P., Albouy, G., Fogel, S. M., Rumpf, J.-J., Klann, J., Buccino, G., Binkofski, F., Classen, J., Karni, A., & Doyon, J. (2016). Cerebral Activation During Initial Motor Learning Forecasts Subsequent Sleep-Facilitated Memory Consolidation in Older Adults. *Cerebral Cortex*, bhv347.
<https://doi.org/10.1093/cercor/bhv347>
- Krakauer, J. W. (2009). Motor Learning and Consolidation : The Case of Visuomotor Rotation. In D. Sternad (Éd.), *Progress in Motor Control* (Vol. 629, p. 405-421). Springer US.
https://doi.org/10.1007/978-0-387-77064-2_21
- Krakauer, J. W., Ghez, C., & Ghilardi, M. F. (2005). Adaptation to Visuomotor Transformations : Consolidation, Interference, and Forgetting. *The Journal of Neuroscience*, 25(2), 473-478.
<https://doi.org/10.1523/JNEUROSCI.4218-04.2005>
- Krakauer, J. W., Ghilardi, M.-F., & Ghez, C. (1999). Independent learning of internal models for kinematic and dynamic control of reaching. *Nature Neuroscience*, 2(11), 1026-1031.
<https://doi.org/10.1038/14826>
- Lakatos, P., Gross, J., & Thut, G. (2019). A New Unifying Account of the Roles of Neuronal Entrainment. *Current Biology*, 29(18), R890-R905. <https://doi.org/10.1016/j.cub.2019.07.075>

- Lehéricy, S., Bardinet, E., Tremblay, L., Van De Moortele, P.-F., Pochon, J.-B., Dormont, D., Kim, D.-S., Yelnik, J., & Ugurbil, K. (2006). Motor control in basal ganglia circuits using fMRI and brain atlas approaches. *Cerebral Cortex*, *16*(2), 149-161. <https://doi.org/10.1093/cercor/bhi089>
- Lehéricy, S., Benali, H., Van De Moortele, P.-F., Péligrini-Issac, M., Waechter, T., Ugurbil, K., & Doyon, J. (2005). Distinct basal ganglia territories are engaged in early and advanced motor sequence learning. *Proceedings of the National Academy of Sciences*, *102*(35), 12566-12571. <https://doi.org/10.1073/pnas.0502762102>
- Maquet, P., Laureys, S., Peigneux, P., Fuchs, S., Petiau, C., Phillips, C., Aerts, J., Del Fiore, G., Degueldre, C., Meulemans, T., Luxen, A., Franck, G., Van Der Linden, M., Smith, C., & Cleeremans, A. (2000). Experience-dependent changes in cerebral activation during human REM sleep. *Nature Neuroscience*, *3*(8), 831-836. <https://doi.org/10.1038/77744>
- Nettersheim, A., Hallschmid, M., Born, J., & Diekelmann, S. (2015). The Role of Sleep in Motor Sequence Consolidation : Stabilization Rather Than Enhancement. *The Journal of Neuroscience*, *35*(17), 6696-6702. <https://doi.org/10.1523/JNEUROSCI.1236-14.2015>
- Ngo, H.-V. V., & Staresina, B. P. (2017). Shifting memories. *eLife*, *6*, e30774. <https://doi.org/10.7554/eLife.30774>
- Ngo, H.-V. V., & Staresina, B. P. (2022). Shaping overnight consolidation via slow-oscillation closed-loop targeted memory reactivation. *Proceedings of the National Academy of Sciences*, *119*(44), e2123428119. <https://doi.org/10.1073/pnas.2123428119>
- Nicolas, J., Carrier, J., Swinnen, S. P., Doyon, J., Albouy, G., & King, B. R. (2023). Targeted memory reactivation during post-learning sleep does not enhance motor memory consolidation in older adults. *Journal of Sleep Research*, e14027. <https://doi.org/10.1111/jsr.14027>

- Nicolas, J., King, B. R., Levesque, D., Lazzouni, L., Coffey, E., Swinnen, S., Doyon, J., Carrier, J., & Albouy, G. (2022). Sigma oscillations protect or reinstate motor memory depending on their temporal coordination with slow waves. *eLife*, *11*, e73930. <https://doi.org/10.7554/eLife.73930>
- Pan, S. C., & Rickard, T. C. (2015). Sleep and motor learning : Is there room for consolidation? *Psychological Bulletin*, *141*(4), 812-834. <https://doi.org/10.1037/bul0000009>
- Peigneux, P., Orban, P., Baiteau, E., Degueldre, C., Luxen, A., Laureys, S., & Maquet, P. (2006). Offline Persistence of Memory-Related Cerebral Activity during Active Wakefulness. *PLoS Biology*, *4*(4), e100. <https://doi.org/10.1371/journal.pbio.0040100>
- Penhune, V. B., & Doyon, J. (2002). Dynamic Cortical and Subcortical Networks in Learning and Delayed Recall of Timed Motor Sequences. *The Journal of Neuroscience*, *22*(4), 1397-1406. <https://doi.org/10.1523/JNEUROSCI.22-04-01397.2002>
- Penhune, V. B., & Steele, C. J. (2012). Parallel contributions of cerebellar, striatal and M1 mechanisms to motor sequence learning. *Behavioural Brain Research*, *226*(2), 579-591. <https://doi.org/10.1016/j.bbr.2011.09.044>
- Poldrack, R. A. (2007). Region of interest analysis for fMRI. *Social Cognitive and Affective Neuroscience*, *2*(1), 67-70. <https://doi.org/10.1093/scan/nsm006>
- Poppenk, J., Evensmoen, H. R., Moscovitch, M., & Nadel, L. (2013). Long-axis specialization of the human hippocampus. *Trends in Cognitive Sciences*, *17*(5), 230-240. <https://doi.org/10.1016/j.tics.2013.03.005>
- Purcell, S. M., Manoach, D. S., Demanuele, C., Cade, B. E., Mariani, S., Cox, R., Panagiotaropoulou, G., Saxena, R., Pan, J. Q., Smoller, J. W., Redline, S., & Stickgold, R. (2017). Characterizing sleep spindles in 11,630 individuals from the National Sleep Research Resource. *Nature Communications*, *8*(1), 15930. <https://doi.org/10.1038/ncomms15930>

- Rasch, B., Buchel, C., Gais, S., & Born, J. (2007). Odor Cues During Slow-Wave Sleep Prompt Declarative Memory Consolidation. *Science*, *315*(5817), 1426-1429.
<https://doi.org/10.1126/science.1138581>
- Rickard, T. C., Cai, D. J., Rieth, C. A., Jones, J., & Ard, M. C. (2008). Sleep does not enhance motor sequence learning. *Journal of Experimental Psychology: Learning, Memory, and Cognition*, *34*(4), 834-842. <https://doi.org/10.1037/0278-7393.34.4.834>
- Rickard, T. C., & Pan, S. C. (2017). Time for considering the possibility that sleep plays no unique role in motor memory consolidation : Reply to Adi-Japha and Karni (2016). *Psychological Bulletin*, *143*(4), 454-458. <https://doi.org/10.1037/bul0000094>
- Rihs, T., Michel, C., & Thut, G. (2009). A bias for posterior α -band power suppression versus enhancement during shifting versus maintenance of spatial attention. *NeuroImage*, *44*(1), 190-199. <https://doi.org/10.1016/j.neuroimage.2008.08.022>
- Robertson, E. M., Pascual-Leone, A., & Press, D. Z. (2004). Awareness Modifies the Skill-Learning Benefits of Sleep. *Current Biology*, *14*(3), 208-212. <https://doi.org/10.1016/j.cub.2004.01.027>
- Rosinvil, T., Bouvier, J., Dubé, J., Lafrenière, A., Bouchard, M., Cyr-Cronier, J., Gosselin, N., Carrier, J., & Lina, J.-M. (2020). Are age and sex effects on sleep slow waves only a matter of electroencephalogram amplitude? *Sleep*, *zsa186*. <https://doi.org/10.1093/sleep/zsa186>
- Schendan, H. E., Searl, M. M., Melrose, R. J., & Stern, C. E. (2003). An fMRI Study of the Role of the Medial Temporal Lobe in Implicit and Explicit Sequence Learning. *Neuron*, *37*(6), 1013-1025.
[https://doi.org/10.1016/S0896-6273\(03\)00123-5](https://doi.org/10.1016/S0896-6273(03)00123-5)
- Schönauer, M., Geisler, T., & Gais, S. (2014). Strengthening Procedural Memories by Reactivation in Sleep. *Journal of Cognitive Neuroscience*, *26*(1), 143-153. https://doi.org/10.1162/jocn_a_00471
- Schreglmann, S. R., Wang, D., Peach, R. L., Li, J., Zhang, X., Latorre, A., Rhodes, E., Panella, E., Cassara, A. M., Boyden, E. S., Barahona, M., Santaniello, S., Rothwell, J., Bhatia, K. P., & Grossman, N. (2021).

- Non-invasive suppression of essential tremor via phase-locked disruption of its temporal coherence. *Nature Communications*, 12(1), 363. <https://doi.org/10.1038/s41467-020-20581-7>
- Spencer, R. M. C., Gouw, A. M., & Ivry, R. B. (2007). Age-related decline of sleep-dependent consolidation. *Learning & Memory*, 14(7), 480-484. <https://doi.org/10.1101/lm.569407>
- Staresina, B. P. (2024). Coupled sleep rhythms for memory consolidation. *Trends in Cognitive Sciences*, 28(4), 339-351. <https://doi.org/10.1016/j.tics.2024.02.002>
- Strange, B. A., Fletcher, P. C., Henson, R. N. A., Friston, K. J., & Dolan, R. J. (1999). Segregating the functions of human hippocampus. *Proceedings of the National Academy of Sciences*, 96(7), 4034-4039. <https://doi.org/10.1073/pnas.96.7.4034>
- Tallon-Baudry, C. (2003). Oscillatory synchrony and human visual cognition. *Journal of Physiology-Paris*, 97(2-3), 355-363. <https://doi.org/10.1016/j.jphysparis.2003.09.009>
- Tallon-Baudry, C., & Bertrand, O. (1999). Oscillatory gamma activity in humans and its role in object representation. *Trends in cognitive sciences*, 3(4), 151-162.
- Thielen, J.-W., Takashima, A., Rutters, F., Tendolkar, I., & Fernández, G. (2015). Transient relay function of midline thalamic nuclei during long-term memory consolidation in humans. *Learning & Memory*, 22(10), 527-531. <https://doi.org/10.1101/lm.038372.115>
- Thut, G. (2006). Band Electroencephalographic Activity over Occipital Cortex Indexes Visuospatial Attention Bias and Predicts Visual Target Detection. *Journal of Neuroscience*, 26(37), 9494-9502. <https://doi.org/10.1523/JNEUROSCI.0875-06.2006>
- Vallat, R., & Walker, M. P. (2021). An open-source, high-performance tool for automated sleep staging. *eLife*, 10, e70092. <https://doi.org/10.7554/eLife.70092>
- Van Der Graaf, F. H. C. E., De Jong, B. M., Maguire, R. P., Meiners, L. C., & Leenders, K. L. (2004). Cerebral activation related to skills practice in a double serial reaction time task : Striatal involvement in

- random-order sequence learning. *Cognitive Brain Research*, 20(2), 120-131.
<https://doi.org/10.1016/j.cogbrainres.2004.02.003>
- Walker, M. P., Brakefield, T., Allan Hobson, J., & Stickgold, R. (2003). Dissociable stages of human memory consolidation and reconsolidation. *Nature*, 425(6958), 616-620.
<https://doi.org/10.1038/nature01930>
- Walker, M. P., Brakefield, T., Morgan, A., Hobson, J. A., & Stickgold, R. (2002). Practice with Sleep Makes Perfect. *Neuron*, 35(1), 205-211. [https://doi.org/10.1016/S0896-6273\(02\)00746-8](https://doi.org/10.1016/S0896-6273(02)00746-8)
- Walker, M. P., Brakefield, T., Seidman, J., Morgan, A., Hobson, J. A., & Stickgold, R. (2003). Sleep and the Time Course of Motor Skill Learning. *Learning & Memory*, 10(4), 275-284.
<https://doi.org/10.1101/lm.58503>
- Walker, M. P., Stickgold, R., Alsop, D., Gaab, N., & Schlaug, G. (2005). Sleep-dependent motor memory plasticity in the human brain. *Neuroscience*, 133(4), 911-917.
<https://doi.org/10.1016/j.neuroscience.2005.04.007>
- Yokoi, A., Arbuckle, S. A., & Diedrichsen, J. (2018). The Role of Human Primary Motor Cortex in the Production of Skilled Finger Sequences. *The Journal of Neuroscience: The Official Journal of the Society for Neuroscience*, 38(6), 1430-1442. <https://doi.org/10.1523/JNEUROSCI.2798-17.2017>
- Yokoi, A., & Diedrichsen, J. (2019). Neural Organization of Hierarchical Motor Sequence Representations in the Human Neocortex. *Neuron*, 103(6), 1178-1190.e7.
<https://doi.org/10.1016/j.neuron.2019.06.017>